# Personalized Federated Learning towards Communication Efficiency, Robustness and Fairness

**Shiyun Lin**[1,2] *   **Yuze Han**[2] *   **Xiang Li**[2]   **Zhihua Zhang**[1,2]
[1]Center for Statistical Science, Peking University
[2]School of Mathematical Sciences, Peking University
shiyunlin@stu.pku.edu.cn   hanyuze97@pku.edu.cn
lx10077@pku.edu.cn   zhzhang@math.pku.edu.cn

## Abstract

Personalized Federated Learning faces many challenges such as expensive communication costs, training-time adversarial attacks, and performance unfairness across devices. Recent developments witness a trade-off between a reference model and local models to achieve personalization. We follow the avenue and propose a personalized FL method towards the three goals. When it is time to communicate, our method projects local models into a shared-and-fixed low-dimensional random subspace and uses infimal convolution to control the deviation between the reference model and projected local models. We theoretically show our method converges for smooth objectives with square regularizers and the convergence dependence on the projection dimension is mild. We also illustrate the benefits of robustness and fairness on a class of linear problems. Finally, we conduct a large number of experiments to show the empirical superiority of our method over several state-of-the-art methods on the three aspects.

## 1   Introduction

Federated Learning (FL) emerges as a new distributed computing paradigm that would perform privately distributed optimization in large-scale networks of remote clients [51]. For the sake of privacy protection, data are generated locally and are kept in the original location during training, which incurs a discrepancy among local data distributions. Furthermore, the nature that FL works as a decentralized system poses greater challenges to its communication efficiency, robustness against adversarial attacks, and fairness on resource allocation [32].

To detour the data heterogeneity issue, one considers to personalize local models [35]. A key feature that any personalization method has is to differ local models from the global model. The simplest way for personalization is training purely with local data on each device. Chen et al. [10] showed that when the degree of data heterogeneity is above some threshold value, pure local training is minimax optimal; otherwise, the global model is minimax optimal. In practice, we prefer a method that intervenes between the two extremes. It brings out another popular approach that interpolates between a reference model and local models [19, 20, 12, 13, 26, 64]. Recently, Li et al. [44] suggested that personalization can be leveraged not only to improve accuracy, but also to allow for competing constraints such as robustness and fairness. Inspired by their work, we would explore the following question:

*Can we balance different constraints of interest (i.e., communication efficiency, robustness and fairness) simultaneously?*

In this paper, we give an affirmative answer to the question by proposing a personalized FL method named as `lp-proj`, whose core is $L^p$-regularization and low-dimensional random projection. We

---

*Equal contribution.

36th Conference on Neural Information Processing Systems (NeurIPS 2022).

employ the idea of controlling the dissimilarity between the global model and local models via a smoothing kernel of infimal convolution. Towards the three goals, the smoothing kernel is designed to regularize the projection of local models in a shared low-dimensional random subspace rather than the original space. In this way, each client only communicates the projected models each time and the server maintains a low-dimensional reference model for regularization. The random subspace is generated once and will not change during training. It makes local models share a similar part in the random subspace and adapt to their local data using components beyond that.

Theoretically, we give convergence analysis for smooth objectives with square regularizer, and show that the convergence dependence on the projection dimension is mild. By analyzing test losses and the corresponding variances across the network on federated linear regression, we show that our proposed method is at least as good as two SOTA methods [13, 44] in terms of Byzantine robustness [37] and performance fairness (see Definition 1).

Finally, we conduct a large number of experiments to show the empirical superiority of our method. It not only shows expectant personalization but also promotes fairness by achieving higher test accuracy more uniformly over clients. It significantly improves communication efficiency, because the subspace dimension is often no more than one-hundredth of the original dimension. Furthermore, it is more resistant to training-time adversarial attacks.

In summary, we propose a personalized FL algorithm and explore its performance in aspects of communication efficiency, robustness and fairness. Our results show that low-dimensional projection brings multiple benefits and is helpful for algorithm design.

## 2    Related Work

There are many works studying personalization and a survey can be found in [35]. It has been studied via multi-task learning [57, 28], meta-learning [9, 31, 16], knowledge distillation [41, 70] and transfer learning  [63, 49].  Hanzely et al. [21] provided convergence analysis for a general personalized framework that requires jointly strongly convex and smooth objectives.

**Communication Efficiency**    To reduce the cost of communication in FL with large-scale networks, existing works could be categorized as gradient compression, model compression and reducing the communication frequency. On gradient compression, sparsification [29, 47], quantization [1] and low-rank approximation [3] are three main directions investigated in the current research. On model compression, Liang et al. [46] suggested learning local representations and a global model only operates on the local representations, Li et al. [38] extended the lottery ticket hypothesis and used network pruning in the FL setting. On communication frequency, Wang et al. [62] used momentum to delay the global aggregation; McMahan et al. [51], Karimireddy et al. [33] performed multiple local updates to lessen the communication rounds. Shahid et al. [56] presented a survey on current progress on communication efficiency in FL. Our proposed method is a model compression approach. Differing from previous works that compress the model each time with different basis, our work focus on a shared-and-fixed low-dimensional subspace which is determined at the beginning of training and will not change later on.

**Fairness in FL**    Zhou et al. [71] suggested that there are three types of fairness in FL: performance fairness, collaboration fairness and model fairness. On performance fairness, an FL system usually promotes uniform accuracy distribution across participants, which is closely related to resource allocation by viewing FL as a joint optimization system over a heterogeneous network. Li et al. [44] gave a formal definition (Definition 1) and some efficient methods have been proposed towards this goal [44, 27].

**Definition 1** (Performance Fairness [44]). *A model $\mathbf{w}_1$ is more fair than $\mathbf{w}_2$ if the test performance distribution of $\mathbf{w}_1$ across the network with $N$ clients is more uniform than that of $\mathbf{w}_2$, i.e.* $\mathrm{var}\left\{F_k(\mathbf{w}_1)\right\}_{k\in[N]} < \mathrm{var}\left\{F_k(\mathbf{w}_2)\right\}_{k\in[N]}$*, where $F_k(\cdot)$ denotes the test loss of client $k \in [N]$ and* var *denotes the variance*[2].

On collaboration fairness, one expects that each participant receives a reward that can fairly reflect its contribution to the FL system, in which way one can build a sound incentive mechanism. Lyu et al. [48] formalized this idea; Yu et al. [69], Xu and Lyu [67] explored on this aspect. On model fairness, one usually concerns ethical issues and seeks to protect some sensitive attributes [15, 22, 36]. Liang et al. [46] suggested learning a fair representation for each client to achieve fairness, Du et al.

---

[2]Equivalently, we can use the standard deviation (std) to measure fairness across the network.

[14] proposed reweighing the objective functions under fairness constraint. In our work, we focus on performance fairness, illustrating the benefits of our method through theoretical analysis and numerical experiments.

**Robust FL**  Typical adversarial attacks include data poisoning and model update poisoning (Byzantine attacks). The former injects abnormal sample points into the training dataset [5, 30, 39, 55, 59, 65, 17], while the latter manipulates communication messages by sending arbitrary updates to the server, both attacks hindering the training processes. In this paper, we mainly aim to defend model update poisoning attacks and achieve Byzantine robustness [37]. An extension to a data poisoning attack is also considered in numerical experiments. Prior works favor Byzantine-robust SGD variants where the server alleviates the attack of Byzantine clients via robust aggregation rules [11, 68, 66, 7]. Beyond that, Li et al. [42] developed a robust distributed method by incorporating the $L^p$-norm regularizer to robustify the objective function. Li et al. [44] used robust aggregation rules to achieve robustness and fairness. Our work leverages the ideas from [42] and [44], but differs from them by embedding the update process in a low-dimensional fixed random subspace. Theoretical analysis shows that with commonly used regularization parameters, our method is no worse than two SOTA methods [13, 44] (see Figure 1). Extensive experiments manifest our method of achieving state-of-the-art performance under various types and intensities of adversarial attacks.

## 3   Methodology

In this section, we present our method which is based on *infimal convolution* and *subspace regularization*. Conventional FL that trains a single global model to fit the "average client" suffers from statistical heterogeneity among massive devices. To enhance accuracy performance, we hope not only to leverage the global model, but also stylize it to fit the local data for each client. To this end, we use *infimal convolution*, which is originally proposed to smooth some extended real-valued convex function $f$ with a sufficiently smooth kernel function $g$ [52]. We apply this technique in FL to bridge local models and global model. Here, $f$ is the usual objective function in the vanilla case, and $g$ is designed to characterize the relationship between local and global models. Given a general function $g$ as the smoothing kernel, the personalized FL using infimal convolution is then formulated as a bi-level problem:

$$\min_{\mathbf{w}\in\mathbb{R}^d} F(\mathbf{w}) := G\left\{F_1(\mathbf{w}), \ldots, F_N(\mathbf{w})\right\}, \tag{1}$$

where $G(\cdot)$ is the aggregation function at the server side[3]. For $k \in \{1, \cdots, N\}$,

$$F_k(\mathbf{w}) = \{f_k \otimes \lambda g\}(\mathbf{w}) := \min_{\mathbf{x}_k \in \mathbb{R}^d} f_k(\mathbf{x}_k) + \lambda g(\mathbf{w} - \mathbf{x}_k) \text{ with } f_k(\mathbf{x}_k) = \mathbb{E}_{\xi_k}\left[\tilde{f}_k(\mathbf{x}_k;\xi_k)\right]. \tag{2}$$

Here, $\otimes$ denotes infimal convolution operator, $\xi_k$ is an independent sample drawn from the distribution $\mathcal{D}_k$, and $\tilde{f}_k(\mathbf{x}_k;\xi_k)$ is the loss function corresponding to this sample. $\mathbf{w}$ and $\mathbf{x}_k$ represent the global and local model parameters, respectively. $\lambda$ is a hyperparameter controlling the degree of personalization. Problem (1) is pure local training if $\lambda = 0$, and is synchronized training when $\lambda \to \infty$.

The smoothing kernel function $g$ is task-specific. Many previous personalized methods can be cast into our infimal convolution framework by setting a proper $g$. For example, Dinh et al. [13], Li et al. [44] used Moreau Envelopes as the regularizer, which is equivalent to $g(\cdot) = \frac{1}{2}\|\cdot\|_2^2$. Li et al. [42] proposed the $L^p$-norm regularization $g(\cdot) = \|\cdot\|_p$ instead. Motivated by the fact that high-dimensional data usually has low-dimensional representation that retains meaningful properties [60], and random projection would preserve similarity of data vectors [6], we propose to regularize the projection of local models in a shared low-dimensional space, which is equivalent to the following smoothing kernel

$$g(\cdot) = \frac{1}{p}\|\boldsymbol{P}(\cdot)\|_p^p, \tag{3}$$

where $p \geq 1$ and $\boldsymbol{P}$ is a $d_{\mathrm{sub}} \times d$ random matrix that is generated initially and will not vary anymore. $d_{\mathrm{sub}}$ is the dimension of the shared-and-fixed random subspace. The choice for $\boldsymbol{P}$ is flexible as the only requirement in our theory is that all the singular values of $\boldsymbol{P}$ are bounded from both sides. In this paper, we consider that $\boldsymbol{P}$ is generated with i.i.d. Gaussian entries and then normalized to have unit $L^2$ norm for each row as suggested by [40]. We comment that with this $g$, $F_k(\mathbf{w})$ is actually a function

---

[3]For simplicity, we set $G(\cdot)$ as the simple average $\frac{1}{N}\sum_{k=1}^N F_k(\mathbf{w})$, but it can also generalize to other forms.

**Algorithm 1** `lp-proj`: Projection-based $L^p$ Regularized Personalized Federated Learning

---

1: **Input**: Communication rounds $T$, local update rounds $R$, client sampling size $S$, regularization coefficient $\lambda$, lower-level problem accuracy $\nu$, step size $\eta$, initial global model $\tilde{\mathbf{w}}_0$, projection matrix $\boldsymbol{P}$, speedup control parameter $\beta$.
2: **for** $t = 0$ to $T - 1$ **do**
3:     Server sends $\tilde{\mathbf{w}}_t$ to all clients.
4:     **for** all $k = 1$ to $N$ clients **do**
5:         $\tilde{\mathbf{w}}_{k,0}^t = \tilde{\mathbf{w}}_t$.
6:         **for** $r = 0$ to $R - 1$ **do**
7:             Independently sample a fresh mini-batch $\tilde{\mathcal{D}}_k$ and minimize the loss function (4) up to accuracy level $\nu$ to get $\mathbf{x}_{k,r}^t$.
8:             Update the local model $\tilde{\mathbf{w}}_{k,r+1}^t$ by (5).
9:         **end for**
10:     **end for**
11:     Server uniformly samples a subset of clients $\mathcal{S}_t$ of size $S$. Each client sends $\tilde{\mathbf{w}}_{k,R}^t$ to the server.
12:     Server updates the global model via    $\tilde{\mathbf{w}}_{t+1} = (1 - \beta)\tilde{\mathbf{w}}_t + \beta \sum_{k \in \mathcal{S}_t} \frac{\tilde{\mathbf{w}}_{k,R}^t}{S}$.
13: **end for**

---

of $\tilde{\mathbf{w}} = \boldsymbol{P}\mathbf{w}$ since $\boldsymbol{P}(\mathbf{w} - \mathbf{x}_k) = \tilde{\mathbf{w}} - \boldsymbol{P}\mathbf{x}_k$. It implies we can only focus on the low-dimensional parameter $\tilde{\mathbf{w}} \in \mathbb{R}^{d_{\text{sub}}}$ at the global level for algorithm description and theoretical analysis.[4]

### 3.1 The Algorithm

In this section, we introduce the algorithm `lp-proj` (see Algorithm 1) for the bi-level optimization problem (1) with smoothing kernel $g$ given by Eqn. (3).

The algorithm `lp-proj` is essentially an alternative minimization method on bi-level optimization. Each client $k$ maintains two parameters: their local parameter $\mathbf{x}_{k,r}^t$ and a copy of the global parameter $\tilde{\mathbf{w}}_{k,r}^t$ with additional subscript $r$ denoting inner iterations and superscript $t$ the communication round. At round $t$, the server broadcasts the latest global model $\tilde{\mathbf{w}}_t$ to all clients. Then each client initializes their version of global model $\tilde{\mathbf{w}}_{k,0}^t$ as $\tilde{\mathbf{w}}_t$ (line 5) and starts to solve the problem via alternative minimization (lines 6-9).

- (line 7) Given a local version of global model $\tilde{\mathbf{w}}_{k,r}^t$, we use gradient descent (GD) to obtain an approximate solution $\mathbf{x}_{k,r}^t$ that minimizes $\tilde{h}_k$ up to accuracy level $\nu$, where

$$\tilde{h}_k(\mathbf{x}_k; \tilde{\mathcal{D}}_k, \tilde{\mathbf{w}}_{k,r}^t) = \frac{1}{|\tilde{\mathcal{D}}_k|} \sum_{\xi_{k,i} \in \tilde{\mathcal{D}}_k} \tilde{f}_k(\mathbf{x}_k; \xi_{k,i}) + \lambda \frac{1}{p} \left\| \tilde{\mathbf{w}}_{k,r}^t - \boldsymbol{P}\mathbf{x}_k \right\|_p^p. \tag{4}$$

  Here $\tilde{\mathcal{D}}_k$ is a mini-batch sampled uniformly and $\xi_{k,i}$ refers to a sample from $\tilde{\mathcal{D}}_k$. The GD iteration is terminated when $\left\| \nabla \tilde{h}_k(\mathbf{x}_{k,r}^t; \tilde{\mathcal{D}}_k, \tilde{\mathbf{w}}_{k,r}^t) \right\|_2^2 \le \nu$ is satisfied.

- (line 8) Given a local parameter $\mathbf{x}_{k,r}^t$, the local version of global model $\tilde{\mathbf{w}}_{k,r}^t$ is updated by one-step gradient descent:

$$\tilde{\mathbf{w}}_{k,r+1}^t = \tilde{\mathbf{w}}_{k,r}^t - \frac{\eta\lambda}{p} \partial_{\tilde{\mathbf{w}}_{k,r}^t} \left\| \tilde{\mathbf{w}}_{k,r}^t - \boldsymbol{P}\mathbf{x}_{k,r}^t \right\|_p^p. \tag{5}$$

After $R$ steps of the alternative updates, each client has its own version of the global model $\tilde{\mathbf{w}}_{k,R}^t$. Then the server accesses a random set of $S$ clients and produces the next global model by a linear combination of the latest $\tilde{\mathbf{w}}_t$ and the average of $\{\tilde{\mathbf{w}}_{k,R}^t\}_{k \in \mathcal{S}_t}$. Here, a hyperparameter $\beta$, which could be viewed as a global step-size, is introduced to control the global update process. Our theorem shows a proper $\beta$ can speed up convergence, but in practice, we find that the test performance only varies moderately for different choices of $\beta$. For simplicity, we only consider $\beta = 1$ in our experiments.

**Communication Efficiency**   In Algorithm 1, we restrict the global model $\tilde{\mathbf{w}}_t$ to lie in a fixed low-dimensional subspace, only $\tilde{\mathbf{w}}_{k,R}^t$ of dimension $d_{\text{sub}}$, instead of the full model $\mathbf{x}_{k,r}^t$ of dimension $d$, is communicated to the server during each round, which leads to much fewer bits for communication

---

[4]Without ambiguity, we term $\tilde{\mathbf{w}}$ as the global model.

compared to vanilla FL. Besides, we remark on the difference between our method and other projection/sketching-based methods. On one hand, distributed sketching [4], which directly projects the data in a low-dimensional space at the start of training, is "one-shot" rather than iterative, while our method projects local models every communication round and the local training proceeds with the full model. On the other hand, sketched-SGD [29] compresses the transmitted messages with different basis every time, while our random subspace is specified at the beginning and would not change after that.

**Robustness and Fairness**    For one thing, by applying projection into a low-dimensional subspace, our method only requires (near) consensus of model parameters of different clients in the low-dimensional subspace, leaving flexibility for the system towards personalization and better generalization to the local data distribution, which could improve performance fairness and robustness when facing adversarial attacks. For the other, introducing $L^p$-norm regularized term to the objective function is equivalent to launching an uncertainty set to the model parameter by rewriting the objective as a constrained optimization problem (e.g., $L^1$-norm is the diamond-shaped uncertainty and $L^2$-norm is the spherical uncertainty), in which way we can enhance accuracy by searching for a model adaptive to the local data distribution in the uncertainty set. Formal analysis on a class of linear problems is provided in Section 4.2.

## 4  Theoretical Analysis

### 4.1  Convergence

In this subsection, we establish the convergence of our algorithm for the case $p = 2$. We first present the assumptions and then state our main results.

**Assumption 1** (Smoothness). $\tilde{f}_k$ *is $L$-smooth, that is, for any* $\mathbf{x}_k, \mathbf{x}'_k \in \mathbb{R}^d$ *and $\xi_k$, we have* $\left\| \nabla \tilde{f}_k(\mathbf{x}'_k; \xi_k) - \nabla \tilde{f}_k(\mathbf{x}_k; \xi_k) \right\|_2 \leq L \left\| \mathbf{x}'_k - \mathbf{x}_k \right\|_2.$

**Assumption 2** (Bounded variance). *The variance of stochastic gradients in each client is bounded, i.e.,* $\mathbb{E}_{\xi_k} \left\| \nabla \tilde{f}_k(\mathbf{x}_k; \xi_k) - \nabla f_k(\mathbf{x}_k) \right\|_2^2 \leq \gamma_f^2.$

**Assumption 3** (Bounded diversity). *The variance of local gradients to global gradient is bounded, i.e.,* $\frac{1}{N} \sum_{k=1}^N \| \nabla f_k(\mathbf{w}) - \nabla f(\mathbf{w}) \|_2^2 \leq \sigma_f^2.$

Assumptions 1, 2 and 3 are standard for convergence analysis. Assumption 1 also guarantees the smoothness of $f_k$. From the generation of $\boldsymbol{P}$, we can show that $\boldsymbol{P}$ has full row rank with high probability (see Proposition 7). Then there exists a $d \times (d - d_{\text{sub}})$ matrix $\boldsymbol{Q}$ such that $(\boldsymbol{P}^\top, \boldsymbol{Q})$ is an invertible matrix and $\boldsymbol{PQ} = \mathbf{0}_{d_{\text{sub}} \times (d - d_{\text{sub}})}$.

**Assumption 4** (Low-dimensional condition). $\tilde{f}_k$ *has a low-dimensional structure, that is,* $\tilde{f}_k(\boldsymbol{P}^\top \mathbf{y}_k + \boldsymbol{Q}\tilde{\mathbf{y}}_k; \xi_k) = \tilde{f}_k(\boldsymbol{P}^\top \mathbf{y}_k; \xi_k)$ *for any* $\mathbf{y}_k \in \mathbb{R}^{d_{\text{sub}}}, \tilde{\mathbf{y}}_k \in \mathbb{R}^{d - d_{\text{sub}}}$. *As a consequence, the same equality also holds with $\tilde{f}_k$ replaced by $f_k$.*

Assumption 4 implies that $\min_{\mathbf{x}_k \in \mathbb{R}^d} \tilde{f}_k(\mathbf{x}_k; \xi_k) = \min_{\mathbf{x}_k \in \text{col}(\boldsymbol{P}^\top)} \tilde{f}_k(\mathbf{x}_k; \xi_k)$, where $\text{col}(\boldsymbol{A})$ denotes the subspace spanned by the column vectors of $\boldsymbol{A}$. This means we can focus on the low-dimensional subspace spanned by the row vectors of $\boldsymbol{P}$. We give an example satisfying the assumption. Suppose $\xi_k$ and $\mathbf{x}_k$ have the same dimensions and $\tilde{f}_k(\mathbf{x}_k; \xi_k) = l(\xi_k^\top \mathbf{x}_k)^5$. If $\xi_k \in \text{col}(\boldsymbol{P}^\top)$, then there exists an $\mathbf{a}_k$ such that $\xi_k = \boldsymbol{P}^\top \mathbf{a}_k$. Decompose $\mathbf{x}_k = \boldsymbol{P}^\top \mathbf{y}_k + \boldsymbol{Q}\tilde{\mathbf{y}}_k$. Then $l(\xi_k^\top \mathbf{x}_k) = l(\mathbf{a}_k \boldsymbol{P}(\boldsymbol{P}^\top \mathbf{y}_k + \boldsymbol{Q}\tilde{\mathbf{y}}_k)) = l(\mathbf{a}_k \boldsymbol{P} \boldsymbol{P}^\top \mathbf{y}_k)$. This implies that for linear models with data lying in $\text{col}(\boldsymbol{P}^\top)$, Assumption 4 holds.

For the general case, it is not easy to verify Assumption 4 directly. Intuitively, we can interpret Assumption 4 as that the data concentrate on a low-dimensional subspace. Then with the total parameters denoted by $\mathbf{x}_k = \boldsymbol{P}^\top \mathbf{y}_k + \boldsymbol{Q}\tilde{\mathbf{y}}_k$, only a low-dimensional linear combination $\mathbf{y}_k = (\boldsymbol{P}\boldsymbol{P}^\top)^{-1} \boldsymbol{P} \mathbf{x}_k$ can affect the value of $\tilde{f}_k$.

Recall that we can view $F_k$ as a function of $\tilde{\mathbf{w}}$ instead of $\mathbf{w}$, with some abuse of notation, we write $F_k$ as $F_k(\tilde{\mathbf{w}}) = \min_{\mathbf{x}_k \in \mathbb{R}^d} \left\{ f_k(\mathbf{x}_k) + \frac{\lambda}{2} \| \tilde{\mathbf{w}} - \boldsymbol{P} \mathbf{x}_k \|_2^2 \right\}$. Then we have the following result.

---

[5]For example, when we fit generalized linear models via maximum likelihood method, the negative (log) likelihood function has this form.

**Lemma 1.** *Suppose that Assumptions 1, 2 and 4 hold and $\lambda > 4L$. With probability at least $1 - 2\exp(-cd_{\mathrm{sub}})$, we have*

$$\frac{1}{\lambda^2}\mathbb{E}\left[\left\|\nabla F_k(\tilde{\mathbf{w}}_{k,r}^t) - \lambda(\tilde{\mathbf{w}}_{k,r}^t - \boldsymbol{P}\mathbf{x}_{k,r}^t)\right\|_2^2\right] \leq \delta^2 := \frac{2\left(1 + C\sqrt{\frac{d_{\mathrm{sub}}}{d}}\right)^6\left(\frac{\gamma_f^2}{|\tilde{\mathcal{D}}_k|} + \nu\right)}{\left[\left(1 - C\sqrt{\frac{d_{\mathrm{sub}}}{d}}\right)^4\lambda - \left(1 + C\sqrt{\frac{d_{\mathrm{sub}}}{d}}\right)^2 L\right]^2},$$

*as long as $C\sqrt{\frac{d_{\mathrm{sub}}}{d}} < 1/30$, where $C, c$ are positive constants.*

Lemma 1 quantifies the error between the exact gradient $\nabla F_k(\tilde{\mathbf{w}}_{k,r}^t)$ and the approximate gradient $\lambda(\tilde{\mathbf{w}}_{k,r}^t - \boldsymbol{P}\mathbf{x}_{k,r}^t)$ due to mini-batch sampling and optimization error of the inner loop. From the expression of $\delta^2$, we can see that this error has a mild dependence on $d_{\mathrm{sub}}$.

Based on Lemma 1, we can establish the convergence of the global model and personalized parameters of Algorithm 1. By the smoothness of $f_k$, we can show that $F_k$ is $L_F$-smooth with $L_F = \lambda^6$. Then the convergence of Algorithm 1 is guaranteed by Theorem 1.

**Theorem 1.** *Suppose that Assumptions 1 to 4 hold and $d_{\mathrm{sub}}/d$ is sufficiently small. Let $\hat{\eta}_0 = \frac{1}{90\lambda^2 L_F}$ with $\lambda \geq \max\{\sqrt{10L^2 + 1}, 4L\}$, $\beta \geq 1$ and $\Delta_F = F(\tilde{\mathbf{w}}_0) - \min_{\tilde{\mathbf{w}} \in \mathbb{R}^{d_{\mathrm{sub}}}} F(\tilde{\mathbf{w}})$. If $t^*$ is uniformly sampled from $\{0, 1, \ldots, T-1\}$, then with probability at least $1 - 2\exp(-cd_{\mathrm{sub}})$, there exists $\eta \leq \frac{\hat{\eta}_0}{\beta R}$ such that*

$$\mathbb{E}\left[\|\nabla F(\tilde{\mathbf{w}}_{t^*})\|_2^2\right] \leq \underbrace{\mathcal{O}\left(\frac{\Delta_F}{\hat{\eta}_0 T}\right)}_{\text{due to initialization}} + \underbrace{\mathcal{O}\left(\frac{(\Delta_F L_F \sigma_F^2(N/S - 1))^{1/2}}{\sqrt{TN}}\right)}_{\text{due to client sampling}}$$

$$+ \underbrace{\mathcal{O}\left(\frac{(\Delta_F L_F)^{2/3}(\sigma_F^2 + \lambda^2\delta^2)^{1/3}}{(\beta T)^{2/3}}\right) + \mathcal{O}\left(\lambda^2\delta^2\right)}_{\text{due to client drift with multiple local updates and approximation errors}} =: \mathcal{O}_0, \tag{6}$$

*where $c$ is a positive constant number, $\sigma_F^2 = \frac{\lambda^2\sigma_f^2}{\lambda^2 - 10L^2}$ measures the bounded diversity of $F_k$, $\delta^2$ defined in Lemma 1 is the approximation error of the inner loop, the expectation is w.r.t. all the randomness except for $\boldsymbol{P}$ and $\mathcal{O}$ hides constants. Moreover, suppose that $\mathbf{x}_k^t$ is a solution to $\left\|\nabla\tilde{h}_k(\mathbf{x}; \tilde{\mathcal{D}}_k, \tilde{\mathbf{w}}_t)\right\|_2^2 \leq \nu$. Then with probability at least $1 - 2\exp(-cd_{\mathrm{sub}})$, we have*

$$\frac{1}{N}\sum_{k=1}^N \mathbb{E}\left[\left\|\boldsymbol{P}\mathbf{x}_k^{t^*} - \tilde{\mathbf{w}}_{t^*}\right\|_2^2\right] \leq \mathcal{O}_0 + \mathcal{O}\left(\frac{\sigma_F^2}{\lambda^2} + \delta^2\right).$$

When there is no client sampling, choosing $\beta = \Theta(NR)$ leads to a sublinear speedup $\mathcal{O}\left(1/(TRN)^{2/3}\right)$. Eqn. (6) shows the average over indices $k$ and $t$ of the distance between personalized parameters (after a linear transformation) and global model parameters converges to $\mathcal{O}\left(\lambda^2\delta^2 + \frac{\sigma_F^2}{\lambda^2} + \delta^2\right)$. Here $\lambda$ can be chosen to trade off different terms.

Our Theorem 1 shares similar forms as Theorem 2 in [13]. The constant term $\mathcal{O}\left(\lambda^2\delta^2\right)$ appears in both theorems and is caused by biased gradients, i.e., we only get a biased estimate of $\nabla F_k$ due to inexact inner optimization (non-zero $\nu$) and batch data (small $|\tilde{\mathcal{D}}_k|$).

## 4.2 Robustness and fairness

In this subsection, we explore the robustness and fairness benefits of `lp-proj` on a class of linear problems and compare `lp-proj` with `Ditto` [44] and `pFedMe` [13]. For ease of analysis, we assume the rows of $\boldsymbol{P}$ are orthogonal. In practice, we can directly use the random matrix generated as in Section 3 without explicit orthogonalization, since high-dimensional random vectors are nearly orthogonal.

---

[6]See Proposition 9 in Appendix A.2 for the details.

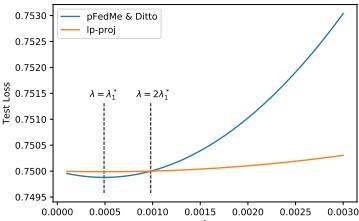
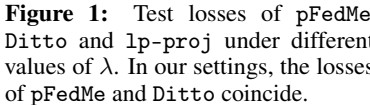

**Figure 1:** Test losses of `pFedMe`, `Ditto` and `lp-proj` under different values of $\lambda$. In our settings, the losses of `pFedMe` and `Ditto` coincide.

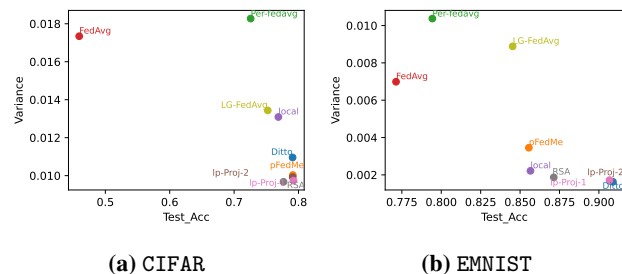

**(a)** CIFAR  **(b)** EMNIST

**Figure 2:** Accuracy-Fairness trade-off of competing methods. (The point closer to the bottom right corner is better.)

**Our Setting**  We focus on a simplified setting where the number of local update steps is infinite, there is only one round of communication and all clients participate in the communication. Then it is natural to set $\beta = 1$. Suppose that the true parameter on client $k$ is $\mathbf{w}_k$, there are $n$ samples on each client and the covariate on client $k$ is $\{\xi_{k,i}\}_{i=1}^n$ and fixed. The observations are generated by $y_{k,i} = \xi_{k,i}^\top \mathbf{w}_k + z_{k,i}$ where the noises $z_{k,i} \overset{\text{i.i.d.}}{\sim} \mathcal{N}(0, \sigma^2)$. For simplicity, we assume $\sum_{i=1}^n \xi_{k,i}\xi_{k,i}^\top = bn\boldsymbol{I}_d$. Then the test loss on client $k$ is $f_k^{\text{te}}(\mathbf{x}_k) = \frac{1}{2n}\sum_{i=1}^n (\xi_{k,i}^\top \mathbf{w}_k + z'_{k,i} - \xi_{k,i}^\top \mathbf{x}_k)^2$, where $z'_{k,i} \sim \mathcal{N}(0, \sigma^2)$ and are independent of $z_{k,i}$.

**Three Attacks**  We examine three types of Byzantine attacks. Denote the message delivered by malicious client $k$ as $\tilde{\mathbf{w}}_k^{(ma)}$, then the attacks are listed as follows.

- **Same-value attacks**: The message sent by a Byzantine client $k$ is set as $\tilde{\mathbf{w}}_k^{(ma)} = c\mathbf{1}_{d_{\text{sub}}}$, where $\mathbf{1}_{d_{\text{sub}}} \in \mathbb{R}^{d_{\text{sub}}}$ is the vector of ones and $c \sim \mathcal{N}(0, \tau^2)$.

- **Sign-flipping attacks**: The transmitted messages are sign-flipped and then scaled, i.e., a Byzantine client $k$ computes the true value $\tilde{\mathbf{w}}_k$ but sends $\tilde{\mathbf{w}}_k^{(ma)} = -|c| \cdot \tilde{\mathbf{w}}_k$ to the server where $c \sim \mathcal{N}(0, \tau^2)$.

- **Gaussian attacks**: The message sent by a Byzantine client $k$ is set as $\tilde{\mathbf{w}}_k^{(ma)} \sim \mathcal{N}(\mathbf{0}_{d_{\text{sub}}}, \tau^2 \boldsymbol{I}_{d_{\text{sub}}})$.

The analyses for different attacks are similar, thus we only focus on the same-value attacks here for illustration. Results for other attacks are deferred to Appendix B.3. Suppose that there are $N_a$ malicious clients, and the heterogeneity is uniform in all dimensions in the sense of Eqn.(25) in Appendix B.3, where we define $\Sigma_1$ to measure data heterogeneity in a single dimension. Let $\lambda_1^* = \frac{(1-1/N)\sigma^2/n}{\Sigma_1 + \frac{N_a}{N^2}(\tau^2 - \sigma^2/(bn))}$. The numerator of $\lambda_1^*$ is the variance of noises over the number of samples. The denominator is the sum of data heterogeneity and variance of attacks. When $\lambda = \lambda_1^*$, `pFedMe`, `Ditto` and `lp-proj` all achieve their corresponding minimal losses. However, we do not know factors affecting $\lambda_1^*$ in advance, implying getting the particular value of $\lambda_1^*$ is possibly hard. Therefore, we need to compare the performance of these methods under different values of $\lambda$.

**Proposition 1** (Formal statement in Theorem 2, Appendix B.3)**.** *Denote the averaged losses on benign clients of pFedMe, Ditto and lp-proj by $L^{Me,\,att1}(\lambda)$, $L^{Di,att1}(\lambda)$ and $L^{l2,att1}(\lambda)$ respectively. The explicit forms of losses are in Appendix B.3. Under the same-value attacks, we have (i) $L^{Me,att1}(\lambda) = L^{Di,att1}(\lambda)$, $\forall \lambda > 0$ and (ii) if $\lambda_1^* < b$, $L^{l2,\,att1}(\lambda) \leq L^{Me,att1}(\lambda)$ if and only if $\lambda \geq \frac{2\lambda_1^*}{1-\lambda_1^*/b}$.*

Porposition 1 implies `lp-proj` outperforms both `pFedMe` and `Ditto` once $\lambda$ is larger than a threshold value $\frac{2\lambda_1^*}{1-\lambda_1^*/b}$, which is slightly larger than $2\lambda_1^*$. The pattern is captured by Figure 1, where we set $n = 200$, $N = 100$, $N_a = 20$, $d = 100$, $d_{\text{sub}} = 10$, $b = 1$, $\sigma = 1$, $\Sigma_1 = 0.1$ and $\tau = 100$. Then $\lambda_1^* \approx 4.9e-4$, a pretty small value. Even for $\lambda < \frac{2\lambda_1^*}{1-\lambda_1^*/b}$, the gap between `lp-proj` and `pFedMe` / `Ditto` is negligible. Thus, `lp-proj` has comparable or beter performance for any $\lambda > 0$.

Now we turn to the performance fairness defined in Definition 1. For simplicity, we further assume that the true parameters $\mathbf{w}_k$ are i.i.d. and distributed as $\mathcal{N}(\mu_w, \boldsymbol{\Sigma}_w)$.

**Proposition 2** (Formal statement in Theorem 3, Appendix B.4)**.** *Denote the variance of test losses on different clients of pFedMe, Ditto and lp-proj by $V^{Me}(\lambda)$, $V^{Di}(\lambda)$ and $V^{l2}(\lambda)$ respectively. We*

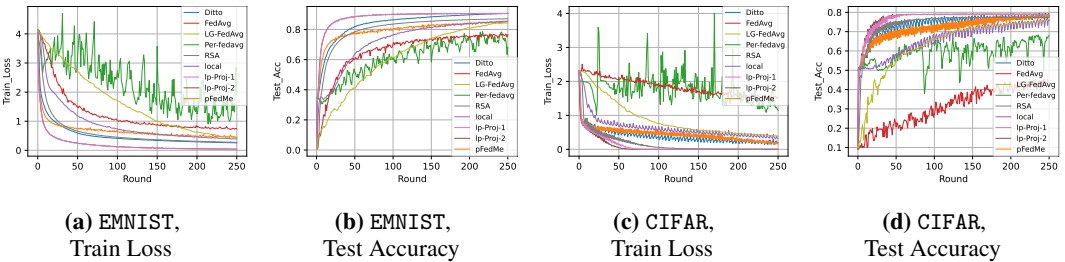

**(a)** `EMNIST`,
Train Loss

**(b)** `EMNIST`,
Test Accuracy

**(c)** `CIFAR`,
Train Loss

**(d)** `CIFAR`,
Test Accuracy

**Figure 3:** Personalization performance of `lp-proj-1`, `lp-proj-2` with other methods on `EMNIST` and `CIFAR`.

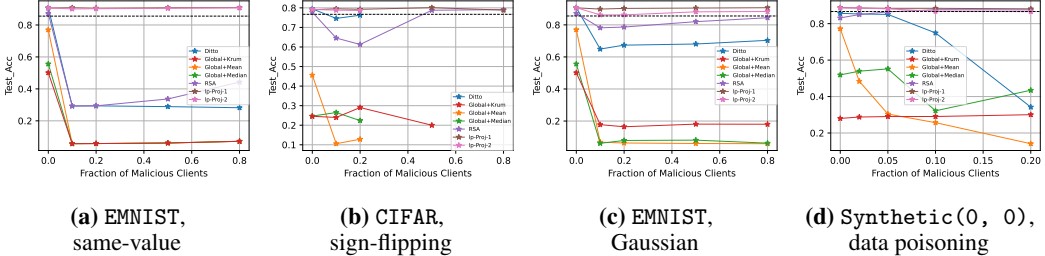

**(a)** `EMNIST`,
same-value

**(b)** `CIFAR`,
sign-flipping

**(c)** `EMNIST`,
Gaussian

**(d)** `Synthetic(0, 0)`,
data poisoning

**Figure 4:** Robustness comparison of different methods, i.e., average test accuracy of benign clients. The dashed black line shows the performance of pure local training. A line with less than 5 points implies the algorithm collapses because the intensity of the given attack exceeds the limit the corresponding algorithm could tolerate.

have for $\forall \lambda > 0$, $\mathbb{E}V^{l2}(\lambda) \leq \mathbb{E}V^{Me}(\lambda) = \mathbb{E}V^{Di}(\lambda)$. *More specifically,* $\mathbb{E}V^{Me}(\lambda) = O(d^2)$ *and* $\mathbb{E}V^{l2}(\lambda) = O(d_{\text{sub}}^2)$, *where the expectation is taken w.r.t. the randomness of* $\mathbf{w}_k$.

Proposition 2 shows `lp-proj` always brings more uniform test losses, no matter what value $\lambda$ is. In particular, $\mathbb{E}V^{l2}(\lambda) = O(d_{\text{sub}}^2)$ while $\mathbb{E}V^{\text{Me}}(\lambda) = O(d^2)$. Since it is likely that $d_{\text{sub}} \ll d$, the advantage of `lp-proj` could be much larger. It implies `lp-proj` is more fair than `pFedMe` and `Ditto`. For the formal theorem, see Appendix B.4.

## 5 Numerical Experiments

In this section, we demonstrate `lp-proj` has the desirable properties through numerical experiments.

**Experimental Setup** We test `lp-proj` as well as other comparable algorithms on six datasets from common ML and FL benchmarks [50, 8]. We consider both convex and non-convex models. For the latter, we consider neural networks including both MLP and CNN. Details about datasets, models and basic information about the FL system are provided in Table 2 in Appendix C. To better model the statistical heterogeneity, we distribute the dataset among clients in a non-iid fashion such that each client only contains partial classes of the data in multi-classification problems.

For each client, the training and testing data are pre-specified as in the ML community, and 20% of training data is randomly extracted to construct a validation set, keeping the remaining 80% as the training set. The training set is used for model fitting and parameter estimation. For each competing method, we use the accuracy performance on the validation set as the tuning criterion and conduct a grid search to choose the best hyper-parameter combination among a prescribed candidate set. All reported results are evaluated on the test dataset. More details about hyperparameter tuning are provided in Appendix C.2. Furthermore, to incorporate partial participation [51, 45], we randomly select 10% of the clients for aggregation at each communication round. The projection dimension of the random subspace for each dataset is chosen based on the full model size and communication budget (see Appendix C.2). Source code for the reproduction of numerical results is available at `https://github.com/desternylin/perfed`.

**Personalization Accuracy Performance** In order to highlight the empirical performance of our proposed method, we compare `lp-proj` with several state-of-the-art personalization methods in the literature, together with a global method and a pure local method. Specifically, we consider the case when $p = 1$ (`lp-proj-1`) and $p = 2$ (`lp-proj-2`). For SOTA methods, we consider `Ditto` [44], `pFedMe` [13], `Per-fedavg` [16], `LG-FedAvg` [46] and `RSA` [42]. A brief description of these approaches is provided in Appendix C.1.

Due to space limitations, we only show the comparisons on training loss and test accuracy on two standard datasets (`EMNIST` and `CIFAR`) in Figure 3. Results on the other datasets are left in Table 3

**Table 1:** Communication performance on `Synthetic(0, 0)` and `EMNIST` datasets. Two aspects are considered: test accuracy on a given byte budget and bytes used to achieve a target test accuracy. A ⋆ on the column "Test Acc" refers to the situation that bytes used in the first iteration of the algorithm have exceeded the budget, and a ⋆ on the column "Used Bytes" means the algorithm could not provide a solution that reaches the target accuracy.

| Method | Synthetic(0, 0) | | | | EMNIST | | | |
| --- | --- | --- | --- | --- | --- | --- | --- | --- |
| | Bytes Budget | Test Acc | Target Acc | Used Bytes | Bytes Budget | Test Acc | Target Acc | Used Bytes |
| FedAvg | 328020 | 0.625 | 0.6 | 597800 | 4236900 | ⋆ | 0.7 | 445851400 |
| Sketch | 328020 | 0.456 | 0.6 | ⋆ | 4236900 | ⋆ | 0.7 | ⋆ |
| lp-proj-1 | 328020 | 0.885 | 0.6 | **4620** | 4236900 | **0.906** | 0.7 | **174720** |
| lp-proj-2 | 328020 | **0.888** | 0.6 | **4620** | 4236900 | **0.906** | 0.7 | 196560 |
| LBGM | 328020 | 0.538 | 0.6 | 365002 | 4236900 | ⋆ | 0.7 | 769902776 |
| QSGD | 328020 | 0.115 | 0.6 | 923350 | 4236900 | ⋆ | 0.7 | 673302175 |
| DGC | 328020 | ⋆ | 0.6 | 391800 | 4236900 | ⋆ | 0.7 | ⋆it |
| LG-FedAvg | \ | \ | \ | \ | 4236900 | 0.071 | 0.7 | 230786010 |

and Figure 5 in Appendix C.3. From these figures, we see that `lp-proj-1` and `lp-proj-2` have comparable or even superior performance than other methods. Furthermore, the training process is more stable as the loss and accuracy curves have less fluctuation.

**Communication Efficiency**    We compare `lp-proj` with the global baseline `FedAvg` [51] and five standard approaches using gradient and model compression, namely `Sketch` [29], `LBGM` [3], `QSGD` [1], `DGC` [47] and `LG-FedAvg` [46]. We quantify the communication cost via total bytes written and read by active clients each round and capture the relation between test accuracy and communicated bytes. Due to space limitations, we only show the results on two datasets in Table 1 (`Synthetic(0, 0)` and `EMNIST`) with the full results left in Table 4 in Appendix C.4.

It is clear that our method is more communication efficient since we only communicate low-dimensional messages. Given a communication budget of bytes, `lp-proj` obtains $\sim 26.3\%$ and $\sim 83.5\%$ test accuracy improvement on `Synthetic(0, 0)` and `EMNIST` datasets respectively. On the other hand, given a target test accuracy, our proposed method needs much fewer bits than the rest and saves the communication cost by `79x` and `1320x` on the two datasets compared with the best competing method. Besides, our method owns flexibility on the choice of the projection dimension $d_{\mathrm{sub}}$, because the convergence dependence of our method on $d_{\mathrm{sub}}$ is mild as predicted by Lemma 1. The compression rate of our proposed methods can be `1000x` or even higher, while that of sketching or gradient compression methods typically is no larger than tens.

**Robustness**    In addition to the three Byzantine attacks introduced in Section 4.2, we consider a stronger data poisoning attack in the following experiments.
- **Data poisoning attacks**: The training samples on malicious clients are poisoned with uniformly randomly chosen noisy labels. Furthermore, when communicating, these clients would scale their transmitted messages to dominate the aggregate update.

For the former three Byzantine attacks, the noise variance $\tau$ is set as 100, 10 and 100 respectively. The corruption levels, i.e., the fractions of malicious clients, are set as $\{0.1, 0.2, 0.5, 0.8\}$. For data poisoning attack, the scaling factor is randomly sampled from $\mathcal{N}(0, 20^2)$ and the corruption levels are from $\{0.02, 0.05, 0.1, 0.2\}$. Under different types of attacks and different levels of corruption, we compare the average test accuracy performance on benign clients of `lp-proj-1` and `lp-proj-2` with various defense baselines, including global training augmented with different robust aggregation techniques, such as median and Krum [7], `Ditto` and `RSA`.

Due to space limitations, we show only a representative figure for each attack in Figure 4. For full results, please refer to Appendix C.5. From Figure 4, we find that under relatively weak attacks, e.g., same-value and Gaussian attacks, the test accuracy of `lp-proj-1` and `lp-proj-2` rarely decays as the fraction of malicious clients increases, while we observe significant drops on the test accuracy for other algorithms once malicious clients exist. On the other hand, under strong attacks, e.g. sign-flipping and data poisoning, an increasing fraction of malicious clients deteriorates the accuracy performance continuously and even collapses the local model if the attack intensity is too large. For example, under the sign-flipping attack, when the fraction of malicious clients exceeds 20%, only `lp-proj-1`, `RSA` and `Global+Krum` work, while all other methods fail to produce a solution. When the attack intensity further increases to 80%, the only robust methods that achieve the desired accuracy are `lp-proj-1` and `RSA`.

The numerical results show that our method is resistant to standard malicious attacks, which roots in the combination of projection and $L^1$-norm subspace regularization that attribute to the robustness.

Consider an extreme example: if the subspace dimension is chosen as 0, then the joint optimization is reduced to pure local training. No matter how serious the adversarial attack is, the local test performance would not be affected. Therefore, random projection helps alleviate the attacks applied in the original space, while the $L^1$-norm helps eliminate outliers further [34].

**Fairness**  To illustrate the accuracy and performance fairness trade-off, we plot the variances of accuracies across the system against the corresponding test accuracies for `lp-proj` and several other different approaches in Figure 2. To examine the performance fairness in isolation, the numerical experiments are performed without adversarial attacks in this part. Furthermore, for each competing method, we select the optimal achievable test accuracy after the 20th communication round, and the corresponding variance is picked up. Due to space limitations, we show the results on two datasets here with the full results left in Figure 6 in Appendix C.6.

The results with respect to performance fairness show that `lp-proj-1` and `lp-proj-2` provide accurate and fair solutions that are comparable to other SOTA methods. In particular, on `CIFAR`, `lp-proj-1` achieves the highest test accuracy of 79.22% with the lowest variance of 0.0097 among all the competitors. Although `RSA` achieves the same variance as `lp-proj-1`, its corresponding test accuracy is only 77.68%, which is 1.54% lower than `lp-proj-1`. On the other hand, on `EMNIST`, despite the optimal approach is `Ditto`, with a test accuracy of 90.89% and the corresponding variance of 0.0016, our proposed method shows comparable performance, e.g., `lp-proj-2` achieves a test accuracy of 90.70% with a variance of 0.0016, which is only slightly inferior to the previous method. Theoretical analysis in Proposition 2 implies that under the linear model, the dependence of the variance on the projection dimension is of squared order, indicating low-dimensional projection helps reduce the variance of test losses among clients. Numerical results suggest that this conclusion may be generalized to broader settings.

## 6   Concluding Remarks

In this paper, we have proposed a simple but powerful personalized FL approach based on infimal convolution and subspace projection that we call `lp-proj`. We have presented the convergence results for smooth objectives with square regularizers. Inherent benefits of robustness and fairness of our method are also illustrated on a class of linear problems. Empirical results show that our approach could significantly save communication costs, improve robustness under various kinds of adversarial attacks, and promote performance fairness. In future work, we would be interested in establishing convergence results for general $L^p$ regularizers and strongly convex objectives, exploring numerical applications on other ML tasks and large-scale datasets, and considering additional constraints, e.g., differential privacy.

## Acknowledgments and Disclosure of Funding

This work has been supported by the National Natural Science Foundation of China (No. 12271011).

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
