# A  Convergence of `lp-proj`

In this section, we first provide some results useful for our analysis, then give several important properties of our method and present the proof of Theorem 1, i.e., the convergence of `lp-proj` for the case $p = 2$.

The framework is adapted from Dinh et al. [13], with some concrete results specific to our settings.

## A.1  Some Useful Results

In this subsection, we provide some existing results useful for our later analysis. We first introduce more definitions.

**Definition 2** (Further definitions). *Suppose that $f_k$ is a function from $\mathbb{R}^d$ to $\mathbb{R}$.*

(a) *$f_k$ is said to be convex, if for any $\mathbf{w}, \mathbf{w}' \in \mathbb{R}^d$ and $0 \le \alpha \le 1$, it holds that*

$$f_k(\alpha\mathbf{w} + (1 - \alpha)\mathbf{w}') \le \alpha f_k(\mathbf{w}) + (1 - \alpha)f_k(\mathbf{w}').$$

*If $f_k$ is differentiable, the above condition is equivalent to that for any $\mathbf{w}, \mathbf{w}' \in \mathbb{R}^d$,*

$$f_k(\mathbf{w}') \ge f_k(\mathbf{w}) + \langle \nabla f_k(\mathbf{w}), \mathbf{w}' - \mathbf{w} \rangle.$$

(b) *$f_k$ is said to be $\mu$-strongly convex for some $\mu > 0$, if for any $\mathbf{w}, \mathbf{w}' \in \mathbb{R}^d$ and $0 \le \alpha \le 1$, it holds that*

$$f_k(\alpha\mathbf{w} + (1 - \alpha)\mathbf{w}') \le \alpha f_k(\mathbf{w}) + (1 - \alpha)f_k(\mathbf{w}') - \frac{\mu\alpha(1 - \alpha)}{2} \|\mathbf{w} - \mathbf{w}'\|_2^2.$$

*If $f_k$ is differentiable, the above condition is equivalent to that for any $\mathbf{w}, \mathbf{w}' \in \mathbb{R}^d$,*

$$f_k(\mathbf{w}') \ge f_k(\mathbf{w}) + \langle \nabla f_k(\mathbf{w}), \mathbf{w}' - \mathbf{w} \rangle + \frac{\mu}{2} \|\mathbf{w}' - \mathbf{w}\|^2.$$

*If $f_k$ is twice differentiable, the above condition is also equivalent to $\nabla^2 f_k \succeq \mu \, \boldsymbol{I}_d$.*

Then we have the following property of strongly convex functions.

**Proposition 3** (Nesterov [54], Theorem 2.1.10). *If $F_k$ is $\mu_F$-strongly convex, then we have that*

$$\|\nabla F_k(\mathbf{w}') - \nabla F_k(\mathbf{w})\|_2 \ge \mu_F \|\mathbf{w}' - \mathbf{w}\|_2$$

*for any $\mathbf{w}, \mathbf{w}' \in \mathbb{R}^d$.*

Proposition 4 provides two useful inequalities, which can be derived from Cauchy-Schwarz Inequality.

**Proposition 4** (Cauchy-Schwarz inequality). *For any $\mathbf{x}_k \in \mathbb{R}^d$, $k = 1, 2, \ldots, M$, we have*

$$\left\| \sum_{k=1}^{M} \mathbf{x}_k \right\|_2^2 \le M \sum_{k=1}^{M} \|\mathbf{x}_k\|_2^2.$$

$$\|\mathbf{x}_1 + \mathbf{x}_2\|_2^2 \le (1 + c) \|\mathbf{x}_1\|_2^2 + (1 + 1/c) \|\mathbf{x}_2\|_2^2.$$

Next, we present the relationships between a function and its conjugate function.

**Proposition 5** (Hiriart-Urruty and Lemaréchal [24], Corollaries X.1.3.6 and X.1.4.4 Theorems X.4.2.1 and X.4.2.2). *Suppose that $f$ is a convex function from $\mathbb{R}^d$ to $\mathbb{R} \cup \{+\infty\}$. Define the conjugate of $f$ as $f^*(\mathbf{u}) = \sup_{\mathbf{x} \in \mathbb{R}^d}\{\langle \mathbf{u}, \mathbf{x} \rangle - f(\mathbf{x})\}$ and the biconjugate of $f$ as $f^{**} = (f^*)^*$. The domain of $f$ is denoted by $\mathrm{dom}\, f = \{\mathbf{x} \in \mathbb{R}^d : f(\mathbf{x}) \in \mathbb{R}\}$. Suppose $c$ is a positive number. Then we have the following results.*

(a) *If $f$ is convex, then $f^{**} = f$.*

(b) *If $f$ is $c$-strongly convex, then $\mathrm{dom} f^* = \mathbb{R}^d$ and $f^*$ is $1/c$-smooth.*

(c) *If $f$ is convex and $c$-smooth, then $f^*$ is $1/c$-strongly convex on every convex subset $C \subset \mathrm{dom}\, \partial f^*$.*

*(d) If $f$ is convex, then $\mathbf{u} \in \partial f(\mathbf{x}) \iff \mathbf{x} \in \partial f^*(\mathbf{u})$.*

Proposition 6 guarantees the approximate isometry of a "flat" matrix with independent rows under certain conditions.

**Proposition 6** (Vershynin [61], Theorem 5.58). *Let $\boldsymbol{A}$ be an $d \times D$ matrix $(d \leq D)$ whose rows $\mathbf{a}_i^\top$ are independent sub-gaussian isotropic random vectors in $\mathbb{R}^d$ with $\|a_j\|_2 = \sqrt{D}$. Then for every $t \geq 0$, the inequality*

$$\sqrt{D} - C\sqrt{d} - t \leq s_{\min}(\boldsymbol{A}) \leq s_{\max}(\boldsymbol{A}) \leq \sqrt{D} + C\sqrt{d} + t$$

*holds with probability at least $1 - 2\exp(-ct^2)$, where $s_{\min}(\boldsymbol{A})$ and $s_{\max}(\boldsymbol{A})$ denote the smallest and the largest singular values of $\boldsymbol{A}$, $C = C_K'$, $c = c_K' > 0$ depend only on the subgaussian norm $K = \max_j \|A_j\|_{\psi_2}$ of the rows.*

For the definitions of sub-gaussian random vectors and the norm $\|\cdot\|_{\psi_2}$, see Definition 5.7 and 5.22 in Vershynin [61]. A random vector is said to be isotropic, if its covariance matrix is the identity matrix.

With Proposition 6, we can prove that our projection matrix $\boldsymbol{P}$ is approximately orthogonal in the sense that all the singular values of $\boldsymbol{P}$ are around 1.

**Proposition 7.** *With probability at least $1 - 2\exp(-cd_{\mathrm{sub}})$, we have $1 - C\sqrt{d_{\mathrm{sub}}/d} \leq s_{\min}(\boldsymbol{P}) \leq s_{\max}(\boldsymbol{P}) \leq 1 + C\sqrt{d_{\mathrm{sub}}/d}$ for some $C, c > 0$, where $s_{\min}(\boldsymbol{P})$ and $s_{\max}(\boldsymbol{P})$ denote the smallest and the largest singular values of $\boldsymbol{P}$.*

*Proof.* For our choice of $\boldsymbol{P}$, we have $\boldsymbol{P} = (\mathbf{a}_1, \mathbf{a}_2, \ldots, \mathbf{a}_{d_{\mathrm{sub}}})^\top$ where the row vectors $\mathbf{a}_i$ are independent and uniformly distributed on the unit sphere of $\mathbb{R}^d$. Example 5.21 in Vershynin [61] implies that each $\sqrt{d}\,\mathbf{a}_i$ is isotropic. Moreover, by Example 5.25, we have that $\left\|\sqrt{d}\,\mathbf{a}_i\right\|_{\psi_2} = C_0$ for some absolute constant $C_0 > 0$.

Then by Proposition 6, we have that $1 - C\sqrt{d_{\mathrm{sub}}/d} \leq s_{\min}(\boldsymbol{P}) \leq s_{\max}(\boldsymbol{P}) \leq 1 + C\sqrt{d_{\mathrm{sub}}/d}$ with probability at least $1 - 2\exp(-cd_{\mathrm{sub}})$ for some positive constants $C$ and $c$. $\qquad\square$

For brevity, we let $s = C\sqrt{d_{\mathrm{sub}}/d}$. If $\sqrt{d_{\mathrm{sub}}/d}$ is sufficiently small, we have $s < 1$. Then Proposition 7 implies that

$$1 - s \leq s_{\min}(\boldsymbol{P}) \leq s_{\max}(\boldsymbol{P}) \leq 1 + s, \; 0 < s < 1 \tag{7}$$

holds with probability at least $1 - 2\exp(-cd_{\mathrm{sub}})$. This implies that $\mathrm{rank}(\boldsymbol{P}) = d_{\mathrm{sub}}$ and the $d_{\mathrm{sub}} \times d_{\mathrm{sub}}$ matrix $\boldsymbol{P}^\top \boldsymbol{P}$ is invertible.

The next proposition is a straightforward consequence of (7).

**Proposition 8.** *If (7) holds, then we have $\|\boldsymbol{P}\mathbf{x}\|_2^2 \leq (1+s)^2 \|\mathbf{x}\|_2^2$ for any $\mathbf{x} \in \mathbb{R}^d$, $\|\boldsymbol{P}\mathbf{x}\|_2^2 \geq (1-s)^2 \|\mathbf{x}\|_2^2$ for any $\mathbf{x} \in \mathrm{col}(\boldsymbol{P}^\top)$ and $(1-s)^2 \|\mathbf{y}\|_2^2 \leq \left\|\boldsymbol{P}^\top \mathbf{y}\right\|_2^2 \leq (1+s)^2 \|\mathbf{y}\|_2^2$ for any $\mathbf{y} \in \mathbb{R}^{d_{\mathrm{sub}}}$. Moreover, if $f(\cdot)$ is an $L$-smooth function from $\mathbb{R}^d$ to $\mathbb{R}$, then $f(\boldsymbol{P}^\top \cdot)$ is a $(1+s)^2 L$-smooth function from $\mathbb{R}^{d_{\mathrm{sub}}}$ to $\mathbb{R}$.*

*Proof.* From (7), it is easy to verify these properties except for the inequality $\|\boldsymbol{P}\mathbf{x}\|_2^2 \geq (1-s)^2 \|\mathbf{x}\|_2^2$ for any $\mathbf{x} \in \mathrm{col}(\boldsymbol{P}^\top)$. Suppose the SVD of $\boldsymbol{P}$ is $\boldsymbol{P} = \boldsymbol{U}\boldsymbol{D}\boldsymbol{V}^\top$ where $\boldsymbol{U}$ is a $d_{\mathrm{sub}} \times d_{\mathrm{sub}}$ orthogonal matrix, $\boldsymbol{D}$ is a $d_{\mathrm{sub}} \times d_{\mathrm{sub}}$ diagonal matrix whose digonal elements are between $1 - s$ and $1 + s$, and $\boldsymbol{V}$ is a $d \times d_{\mathrm{sub}}$ matrix with orthogonal column vectors. For $\mathbf{x} = \boldsymbol{P}^\top \mathbf{y}$, we have

$$\|\boldsymbol{P}\mathbf{x}\|_2^2 = \left\|\boldsymbol{P}\boldsymbol{P}^\top \mathbf{y}\right\|_2^2 = \mathbf{y}^\top \boldsymbol{P}\boldsymbol{P}^\top \boldsymbol{P}\boldsymbol{P}^\top \mathbf{y} = \mathbf{y}^\top \boldsymbol{P}\boldsymbol{V}\boldsymbol{D}^2 \boldsymbol{V}^\top \boldsymbol{P}^\top \mathbf{y}.$$

If $\mathbf{y} \neq \mathbf{0}_{d_{\mathrm{sub}}}$, $\boldsymbol{V}^\top \boldsymbol{P}^\top \mathbf{y} = \boldsymbol{D}\boldsymbol{U}^\top \mathbf{y} \neq \mathbf{0}_{d_{\mathrm{sub}}}$. Since $\boldsymbol{D}^2 \succeq (1-s)^2 \boldsymbol{I}_{d_{\mathrm{sub}}}$. It follows that

$$\|\boldsymbol{P}\mathbf{x}\|_2^2 \geq (1-s)^2 \mathbf{y}^\top \boldsymbol{P}\boldsymbol{V}\boldsymbol{V}^\top \boldsymbol{P}^\top \mathbf{y} = (1-s)^2 \mathbf{y}^\top \boldsymbol{U}\boldsymbol{D}\boldsymbol{D}\boldsymbol{U}^\top \mathbf{y}$$

$$= (1-s)^2 \mathbf{y}^\top \boldsymbol{U}\boldsymbol{D}\boldsymbol{V}^\top \boldsymbol{V}\boldsymbol{D}\boldsymbol{U}^\top \mathbf{y} = (1-s)^2 \left\|\boldsymbol{P}^\top \mathbf{y}\right\|_2^2 = (1-s)^2 \|\mathbf{x}\|_2^2.$$

This completes the proof. $\qquad\square$

## A.2 Important Properties

In this subsection, we give several important properties of $F_k$ and $\mathbf{x}_{k,r}^t$. First, we can establish the smoothness of $F_k$ as follows.

**Proposition 9.** *Suppose Assumption 1, 4 and (7) hold with $0 < s < 1/30$ and $\lambda > 4L$. Then $F_k$ is $L_F$-smooth with $L_F = \lambda$. Moreover, $\nabla F_k(\tilde{\mathbf{w}}) = \lambda(\tilde{\mathbf{w}} - \boldsymbol{P}\boldsymbol{P}^\top \hat{\mathbf{y}}_k)$, where $\hat{\mathbf{y}}_k = \operatorname{argmin}_{\mathbf{y}_k \in \mathbb{R}^{d_{\text{sub}}}} \left\{ f_k(\boldsymbol{P}^\top \mathbf{y}_k) + \frac{\lambda}{2} \left\| \tilde{\mathbf{w}} - \boldsymbol{P}\boldsymbol{P}^\top \mathbf{y}_k \right\|_2^2 \right\}$.*

For the square regularizer, the local update iteration becomes $\tilde{\mathbf{w}}_{k,r+1}^t = \tilde{\mathbf{w}}_{k,r}^t - \eta\lambda(\tilde{\mathbf{w}}_{k,r}^t - \boldsymbol{P}\mathbf{x}_{k,r}^t)$. Note that Proposition 9 implies that $\nabla F_k(\tilde{\mathbf{w}}_{k,r}^t) = \lambda(\tilde{\mathbf{w}}_{k,r}^t - \boldsymbol{P}\boldsymbol{P}^\top \hat{\mathbf{y}}_{k,r}^t)$ where $\hat{\mathbf{y}}_{k,r}^t = \operatorname{argmin}_{\mathbf{y}_k \in \mathbb{R}^{d_{\text{sub}}}} \left\{ f_k(\boldsymbol{P}^\top \mathbf{y}_k) + \frac{\lambda}{2} \left\| \tilde{\mathbf{w}}_{k,r}^t - \boldsymbol{P}\boldsymbol{P}^\top \mathbf{y}_k \right\|_2^2 \right\}$. Then our local update can be viewed as an approximation of gradient descent on $F_k$.

Then we restate Lemma 1 as follows.

**Lemma 2.** *Suppose that Assumptions 1, 2, 4, and (7) hold with $0 < s < 1/30$ and $\lambda > 4L$. For a fixed $\tilde{\mathbf{w}}_{k,r}^t$, we have*

$$\frac{1}{\lambda^2}\mathbb{E}\left[\left\|\nabla F_k(\tilde{\mathbf{w}}_{k,r}^t) - \lambda(\tilde{\mathbf{w}}_{k,r}^t - \boldsymbol{P}\mathbf{x}_{k,r}^t)\right\|_2^2\right] \leq \delta^2 := \frac{2(1+s)^6}{((1-s)^4\lambda - (1+s)^2L)^2}\left(\frac{\gamma_f^2}{|\tilde{\mathcal{D}}_k|} + \nu\right). \quad (8)$$

The next lemma provides the bounded diversity of $F_k$.

**Lemma 3.** *If Assumption 1, 3, 4 and (7) holds with $0 < s < 1/30$, and $\lambda > \sqrt{10}L$, then we have*

$$\sum_{k=1}^N \|\nabla F_k(\tilde{\mathbf{w}}) - \nabla F(\tilde{\mathbf{w}})\|_2^2 \leq \frac{10L^2}{\lambda^2 - 10L^2}\|\nabla F(\tilde{\mathbf{w}})\|_2^2 + 3\underbrace{\frac{\lambda^2}{\lambda^2 - 10L^2}\sigma_f^2}_{=:\sigma_F^2}.$$

## A.3 Proof of Proposition 9

This proof is adapted from Hoheisel et al. [25].

Let $\varphi_\lambda(\mathbf{y}) = f_k(\boldsymbol{P}^\top\mathbf{y}) + \frac{\lambda}{2}\left\|\boldsymbol{P}\boldsymbol{P}^\top\mathbf{y}\right\|_2^2$. By (7), we have that the smallest eigenvalue of $\boldsymbol{P}\boldsymbol{P}^\top\boldsymbol{P}\boldsymbol{P}^\top$ is no less than $(1-s)^4$. Since $\lambda > 4L$ and $0 < s < 1/30$, $\varphi_\lambda(\mathbf{y})$ is $((1-s)^4\lambda - (1+s)^2L)$-strongly convex. Similarly, the function $f_k(\boldsymbol{P}^\top\mathbf{y}) + \frac{\lambda}{2}\left\|\tilde{\mathbf{w}} - \boldsymbol{P}\boldsymbol{P}^\top\mathbf{y}\right\|_2^2$ is also $((1-s)^4\lambda - (1+s)^2L)$-strongly convex. Such an $\hat{\mathbf{y}}_k$ exists and is unique. By Proposition 5, $\varphi_\lambda^*$ is a continuously differentiable function defined on $\mathbb{R}^{d_{\text{sub}}}$ and $\nabla\varphi_\lambda^* = (\nabla\varphi_\lambda)^{-1}$.

Then we have

$$\begin{aligned}
F_k(\mathbf{w}) &= \min_{\mathbf{x}_k \in \mathbb{R}^d}\left\{f_k(\mathbf{x}_k) + \frac{\lambda}{2}\|\tilde{\mathbf{w}} - \boldsymbol{P}\mathbf{x}_k\|_2^2\right\} \\
&= \min_{\mathbf{y} \in \mathbb{R}^{d_{\text{sub}}}}\left\{f_k(\boldsymbol{P}^\top\mathbf{y}) + \frac{\lambda}{2}\left\|\tilde{\mathbf{w}} - \boldsymbol{P}\boldsymbol{P}^\top\mathbf{y}\right\|_2^2\right\} \\
&= \frac{\lambda}{2}\|\tilde{\mathbf{w}}\|_2^2 - \sup_{\mathbf{y}\in\mathbb{R}^{d_{\text{sub}}}}\left\{\lambda\langle\tilde{\mathbf{w}}, \boldsymbol{P}\boldsymbol{P}^\top\mathbf{y}\rangle - f_k(\boldsymbol{P}^\top\mathbf{y}) - \frac{\lambda}{2}\left\|\boldsymbol{P}\boldsymbol{P}^\top\mathbf{y}\right\|_2^2\right\} \\
&= \frac{\lambda}{2}\|\tilde{\mathbf{w}}\|_2^2 - \varphi_\lambda^*(\lambda\boldsymbol{P}\boldsymbol{P}^\top\tilde{\mathbf{w}}),
\end{aligned}$$

where the second equality is by Assumption 4. Then $\nabla F_k(\mathbf{w}) = \lambda\tilde{\mathbf{w}} - \lambda\boldsymbol{P}\boldsymbol{P}^\top\nabla\varphi_\lambda^*(\lambda\boldsymbol{P}\boldsymbol{P}^\top\tilde{\mathbf{w}})$. On the other hand, we have

$$\begin{aligned}
\hat{\mathbf{y}}_k &= \operatorname{argmin}_{\mathbf{y}\in\mathbb{R}^{d_{\text{sub}}}}\left\{f_k(\boldsymbol{P}^\top\mathbf{y}) + \frac{\lambda}{2}\left\|\tilde{\mathbf{w}} - \boldsymbol{P}\boldsymbol{P}^\top\mathbf{y}\right\|_2^2\right\} \\
&= \operatorname{argmin}_{\mathbf{y}\in\mathbb{R}^{d_{\text{sub}}}}\left\{\varphi_\lambda(\mathbf{y}) - \lambda\langle\tilde{\mathbf{w}}, \boldsymbol{P}\boldsymbol{P}^\top\mathbf{y}\rangle\right\}.
\end{aligned}$$

The first-order condition implies $\nabla\varphi_\lambda(\hat{\mathbf{y}}_k) = \lambda \boldsymbol{P}\boldsymbol{P}^\top \tilde{\mathbf{w}}$. It follows that $\hat{\mathbf{y}}_k = \nabla\varphi_\lambda^*(\lambda\boldsymbol{P}\boldsymbol{P}^\top\tilde{\mathbf{w}})$. Finally, we obtain $\nabla F_k(\tilde{\mathbf{w}}) = \lambda\tilde{\mathbf{w}} - \lambda\boldsymbol{P}\boldsymbol{P}^\top\hat{\mathbf{y}}_k$.

Now we prove the Lipschitz continuity of $\nabla F_k$. Let $\psi_\lambda(\mathbf{x}) = f_k(\mathbf{x}) + \frac{\lambda}{2}\|\boldsymbol{P}\mathbf{x}\|_2^2$ and $\hat{\mathbf{x}}_k = \boldsymbol{P}^\top\hat{\mathbf{y}}_k$. By Assumption 4, we have

$$\hat{\mathbf{x}}_k \in \underset{\mathbf{x}\in\mathbb{R}^d}{\operatorname{argmin}}\left\{f_k(\mathbf{x}) + \frac{\lambda}{2}\|\tilde{\mathbf{w}} - \boldsymbol{P}\mathbf{x}\|_2^2\right\} = \underset{\mathbf{x}\in\mathbb{R}^d}{\operatorname{argmin}}\left\{\psi_\lambda(\mathbf{x}) - \lambda\langle\tilde{\mathbf{w}}, \boldsymbol{P}\mathbf{x}\rangle\right\}.$$

The first-order condition implies $\nabla\psi_\lambda(\hat{\mathbf{x}}_k) = \lambda\boldsymbol{P}^\top\tilde{\mathbf{w}}$. By Proposition 8, $\psi_\lambda$ is $\left((1-s)^2\lambda - L\right)$-strongly convex on $\operatorname{col}(\boldsymbol{P}^\top)$. Then we have that for any $\mathbf{x}\in\operatorname{col}(\boldsymbol{P}^\top)$, it holds that

$$\psi_\lambda(\hat{\mathbf{x}}_k) \le \psi_\lambda(\mathbf{x}) + \lambda\langle\boldsymbol{P}^\top\tilde{\mathbf{w}}, \hat{\mathbf{x}}_k - \mathbf{x}\rangle - \frac{1}{2}\left((1-s)^2\lambda - L\right)\|\mathbf{x} - \hat{\mathbf{x}}_k\|_2^2.$$

Recalling the definition of $\psi_\lambda$, we obtain

$$f_k(\hat{\mathbf{x}}_k) + \frac{\lambda}{2}\|\boldsymbol{P}\hat{\mathbf{x}}_k\|_2^2 - \frac{\lambda}{2}\|\boldsymbol{P}\mathbf{x}\|_2^2 - \lambda\langle\boldsymbol{P}^\top\tilde{\mathbf{w}}, \hat{\mathbf{x}}_k - \mathbf{x}\rangle + \frac{\lambda}{2}\|\boldsymbol{P}\hat{\mathbf{x}}_k - \boldsymbol{P}\mathbf{x}\|_2^2$$

$$\le f_k(\mathbf{x}) + \left(\frac{L}{2} - \frac{(1-s)^2\lambda}{2}\right)\|\mathbf{x} - \hat{\mathbf{x}}_k\|_2^2 + \frac{\lambda}{2}\|\boldsymbol{P}\hat{\mathbf{x}}_k - \boldsymbol{P}\mathbf{x}\|_2^2.$$

By Proposition 8, we have $\|\boldsymbol{P}\hat{\mathbf{x}}_k - \boldsymbol{P}\mathbf{x}\|_2^2 \le (1+s)^2\|\hat{\mathbf{x}}_k - \mathbf{x}\|_2^2$. It follows that

$$f_k(\hat{\mathbf{x}}_k) + \frac{\lambda}{2}\|\boldsymbol{P}\hat{\mathbf{x}}_k\|_2^2 - \frac{\lambda}{2}\|\boldsymbol{P}\mathbf{x}\|_2^2 - \lambda\langle\boldsymbol{P}^\top\tilde{\mathbf{w}}, \hat{\mathbf{x}}_k - \mathbf{x}\rangle + \frac{\lambda}{2}\|\boldsymbol{P}\hat{\mathbf{x}}_k - \boldsymbol{P}\mathbf{x}\|_2^2$$

$$\le f_k(\mathbf{x}) + \left(\frac{L}{2} + 2s\lambda\right)\|\mathbf{x} - \hat{\mathbf{x}}_k\|_2^2,$$

which is equivalent to

$$f_k(\hat{\mathbf{x}}_k) + \lambda\langle\boldsymbol{P}^\top\boldsymbol{P}\hat{\mathbf{x}}_k - \boldsymbol{P}^\top\tilde{\mathbf{w}}, \hat{\mathbf{x}}_k - \mathbf{x}\rangle \le f_k(\mathbf{x}) + \left(\frac{L}{2} + 2s\lambda\right)\|\mathbf{x} - \hat{\mathbf{x}}_k\|_2^2. \tag{9}$$

For a $\tilde{\mathbf{w}}' \ne \tilde{\mathbf{w}}$, let $\hat{\mathbf{y}}_k' = \operatorname{argmin}_{\mathbf{y}\in\mathbb{R}^{d_{\mathrm{sub}}}}\left\{f_k(\boldsymbol{P}^\top\mathbf{y}) + \frac{\lambda}{2}\|\tilde{\mathbf{w}}' - \boldsymbol{P}\boldsymbol{P}^\top\mathbf{y}\|_2^2\right\}$ and $\hat{\mathbf{x}}_k' = \boldsymbol{P}^\top\hat{\mathbf{y}}_k'$. Then we also have $\hat{\mathbf{x}}_k' \in \operatorname{col}(\boldsymbol{P}^\top)$. Replacing $\mathbf{x}$ by $\hat{\mathbf{x}}_k'$ in (9) gives

$$f_k(\hat{\mathbf{x}}_k) + \lambda\langle\boldsymbol{P}^\top\boldsymbol{P}\hat{\mathbf{x}}_k - \boldsymbol{P}^\top\tilde{\mathbf{w}}, \hat{\mathbf{x}}_k - \hat{\mathbf{x}}_k'\rangle \le f_k(\hat{\mathbf{x}}_k') + \left(\frac{L}{2} + 2s\lambda\right)\|\hat{\mathbf{x}}_k' - \hat{\mathbf{x}}_k\|_2^2.$$

Changing the orders of $\hat{\mathbf{x}}_k$ and $\hat{\mathbf{x}}_k'$ leads to

$$f_k(\hat{\mathbf{x}}_k') + \lambda\langle\boldsymbol{P}^\top\boldsymbol{P}\hat{\mathbf{x}}_k' - \boldsymbol{P}^\top\tilde{\mathbf{w}}', \hat{\mathbf{x}}_k' - \hat{\mathbf{x}}_k\rangle \le f_k(\hat{\mathbf{x}}_k) + \left(\frac{L}{2} + 2s\lambda\right)\|\hat{\mathbf{x}}_k' - \hat{\mathbf{x}}_k\|_2^2.$$

Adding the above two inequalities and rearranging terms yields

$$\lambda\langle\boldsymbol{P}^\top\boldsymbol{P}(\hat{\mathbf{x}}_k - \hat{\mathbf{x}}_k'), \hat{\mathbf{x}}_k - \hat{\mathbf{x}}_k'\rangle - (L + 4s\lambda)\|\hat{\mathbf{x}}_k' - \hat{\mathbf{x}}_k\|_2^2 \le \lambda\langle\boldsymbol{P}^\top(\tilde{\mathbf{w}} - \tilde{\mathbf{w}}'), \hat{\mathbf{x}}_k - \hat{\mathbf{x}}_k'\rangle.$$

By Proposition 8, $\langle\boldsymbol{P}^\top\boldsymbol{P}(\hat{\mathbf{x}}_k - \hat{\mathbf{x}}_k'), \hat{\mathbf{x}}_k - \hat{\mathbf{x}}_k'\rangle = \|\boldsymbol{P}(\hat{\mathbf{x}}_k - \hat{\mathbf{x}}_k')\|_2^2 \ge (1-s)^2\|\hat{\mathbf{x}}_k - \hat{\mathbf{x}}_k'\|_2^2$. Then we have

$$\left((1-6s-s^2)\lambda - L\right)\|\hat{\mathbf{x}}_k' - \hat{\mathbf{x}}_k\|_2^2 \le \lambda\langle\boldsymbol{P}^\top(\tilde{\mathbf{w}} - \tilde{\mathbf{w}}'), \hat{\mathbf{x}}_k - \hat{\mathbf{x}}_k'\rangle.$$

Since $s < 1/30$ and $\lambda > 4L$, we have $(1-6s-s^2)\lambda - L > 0$. Dividing both sides by $(1-6s-s^2)\lambda - L - L$ gives

$$\|\hat{\mathbf{x}}_k' - \hat{\mathbf{x}}_k\|_2^2 \le \frac{1}{1-6s-s^2-L/\lambda}\langle\boldsymbol{P}^\top(\tilde{\mathbf{w}} - \tilde{\mathbf{w}}'), \hat{\mathbf{x}}_k - \hat{\mathbf{x}}_k'\rangle. \tag{10}$$

Then we have

$$\frac{1}{\lambda^2}\|\nabla F_k(\tilde{\mathbf{w}}) - \nabla F_k(\tilde{\mathbf{w}}')\|_2^2 = \|\tilde{\mathbf{w}} - \tilde{\mathbf{w}}' - \boldsymbol{P}(\hat{\mathbf{x}}_k - \hat{\mathbf{x}}_k')\|_2^2$$

$$= \|\tilde{\mathbf{w}} - \tilde{\mathbf{w}}'\|_2^2 - 2\langle\tilde{\mathbf{w}} - \tilde{\mathbf{w}}', \boldsymbol{P}(\hat{\mathbf{x}}_k - \hat{\mathbf{x}}_k')\rangle + \|\boldsymbol{P}(\hat{\mathbf{x}}_k - \hat{\mathbf{x}}_k')\|_2^2$$

$$\le \|\tilde{\mathbf{w}} - \tilde{\mathbf{w}}'\|_2^2 - 2\langle\tilde{\mathbf{w}} - \tilde{\mathbf{w}}', \boldsymbol{P}(\hat{\mathbf{x}}_k - \hat{\mathbf{x}}_k')\rangle + (1+s)^2\|\hat{\mathbf{x}}_k - \hat{\mathbf{x}}_k'\|_2^2$$

$$\le \|\tilde{\mathbf{w}} - \tilde{\mathbf{w}}'\|_2^2 + \left(\frac{(1+s)^2}{1-6s-s^2-L/\lambda} - 2\right)\langle\tilde{\mathbf{w}} - \tilde{\mathbf{w}}', \boldsymbol{P}(\hat{\mathbf{x}}_k - \hat{\mathbf{x}}_k')\rangle,$$

where the first inequality is due to Proposition 8 and the second one is due to (10). Since $\lambda > 4L$ and $s < 1/30$, we have $\frac{(1+s)^2}{1-6s-s^2-L/\lambda} - 2 < 0$. As a result, $\|\nabla F_k(\tilde{\mathbf{w}}) - \nabla F_k(\tilde{\mathbf{w}}')\|_2^2 \le \lambda^2\|\tilde{\mathbf{w}} - \tilde{\mathbf{w}}'\|_2^2$.

## A.4  Proof of Lemma 2

Let $\mathbf{x}_{k,r}^t = \boldsymbol{P}^\top \mathbf{y}_{k,r}^t + \boldsymbol{Q}\tilde{\mathbf{y}}_{k,r}^t$. Recall that we have $\boldsymbol{PQ} = \mathbf{0}_{d_{\mathrm{sub}} \times (d - d_{\mathrm{sub}})}$. Then by Proposition 9 and (7), we have $\left\| \nabla F_k(\tilde{\mathbf{w}}_{k,r}^t) - \lambda(\tilde{\mathbf{w}}_{k,r}^t - \boldsymbol{P}\mathbf{x}_{k,r}^t) \right\|_2 = \lambda \left\| \boldsymbol{PP}^\top (\hat{\mathbf{y}}_{k,r}^t - \mathbf{y}_{k,r}^t) \right\|_2 \leq (1+s)^2 \lambda \left\| \hat{\mathbf{y}}_{k,r}^t - \mathbf{y}_{k,r}^t \right\|_2$ where $\hat{\mathbf{y}}_{k,r}^t = \mathrm{argmin}_{\mathbf{y}_k \in \mathbb{R}^{d_{\mathrm{sub}}}} \left\{ f_k(\boldsymbol{P}^\top \mathbf{y}_k) + \frac{\lambda}{2} \left\| \tilde{\mathbf{w}}_{k,r}^t - \boldsymbol{PP}^\top \mathbf{y}_k \right\|_2^2 \right\}$. Then we focus on the distance between $\hat{\mathbf{y}}_{k,r}^t$ and $\mathbf{y}_{k,r}^t$.

Recall the definition of $\tilde{h}_k$ in Eqn. 4). Throughout this proof, $\tilde{\mathbf{w}}_{k,r}^t$ and $\tilde{\mathcal{D}}_k$ are fixed, so we omit the dependence of $\tilde{h}_k$ on these parameters for brevity. For any $\mathbf{x}_k = (\boldsymbol{P}^\top, \boldsymbol{Q}) \begin{pmatrix} \mathbf{y}_k \\ \tilde{\mathbf{y}}_k \end{pmatrix}$, we have $\begin{pmatrix} \partial_{\mathbf{y}_k} \tilde{h}_k \\ \partial_{\tilde{\mathbf{y}}_k} \tilde{h}_k \end{pmatrix} = \begin{pmatrix} \boldsymbol{P} \\ \boldsymbol{Q}^\top \end{pmatrix} \nabla_{\mathbf{x}_k} \tilde{h}_k$. By Assumption 4, we have $\partial_{\tilde{\mathbf{y}}_k} \tilde{h}_k = \mathbf{0}$. Then with some abuse of notation, we can view $\tilde{h}_k$ as a function of $\mathbf{y}_k$:

$$\tilde{h}_k(\mathbf{y}_k) = \frac{1}{|\tilde{\mathcal{D}}_k|} \sum_{\xi_{k,i} \in \tilde{\mathcal{D}}_k} \tilde{f}_k(\boldsymbol{P}^\top \mathbf{y}_k; \xi_{k,i}) + \frac{\lambda}{2} \left\| \tilde{\mathbf{w}}_{k,r}^t - \boldsymbol{PP}^\top \mathbf{y}_k \right\|_2^2,$$

and it holds that $\nabla_{\mathbf{y}_k} \tilde{h}_k = \boldsymbol{P} \nabla_{\mathbf{x}_k} \tilde{h}_k$.

For convenience, let $h_k(\mathbf{x}_k) = f_k(\mathbf{x}_k) + \frac{\lambda}{2} \left\| \tilde{\mathbf{w}}_{k,r}^t - \boldsymbol{P}\mathbf{x}_k \right\|_2^2$ and $\hat{\mathbf{x}}_{k,r}^t = \boldsymbol{P}^\top \hat{\mathbf{y}}_{k,r}^t$. By Proposition 8, $\tilde{h}_k$ is $\left( (1-s)^4 \lambda - (1+s)^2 L \right)$-strongly convex in $\mathbf{y}_k$ and $\nabla_{\mathbf{y}_k} h_k(\hat{\mathbf{y}}_{k,r}^t) = \mathbf{0}$. Then by (7) and Propositions 3 and 4, we have

$$\mathbb{E}_{\tilde{\mathcal{D}}_k} \left\| \hat{\mathbf{y}}_{k,r}^t - \mathbf{y}_{k,r}^t \right\|_2^2 \leq \frac{1}{((1-s)^4 \lambda - (1+s)^2 L)^2} \mathbb{E}_{\tilde{\mathcal{D}}_k} \left\| \nabla_{\mathbf{y}_k} \tilde{h}_k(\hat{\mathbf{y}}_{k,r}^t) - \nabla_{\mathbf{y}_k} \tilde{h}_k(\mathbf{y}_{k,r}^t) \right\|_2^2$$

$$\leq \frac{(1+s)^2}{((1-s)^4 \lambda - (1+s)^2 L)^2} \mathbb{E}_{\tilde{\mathcal{D}}_k} \left\| \nabla_{\mathbf{x}_k} \tilde{h}_k(\hat{\mathbf{x}}_{k,r}^t) - \nabla_{\mathbf{x}_k} \tilde{h}_k(\mathbf{x}_{k,r}^t) \right\|_2^2$$

$$\leq \frac{2(1+s)^2}{((1-s)^4 \lambda - (1+s)^2 L)^2} \left( \mathbb{E}_{\tilde{\mathcal{D}}_k} \left\| \nabla_{\mathbf{x}_k} \tilde{h}_k(\hat{\mathbf{x}}_{k,r}^t) - \nabla h_k(\hat{\mathbf{x}}_{k,r}^t) \right\|_2^2 + \mathbb{E}_{\tilde{\mathcal{D}}_k} \left\| \nabla_{\mathbf{x}_k} \tilde{h}_k(\mathbf{x}_{k,r}^t) \right\|_2^2 \right)$$

$$\leq \frac{2(1+s)^2}{((1-s)^4 \lambda - (1+s)^2 L)^2} \left( \mathbb{E}_{\tilde{\mathcal{D}}_k} \left\| \frac{1}{|\tilde{\mathcal{D}}_k|} \sum_{\xi_{k,i} \in \tilde{\mathcal{D}}_k} \nabla \tilde{f}_k(\hat{\mathbf{x}}_{k,r}^t; \xi_{k,i}) - \nabla f_k(\hat{\mathbf{x}}_{k,r}^t) \right\|_2^2 + \nu \right)$$

$$\leq \frac{2(1+s)^2}{((1-s)^4 \lambda - (1+s)^2 L)^2} \left( \frac{1}{|\tilde{\mathcal{D}}_k|^2} \sum_{\xi_{k,i} \in \tilde{\mathcal{D}}_k} \mathbb{E}_{\xi_{k,i}} \left\| \nabla \tilde{f}_k(\hat{\mathbf{x}}_{k,r}^t; \xi_{k,i}) - \nabla f_k(\hat{\mathbf{x}}_{k,r}^t) \right\|_2^2 + \nu \right)$$

$$\leq \frac{2(1+s)^2}{((1-s)^4 \lambda - (1+s)^2 L)^2} \left( \frac{\gamma_f^2}{|\tilde{\mathcal{D}}_k|} + \nu \right),$$

where the fourth inequality is due to $\xi_{k,i}$ are independent and $\mathbb{E}_{\xi_{k,i}} \nabla \tilde{f}_k(\hat{\mathbf{x}}_{k,r}^t; \xi_{k,i}) = f_k(\hat{\mathbf{x}}_{k,r}^t)$ and the last inequality is by Assumption 2. Then by Proposition 9 and (7), we have

$$\frac{1}{\lambda^2} \mathbb{E} \left[ \left\| \nabla F_k(\tilde{\mathbf{w}}_{k,r}^t) - \lambda(\tilde{\mathbf{w}}_{k,r}^t - \boldsymbol{P}\mathbf{x}_{k,r}^t) \right\|_2^2 \right] \leq \frac{2(1+s)^6}{((1-s)^4 \lambda - (1+s)^2 L)^2} \left( \frac{\gamma_f^2}{|\tilde{\mathcal{D}}_k|} + \nu \right).$$

## A.5  Proof of Lemma 3

If $f_k$ is $L$-smooth, by Proposition 9, we have

$$\|\nabla F_k(\tilde{\mathbf{w}}) - \nabla F(\tilde{\mathbf{w}})\|_2^2 = \left\| \lambda(\tilde{\mathbf{w}} - \boldsymbol{P}\hat{\mathbf{x}}_k) - \frac{1}{N} \sum_{i=1}^{N} \lambda(\tilde{\mathbf{w}} - \boldsymbol{P}\hat{\mathbf{x}}_i) \right\|_2^2,$$

where $\hat{\mathbf{x}}_k = \boldsymbol{P}^\top \hat{\mathbf{y}}_k$ with $\hat{\mathbf{y}}_k = \mathrm{argmin}_{\mathbf{y}_k \in \mathbb{R}^{d_{\mathrm{sub}}}} \left\{ f_k(\boldsymbol{P}^\top \mathbf{y}_k) + \frac{\lambda}{2} \left\| \tilde{\mathbf{w}} - \boldsymbol{PP}^\top \mathbf{y}_k \right\|_2^2 \right\}$.

The first-order condition implies $P \nabla f_k(P^\top \hat{\mathbf{y}}_k) = \lambda P P^\top (\tilde{\mathbf{w}} - P P^\top \hat{\mathbf{y}}_k)$, which implies $P \nabla f_k(\hat{\mathbf{x}}_k) = \lambda P P^\top (\tilde{\mathbf{w}} - P \hat{\mathbf{x}}_k)$. By (7), it is easy to verify $\left\| (P P^\top)^{-1} P \right\|_2 \leq (1 - s)^{-1}$ through SVD. Then we have

$$
\begin{aligned}
\| \nabla F_k(\tilde{\mathbf{w}}) - \nabla F(\tilde{\mathbf{w}}) \|_2^2 &= \left\| (P P^\top)^{-1} P \left( \nabla f_k(\hat{\mathbf{x}}_k) - \frac{1}{N} \sum_{i=1}^{N} \nabla f_i(\hat{\mathbf{x}}_i) \right) \right\|_2^2 \\
&\leq (1 - s)^{-2} \left\| \left( \nabla f_k(\hat{\mathbf{x}}_k) - \frac{1}{N} \sum_{i=1}^{N} \nabla f_i(\hat{\mathbf{x}}_i) \right) \right\|_2^2 \\
&\leq 2(1 - s)^{-2} \left\| \left( \nabla f_k(\hat{\mathbf{x}}_k) - \frac{1}{N} \sum_{i=1}^{N} \nabla f_i(\hat{\mathbf{x}}_k) \right) \right\|_2^2 \\
&\quad + 2(1 - s)^{-2} \left\| \left( \frac{1}{N} \sum_{i=1}^{N} \nabla f_i(\hat{\mathbf{x}}_k) - \frac{1}{N} \sum_{i=1}^{N} \nabla f_i(\hat{\mathbf{x}}_i) \right) \right\|_2^2,
\end{aligned}
$$

where the last inequality is by Proposition 4.

Taking the average over the devices, we obtain that

$$
\frac{1}{N} \sum_{k=1}^{N} \| \nabla F_k(\tilde{\mathbf{w}}) - \nabla F(\tilde{\mathbf{w}}) \|_2^2 \leq 2(1 - s)^{-2} \sum_{k=1}^{N} \| (\nabla f_k(\hat{\mathbf{x}}_k) - \nabla f(\hat{\mathbf{x}}_k)) \|_2^2 \tag{11}
$$

$$
+ 2(1 - s)^{-2} \sum_{k=1}^{N} \left\| \frac{1}{N} \sum_{i=1}^{N} (\nabla f_i(\hat{\mathbf{x}}_k) - \nabla f_i(\hat{\mathbf{x}}_i)) \right\|_2^2
$$

$$
\leq 2(1 - s)^{-2} \sigma_f^2 + \frac{2(1 - s)^{-2}}{N^2} \sum_{k=1}^{N} \sum_{i=1}^{N} \| \nabla f_i(\hat{\mathbf{x}}_k) - \nabla f_i(\hat{\mathbf{x}}_i) \|_2^2, \tag{12}
$$

where the last inequality is by Assumption 3 and Proposition 4. By the smoothness of $f_i$, we have

$$
\begin{aligned}
\| \nabla f_i(\hat{\mathbf{x}}_k) - \nabla f_i(\hat{\mathbf{x}}_i) \|_2^2 &\leq L^2 \| \hat{\mathbf{x}}_k - \hat{\mathbf{x}}_i \|_2^2 = L^2 \left\| P^\top (\hat{\mathbf{y}}_k - \hat{\mathbf{y}}_i) \right\|_2^2 \\
&= L^2 \left\| P^\top (P P^\top)^{-1} P P^\top (\hat{\mathbf{y}}_k - \hat{\mathbf{y}}_i) \right\|_2^2 \\
&\leq (1 - s)^{-2} L^2 \left\| P P^\top (\hat{\mathbf{y}}_k - \hat{\mathbf{y}}_i) \right\|_2^2 \\
&= (1 - s)^{-2} L^2 \| P \hat{\mathbf{x}}_k - P \hat{\mathbf{x}}_i \|_2^2 \\
&\leq 2(1 - s)^{-2} L^2 \left( \| P \hat{\mathbf{x}}_k - \tilde{\mathbf{w}} \|_2^2 + \| P \hat{\mathbf{x}}_i - \tilde{\mathbf{w}} \|_2^2 \right) \\
&= \frac{2(1 - s)^{-2} L^2}{\lambda^2} \left( \| \nabla F_k(\tilde{\mathbf{w}}) \|_2^2 + \| \nabla F_i(\tilde{\mathbf{w}}) \|_2^2 \right), \tag{13}
\end{aligned}
$$

where the third inequality is by Proposition 4 and the last equality is by Proposition 9. Substituting (13) into (12) gives

$$
\begin{aligned}
\frac{1}{N} \sum_{k=1}^{N} \| \nabla F_k(\tilde{\mathbf{w}}) - \nabla F(\tilde{\mathbf{w}}) \|_2^2 &\leq 2(1 - s)^{-2} \sigma_f^2 + \frac{8(1 - s)^{-4} L^2}{\lambda^2} \frac{1}{N} \sum_{k=1}^{N} \| \nabla F_k(\tilde{\mathbf{w}}) \|_2^2 \\
&\leq 3\sigma_f^2 + \frac{10 L^2}{\lambda^2} \frac{1}{N} \sum_{k=1}^{N} \| \nabla F_k(\tilde{\mathbf{w}}) \|_2^2 \\
&= 3\sigma_f^2 + \frac{10 L^2}{\lambda^2} \left( \frac{1}{N} \sum_{k=1}^{N} \| \nabla F_k(\tilde{\mathbf{w}}) - \nabla F(\tilde{\mathbf{w}}) \|_2^2 + \| \nabla F(\tilde{\mathbf{w}}) \|_2^2 \right),
\end{aligned}
$$

where the second inequality follows from $s < 1/30$ and the last equality is due to the fact that $\mathbb{E}[\| X \|_2^2] = \mathbb{E}[\| X - \mathbb{E}[X] \|_2^2] + \| \mathbb{E}[X] \|_2^2$ for a random vector $X$. Finally, rearrange the terms yields

$$
\frac{1}{N} \sum_{k=1}^{N} \| \nabla F_k(\tilde{\mathbf{w}}) - \nabla F(\tilde{\mathbf{w}}) \|_2^2 \leq \frac{3\lambda^2}{\lambda^2 - 10 L^2} \sigma_f^2 + \frac{10 L^2}{\lambda^2 - 10 L^2} \| \nabla F(\tilde{\mathbf{w}}) \|_2^2.
$$

## A.6 Proof of Theorem 1

In this subsection, we give the proof of Theorem 1.

We rewrite the local update as

$$\tilde{\mathbf{w}}_{k.r+1}^t = \tilde{\mathbf{w}}_{k,r}^t - \eta \underbrace{\lambda(\tilde{\mathbf{w}}_{k,r}^t - \boldsymbol{P}\mathbf{x}_{k,r}^t)}_{=:\mathbf{g}_{k,r}^t}, \tag{14}$$

which implies $\eta \sum_{r=0}^{R-1} \mathbf{g}_{k,r}^t = \sum_{r=0}^{R-1}(\tilde{\mathbf{w}}_{k,r}^t - \tilde{\mathbf{w}}_{k,r+1}^t) = \tilde{\mathbf{w}}_{k,0}^t - \tilde{\mathbf{w}}_{k,R}^t = \tilde{\mathbf{w}}_t - \tilde{\mathbf{w}}_{k,R}^t$. Then $\mathbf{g}_{k,r}^t$ can be considered as a biased estimate of $\nabla F_k(\tilde{\mathbf{w}}_{k,r}^t)$ and the global update rule becomes

$$\tilde{\mathbf{w}}_{t+1} = (1-\beta)\tilde{\mathbf{w}}_t + \frac{\beta}{S}\sum_{k\in\mathcal{S}_t}\tilde{\mathbf{w}}_{k,R}^t = \tilde{\mathbf{w}}_t - \frac{\beta}{S}\sum_{k\in\mathcal{S}_t}(\tilde{\mathbf{w}}_t - \tilde{\mathbf{w}}_{k,R}^t) = \tilde{\mathbf{w}}_t - \underbrace{\eta\beta R}_{=:\tilde{\eta}}\underbrace{\frac{1}{SR}\sum_{k\in\mathcal{S}^t}\sum_{r=0}^{R-1}\mathbf{g}_{k,r}^t}_{=:\mathbf{g}_t}, \tag{15}$$

where $\tilde{\eta}$ and $\mathbf{g}_t$ can be interpreted as the step size and the approximate stochastic gradient of the global update, respectively.

The next two lemmas are from Dinh et al. [13]. Lemma 4 states that the diversity of $F_k$ w.r.t. client sampling can be bounded by the diversity w.r.t. all clients. Lemma 5 gives an upper bound on the drift error of the inner loop.

**Lemma 4** (Dinh et al. [13], Lemma 4, bounded diversity of $F_k$ w.r.t. to client sampling).

$$\mathbb{E}_{\mathcal{S}_t}\left\|\frac{1}{S}\sum_{k\in\mathcal{S}_t}\nabla F_k(\tilde{\mathbf{w}}_t) - \nabla F(\tilde{\mathbf{w}}_t)\right\|_2^2 \leq \frac{N/S-1}{N-1}\sum_{i=1}^N \frac{1}{N}\|\nabla F_k(\tilde{\mathbf{w}}_t) - \nabla F(\tilde{\mathbf{w}}_t)\|_2^2.$$

**Lemma 5** (Bounded client drift error). *Suppose that Assumptions 1, 2, 4 and (7) hold with $0 < s < 1/30$. For $\tilde{\eta} \leq \frac{\beta}{5L_F}$, we have*

$$\frac{1}{NR}\sum_{k=1}^N\sum_{r=0}^{R-1}\mathbb{E}\left[\|\mathbf{g}_{k,r}^t - \nabla F_k(\tilde{\mathbf{w}}_t)\|_2^2\right] \leq 2\lambda^2\delta^2 + \frac{4L_F^2\tilde{\eta}^2}{\beta^2}\left(\frac{7}{N}\sum_{k=1}^N\mathbb{E}\left[\|\nabla F_k(\tilde{\mathbf{w}}_t)\|_2^2\right] + 10\lambda^2\delta^2\right),$$

*where $\delta^2$ is defined in Lemma 2.*

*Proof.* By Proposition 4, we have

$$\mathbb{E}\left[\|\mathbf{g}_{k,r}^t - \nabla F_k(\tilde{\mathbf{w}}_t)\|_2^2\right] \leq 2\mathbb{E}\left[\|\mathbf{g}_{k,r}^t - \nabla F_k(\tilde{\mathbf{w}}_{k,r}^t)\|_2^2\right] + 2\mathbb{E}\left[\|\nabla F_k(\tilde{\mathbf{w}}_{k,r}^t) - \nabla F_k(\tilde{\mathbf{w}}_t)\|_2^2\right]$$

$$\leq 2\mathbb{E}\left[\|\mathbf{g}_{k,r}^t - \nabla F_k(\tilde{\mathbf{w}}_{k,r}^t)\|_2^2\right] + 2L_F^2\mathbb{E}\left[\|\tilde{\mathbf{w}}_{k,r}^t - \tilde{\mathbf{w}}_t\|_2^2\right]$$

$$\leq 2\lambda^2\delta^2 + 2L_F^2\mathbb{E}\left[\|\tilde{\mathbf{w}}_{k,r}^t - \tilde{\mathbf{w}}_t\|_2^2\right], \tag{16}$$

where the second inequality is by Proposition 9, and the last inequality is by Lemma 2. Next, we bound the second term $\|\tilde{\mathbf{w}}_{k,r}^t - \tilde{\mathbf{w}}_t\|_2^2$. By Proposition 4, for $r \geq 1$, we have

$$\mathbb{E}\left[\|\tilde{\mathbf{w}}_{k,r}^t - \tilde{\mathbf{w}}_t\|_2^2\right] = \mathbb{E}\left[\|\tilde{\mathbf{w}}_{k,r-1}^t - \tilde{\mathbf{w}}_t - \eta\mathbf{g}_{k,r-1}^t\|_2^2\right]$$

$$\leq \left(1 + \frac{1}{4R}\right)\mathbb{E}\left[\|\tilde{\mathbf{w}}_{k,r-1}^t - \tilde{\mathbf{w}}_t - \eta\nabla F_k(\tilde{\mathbf{w}}_t)\|_2^2\right] + (1+4R)\eta^2\mathbb{E}\left[\|\mathbf{g}_{k,r-1}^t - \nabla F_k(\tilde{\mathbf{w}}_t)\|_2^2\right]$$

$$\leq \left(1 + \frac{1}{4R}\right)^2\mathbb{E}\left[\|\tilde{\mathbf{w}}_{k,r-1}^t - \tilde{\mathbf{w}}_t\|_2^2\right] + \left(1 + \frac{1}{4R}\right)(1+4R)\eta^2\mathbb{E}\left[\|\nabla F_k(\tilde{\mathbf{w}}_t)\|_2^2\right]$$

$$+ (1+4R)\eta^2\mathbb{E}\left[\|\mathbf{g}_{k,r-1}^t - \nabla F_k(\tilde{\mathbf{w}}_t)\|_2^2\right]. \tag{17}$$

Recall that $\tilde{\eta} = \eta\beta R \leq \frac{\beta}{5L_F}$ and $R \geq 1$. Then we have $\left(1 + \frac{1}{4R}\right)^2 \leq 1 + \frac{9}{16R}$, $\left(1 + \frac{1}{4R}\right)(1+4R) \leq \frac{25}{4}R$ and $(1+4R)\eta^2 \leq 5R\eta^2 \leq 5R\frac{1}{25R^2L_F^2} = \frac{1}{5RL_F^2}$. Substituting these inequalities and (16) into (17) yields

$$\mathbb{E}\left[\|\tilde{\mathbf{w}}_{k,r}^t - \tilde{\mathbf{w}}_t\|_2^2\right] \leq \left(1 + \frac{9}{16R}\right)\mathbb{E}\left[\|\tilde{\mathbf{w}}_{k,r-1}^t - \tilde{\mathbf{w}}_t\|_2^2\right] + \frac{25}{4}R\eta^2\mathbb{E}\left[\|\nabla F_k(\tilde{\mathbf{w}}_t)\|_2^2\right]$$

$$+ 10R\eta^2\lambda^2\delta^2 + \frac{2}{5R}\mathbb{E}\left[\left\|\tilde{\mathbf{w}}_{k,r-1}^t - \tilde{\mathbf{w}}_t\right\|_2^2\right]$$

$$\leq \left(1 + \frac{1}{R}\right)\mathbb{E}\left[\left\|\tilde{\mathbf{w}}_{k,r-1}^t - \tilde{\mathbf{w}}_t\right\|_2^2\right] + 7R\eta^2\mathbb{E}\left[\left\|\nabla F_k(\tilde{\mathbf{w}}_t)\right\|_2^2\right] + 10R\eta^2\lambda^2\delta^2.$$

(18)

Note that (18) holds for any $1 \leq r \leq R$ and $\tilde{\mathbf{w}}_{k,0}^t = \tilde{\mathbf{w}}_t$. Applying (18) recursively, we obtain

$$\mathbb{E}\left[\left\|\tilde{\mathbf{w}}_{k,r}^t - \tilde{\mathbf{w}}_t\right\|_2^2\right] \leq \left(7R\eta^2\mathbb{E}\left[\left\|\nabla F_k(\tilde{\mathbf{w}}_t)\right\|_2^2\right] + 10R\eta^2\lambda^2\delta^2\right)\sum_{i=0}^{R-1}\left(1 + \frac{1}{R}\right)^i.$$

Since $(1 + x/n)^n \leq e^x$ for any $x \in \mathbb{R}$, we have $\sum_{i=0}^{R-1}(1+1/R)^i = \frac{(1+1/R)^R-1}{1/R} \leq \frac{e-1}{1/R} \leq 2R$. This implies

$$\mathbb{E}\left[\left\|\tilde{\mathbf{w}}_{k,r}^t - \tilde{\mathbf{w}}_t\right\|_2^2\right] \leq \frac{14\tilde{\eta}^2}{\beta^2}\mathbb{E}\left[\left\|\nabla F_k(\tilde{\mathbf{w}}_t)\right\|_2^2\right] + \frac{20\tilde{\eta}^2\lambda^2\delta^2}{\beta^2}.$$

(19)

Substituting (19) into (16) yields

$$\mathbb{E}\left[\left\|\mathbf{g}_{k,r}^t - \nabla F_k(\tilde{\mathbf{w}}_t)\right\|_2^2\right] \leq 2\lambda^2\delta^2 + \frac{4L_F^2\tilde{\eta}^2}{\beta^2}\left(7\mathbb{E}\left[\left\|\nabla F_k(\tilde{\mathbf{w}}_t)\right\|_2^2\right] + 10\lambda^2\delta^2\right).$$

Taking the average over the indices $k$ and $r$, we obtain the desired result. $\qquad\square$

Now we complete the proof of Theorem 1.

*Proof of Theorem 1.* We first assume $\eta \leq \frac{\hat{\eta}_0}{\beta R}$. The exact value of $\eta$ will be determined later. By Proposition 7, we have (7) holds with probability at least $1 - 2\exp(-cd_{\text{sub}})$ and $0 < s < 1/30$ as long as $d_{\text{sub}}/d$ is sufficiently small. Throughout the proof, we assume this inequality holds.

Recall that with $\tilde{\eta}$ and $\mathbf{g}_t$ defined in (15), we have $\tilde{\mathbf{w}}_{t+1} = \tilde{\mathbf{w}}_t - \tilde{\eta}\mathbf{g}_t$. By Proposition 9, $F_k$ is $L_F$-smooth, then $F$ is also $L_F$-smooth. This implies that

$$\mathbb{E}\left[F(\tilde{\mathbf{w}}_{t+1}) - F(\tilde{\mathbf{w}}_t)\right]$$

$$\leq \mathbb{E}\left[\langle\nabla F(\tilde{\mathbf{w}}_t), \tilde{\mathbf{w}}_{t+1} - \tilde{\mathbf{w}}_t\rangle\right] + \frac{L_F}{2}\mathbb{E}\left[\left\|\tilde{\mathbf{w}}_{t+1} - \tilde{\mathbf{w}}_t\right\|_2^2\right]$$

$$= -\tilde{\eta}\mathbb{E}\left[\langle\nabla F(\tilde{\mathbf{w}}_t), \mathbf{g}_t\rangle\right] + \frac{\tilde{\eta}^2 L_F}{2}\mathbb{E}\left[\left\|\mathbf{g}_t\right\|_2^2\right]$$

$$= -\tilde{\eta}\mathbb{E}\left[\left\|\nabla F(\tilde{\mathbf{w}}_t)\right\|_2^2\right] - \tilde{\eta}\mathbb{E}\left[\langle\nabla F(\tilde{\mathbf{w}}_t), \mathbf{g}_t - \nabla F(\tilde{\mathbf{w}}_t)\rangle\right] + \frac{\tilde{\eta}^2 L_F}{2}\mathbb{E}\left[\left\|\mathbf{g}_t\right\|_2^2\right]$$

$$\leq -\tilde{\eta}\mathbb{E}\left[\left\|\nabla F(\tilde{\mathbf{w}}_t)\right\|_2^2\right] + \frac{\tilde{\eta}}{2}\mathbb{E}\left[\left\|\nabla F(\tilde{\mathbf{w}}_t)\right\|_2^2\right] + \frac{\tilde{\eta}}{2}\mathbb{E}\left[\left\|\frac{1}{NR}\sum_{k=1}^{N}\sum_{r=0}^{R-1}(\mathbf{g}_{k,r}^t - \nabla F_k(\tilde{\mathbf{w}}_t))\right\|_2^2\right] + \frac{\tilde{\eta}^2 L_F}{2}\mathbb{E}\left[\left\|\mathbf{g}_t\right\|_2^2\right],$$

(20)

where $\mathbf{g}_{k,r}^t$ is defined in (14) and the last inequality is by Cauchy-Schwarz inequality. Next from the proof of Lemma 3 in Dinh et al. [13], we have

$$\mathbb{E}_{\mathcal{S}_t}\left[\left\|\mathbf{g}_t\right\|_2^2\right] \leq 3\mathbb{E}_{\mathcal{S}_t}\left[\frac{1}{NR}\sum_{k=1}^{N}\sum_{r=0}^{R-1}\left\|\mathbf{g}_{k,r}^t - \nabla F_k(\tilde{\mathbf{w}}_t)\right\|_2^2 + \left\|\frac{1}{S}\sum_{k\in\mathcal{S}_t}\nabla F_k(\tilde{\mathbf{w}}_t) - \nabla F(\tilde{\mathbf{w}}_t)\right\|_2^2 + \left\|\nabla F(\tilde{\mathbf{w}}_t)\right\|_2^2\right].$$

(21)

We defer the proof of (21) to the end of this subsection. Recall that $\hat{\eta}_0 = \frac{1}{90\lambda^2 L_F}$, $\beta \geq 1$ and $\lambda \geq 1$. $\eta \leq \frac{\hat{\eta}_0}{\beta R}$ implies that $\tilde{\eta} = \beta R\eta \leq \frac{\beta}{5L_F}$. Substituting (21) into (20) yields

$$\mathbb{E}\left[F(\tilde{\mathbf{w}}_{t+1}) - F(\tilde{\mathbf{w}}_t)\right]$$

$$\leq -\frac{\tilde{\eta}}{2}\mathbb{E}\left[\left\|\nabla F(\tilde{\mathbf{w}}_t)\right\|_2^2\right] + \left(\frac{\tilde{\eta}}{2} + \frac{3\tilde{\eta}^2 L_F}{2}\right)\frac{1}{NR}\sum_{r=0}^{R-1}\sum_{k=1}^{N}\mathbb{E}\left[\left\|\mathbf{g}_{k,r}^t - \nabla F_k(\tilde{\mathbf{w}}_t)\right\|_2^2\right]$$

$$+ \frac{3\tilde{\eta}^2 L_F}{2}\mathbb{E}\left[\frac{1}{S}\sum_{k\in\mathcal{S}_t}\nabla F_k(\tilde{\mathbf{w}}_t) - \nabla F(\tilde{\mathbf{w}}_t)\right] + \frac{3\tilde{\eta}^2 L_F}{2}\mathbb{E}\left[\left\|\nabla F(\tilde{\mathbf{w}}_t)\right\|_2^2\right]$$

$$\leq -\frac{\tilde{\eta}(1-3\tilde{\eta}L_F)}{2}\mathbb{E}\left[\|\nabla F(\tilde{\mathbf{w}}_t)\|_2^2\right] + \frac{3\tilde{\eta}^2 L_F}{2}\frac{N/S-1}{N-1}\sum_{k=1}^{N}\frac{1}{N}\mathbb{E}\left[\|\nabla F_k(\tilde{\mathbf{w}}_t) - \nabla F(\tilde{\mathbf{w}}_t)\|_2^2\right]$$

$$+ \frac{\tilde{\eta}(1+3\tilde{\eta}L_F)}{2}\left[2\lambda^2\delta^2 + \frac{4L_F^2\tilde{\eta}^2}{\beta^2}\left(\frac{7}{N}\sum_{k=1}^{N}\mathbb{E}\left[\|\nabla F_k(\tilde{\mathbf{w}}_t) - \nabla F(\tilde{\mathbf{w}}_t)\|_2^2\right] + 7\mathbb{E}\left[\|\nabla F(\tilde{\mathbf{w}}_t)\|_2^2\right] + 10\lambda^2\delta^2\right)\right]$$

$$\leq -\frac{\tilde{\eta}(1-3\tilde{\eta}L_F)}{2}\mathbb{E}\left[\|\nabla F(\tilde{\mathbf{w}}_t)\|_2^2\right] + \frac{3\tilde{\eta}^2 L_F}{2}\frac{N/S-1}{N-1}\left(3\sigma_F^2 + \frac{10L^2}{\lambda^2-10L^2}\mathbb{E}\left[\|\nabla F(\tilde{\mathbf{w}}_t)\|_2^2\right]\right)$$

$$+ \frac{\tilde{\eta}(1+3\tilde{\eta}L_F)}{2}\left[2\lambda^2\delta^2 + \frac{4L_F^2\tilde{\eta}^2}{\beta^2}\left(21\sigma_F^2 + \frac{7\lambda^2}{\lambda^2-10L^2}\mathbb{E}\left[\|\nabla F(\tilde{\mathbf{w}}_t)\|_2^2\right] + 10\lambda^2\delta^2\right)\right]$$

$$= -\tilde{\eta}\left[\frac{1}{2} - \tilde{\eta}L_F\left(\frac{3}{2} + \frac{15L^2}{\lambda^2-10L^2}\frac{N/S-1}{N-1} + \frac{14(1+3\tilde{\eta}L_F)\lambda^2\tilde{\eta}L_F}{\beta^2(\lambda^2-10L^2)}\right)\right]\mathbb{E}\left[\|\nabla F(\tilde{\mathbf{w}}_t)\|_2^2\right]$$

$$+ \frac{\tilde{\eta}^3}{\beta^2}(1+3\tilde{\eta}L_F)2L_F^2(21\sigma_F^2 + 10\lambda^2\delta^2) + \tilde{\eta}^2\frac{9}{2}L_F\sigma_F^2\frac{N/S-1}{N-1} + \tilde{\eta}(1+3\tilde{\eta}L_F)\lambda^2\delta^2,$$

where the second inequality is by Lemmas 5 and 4 and the fact that $\mathbb{E}[\|X\|_2^2] = \mathbb{E}[\|X - \mathbb{E}[X]\|_2^2] + \|\mathbb{E}[X]\|_2^2$ for a random vector $X$, and the last inequality is by Lemma 3.

Clearly, we also have $\tilde{\eta} \leq \frac{\beta}{2L_F}$, which implies that $1 + 3\tilde{\eta}L_F \leq 1 + 3\beta/2 \leq 3\beta$. Recall that $\lambda^2 - 10L^2 \geq 1$ and $\frac{N/S-1}{N-1} \leq 1$. Then we have

$$\frac{3}{2} + \frac{15L^2}{\lambda^2-10L^2}\frac{N/S-1}{N-1} + \frac{14(1+3\tilde{\eta}L_F)\lambda^2\tilde{\eta}L_F}{\beta^2(\lambda^2-10L^2)} \leq \frac{3}{2} + 15L^2 + 21\lambda^2 \leq \frac{45}{2}\lambda^2.$$

Since $\tilde{\eta} = \beta R\eta \leq \frac{1}{90\lambda^2 L_F}$, then

$$\frac{1}{2} - \tilde{\eta}L_F\left(\frac{3}{2} + \frac{15L^2}{\lambda^2-10L^2}\frac{N/S-1}{N-1} + \frac{14(1+3\tilde{\eta}L_F)\lambda^2\tilde{\eta}L_F}{\beta^2(\lambda^2-10L^2)}\right) \geq \frac{1}{2} - \frac{45\lambda^2\tilde{\eta}L_F}{2} \geq \frac{1}{4},$$

Moreover, the choice of $\lambda$ implies $\lambda \geq 1$. Then we have $1 + 3\tilde{\eta}L_F \leq 1 + \frac{1}{15\lambda^2} \leq 2$. It follows that

$$\mathbb{E}\left[F(\tilde{\mathbf{w}}_{t+1}) - F(\tilde{\mathbf{w}}_t)\right] \leq -\frac{\tilde{\eta}}{4}\mathbb{E}\left[\|\nabla F(\tilde{\mathbf{w}}_t)\|_2^2\right] + \frac{\tilde{\eta}^3}{\beta^2}\underbrace{4L_F^2(21\sigma_F^2+10\lambda^2\delta^2)}_{=:C_1} + \tilde{\eta}^2\underbrace{5L_F\sigma_F^2\frac{N/S-1}{N-1}}_{=:C_2} + \tilde{\eta}\underbrace{2\lambda^2\delta^2}_{=:C_3}.$$

By rearranging the terms and telescoping, we obtain

$$\frac{1}{4T}\sum_{t=0}^{T-1}\mathbb{E}\left[\|\nabla F(\tilde{\mathbf{w}}_t)\|_2^2\right] \leq \frac{\mathbb{E}\left[F(\tilde{\mathbf{w}}_0) - F(\tilde{\mathbf{w}}_T)\right]}{\tilde{\eta}T} + \frac{\tilde{\eta}^2}{\beta^2}C_1 + \tilde{\eta}C_2 + C_3. \tag{22}$$

Now we use the techniques in Karimireddy et al. [33], Arjevani et al. [2], Stich [58] to specify the value of $\tilde{\eta}$. Recall that we need to ensure $\eta \leq \frac{\hat{\eta}_0}{\beta R}$ (i.e., $\tilde{\eta} \leq \hat{\eta}_0$).

- If $\hat{\eta}_0^3 \geq \frac{\beta^2\Delta_F}{TC_1}$ or $\hat{\eta}_0^2 \geq \frac{\Delta_F}{TC_2}$, then the first term on the right-hand side of (22) is no large than the sum of the second and third terms. We choose $\tilde{\eta} = \min\left\{\left(\frac{\beta^2\Delta_F}{TC_1}\right)^{1/3}, \left(\frac{\Delta_F}{TC_2}\right)^{1/2}\right\}$. Then we have $\tilde{\eta} \leq \hat{\eta}_0$ and

$$\frac{1}{4T}\sum_{t=0}^{T-1}\mathbb{E}\left[\|\nabla F(\tilde{\mathbf{w}}_t)\|_2^2\right] \leq 2\frac{\Delta_F^{2/3}C_1^{1/3}}{(\beta T)^{2/3}} + 2\frac{(\Delta_F C_2)^{1/2}}{\sqrt{T}} + C_3.$$

- If $\hat{\eta}_0^3 < \frac{\beta^2\Delta_F}{TC_1}$ and $\hat{\eta}_0^2 < \frac{\Delta_F}{TC_2}$, then the first term on the right-hand side of (22) is larger than the second and third terms. We choose $\tilde{\eta} = \hat{\eta}_0$ and obtain

$$\frac{1}{4T}\sum_{t=0}^{T-1}\mathbb{E}\left[\|\nabla F(\tilde{\mathbf{w}}_t)\|_2^2\right] \leq 3\frac{\Delta_F}{\hat{\eta}_0 T} + C_3.$$

Combine the two cases and sampling $t^*$ uniformly from $\{0, 1, \ldots, T-1\}$, we have

$$\mathbb{E}\left[\|\nabla F(\tilde{\mathbf{w}}_{t^*})\|_2^2\right] = \frac{1}{T}\sum_{t=0}^{T-1}\mathbb{E}\left[\|\nabla F(\tilde{\mathbf{w}}_t)\|_2^2\right]$$

$$\leq \mathcal{O}\left(\frac{\Delta_F}{\hat{\eta}_0 T} + \frac{\Delta_F^{2/3}L_F^{2/3}\left(\sigma_F^2 + \lambda^2\delta^2\right)^{1/3}}{\beta^{2/3}T^{2/3}} + \frac{\left(\Delta_F L_F \sigma_F^2(N/S-1)\right)^{1/2}}{\sqrt{TN}} + \lambda^2\delta^2\right) =: \mathcal{O}_0.$$

Now we prove the second inequality. Let $\hat{\mathbf{y}}_k^t = \operatorname{argmin}_{\mathbf{y}_k \in \mathbb{R}^{d_{\mathrm{sub}}}}\left\{f_k(\boldsymbol{P}^\top\mathbf{y}_k) + \frac{\lambda}{2}\left\|\tilde{\mathbf{w}}_t - \boldsymbol{P}\boldsymbol{P}^\top\mathbf{y}_k\right\|_2^2\right\}$
and $\hat{\mathbf{x}}_k^t = \boldsymbol{P}^\top\hat{\mathbf{y}}_k^t$. By Proposition 4, we have

$$\frac{1}{N}\sum_{k=1}^{N}\mathbb{E}\left[\left\|\boldsymbol{P}\mathbf{x}_k^t - \tilde{\mathbf{w}}_t\right\|_2^2\right] \leq \frac{2}{N}\sum_{k=1}^{N}\mathbb{E}\left[\left\|\boldsymbol{P}\mathbf{x}_k^t - \boldsymbol{P}\hat{\mathbf{x}}_k^t\right\|_2^2 + \left\|\boldsymbol{P}\hat{\mathbf{x}}_k^t - \tilde{\mathbf{w}}_t\right\|_2^2\right]$$

$$\leq 2\delta^2 + \frac{2}{N}\sum_{i=1}^{N}\frac{\mathbb{E}\left[\|\nabla F_k(\tilde{\mathbf{w}}_t)\|_2^2\right]}{\lambda^2}, \tag{23}$$

where the last inequality is by Proposition 9 and Lemma 2. Due to Lemma 3 and the fact that $\mathbb{E}[\|X\|_2^2] = \mathbb{E}[\|X - \mathbb{E}[X]\|_2^2] + \|\mathbb{E}[X]\|_2^2$ for a random vector $X$, we have

$$\frac{1}{N}\sum_{i=1}^{N}\mathbb{E}\left[\|\nabla F_k(\tilde{\mathbf{w}}_t)\|_2^2\right] \leq \frac{1}{N}\sum_{i=1}^{N}\mathbb{E}\left[\|\nabla F_k(\tilde{\mathbf{w}}_t) - \nabla F(\tilde{\mathbf{w}}_t)\|_2^2 + \|\nabla F(\tilde{\mathbf{w}}_t)\|_2^2\right]$$

$$\leq 3\sigma_F^2 + \frac{\lambda^2}{\lambda^2 - 10L^2}\|\nabla F(\tilde{\mathbf{w}}_t)\|_2^2. \tag{24}$$

Substituting (24) into (23) and taking the average over the index $t$, we obtain

$$\frac{1}{TN}\sum_{t=0}^{T-1}\sum_{k=1}^{N}\mathbb{E}\left[\left\|\boldsymbol{P}\mathbf{x}_k^t - \tilde{\mathbf{w}}_t\right\|_2^2\right] \leq \frac{2}{\lambda^2 - 10L^2}\frac{1}{T}\sum_{t=0}^{T-1}\mathbb{E}\left[\|\nabla F(\tilde{\mathbf{w}}_t)\|_2^2\right] + 2\delta^2 + \frac{6\sigma_F^2}{\lambda^2}$$

$$\leq \mathcal{O}_0 + \mathcal{O}\left(\delta^2 + \frac{\sigma_F^2}{\lambda^2}\right),$$

where the last inequality is due to $\lambda \geq \sqrt{10L^2 + 1}$. This completes the proof. $\qquad\square$

Now we prove (21).

*Proof of (21).* By Proposition 4, we have

$$\mathbb{E}_{\mathcal{S}_t}\left[\|\mathbf{g}_t\|_2^2\right] \leq 3\mathbb{E}_{\mathcal{S}_t}\left[\left\|\frac{1}{SR}\sum_{k \in \mathcal{S}_t}\sum_{r=0}^{R-1}(\mathbf{g}_{k,r}^t - \nabla F_k(\tilde{\mathbf{w}}_t))\right\|_2^2 + \left\|\frac{1}{S}\sum_{k \in \mathcal{S}_t}\nabla F_k(\tilde{\mathbf{w}}_t) - \nabla F(\tilde{\mathbf{w}}_t)\right\|_2^2 + \|\nabla F(\tilde{\mathbf{w}}_t)\|_2^2\right]$$

$$\leq 3\mathbb{E}_{\mathcal{S}_t}\left[\frac{1}{SR}\sum_{k \in \mathcal{S}_t}\sum_{r=0}^{R-1}\left\|\mathbf{g}_{k,r}^t - \nabla F_k(\tilde{\mathbf{w}}_t)\right\|_2^2 + \left\|\frac{1}{S}\sum_{k \in \mathcal{S}_t}\nabla F_k(\tilde{\mathbf{w}}_t) - \nabla F(\tilde{\mathbf{w}}_t)\right\|_2^2 + \|\nabla F(\tilde{\mathbf{w}}_t)\|_2^2\right].$$

If we only consider the randomness from the sampling of $\mathcal{S}_t$, $\mathbf{g}_{k,r}^t$ and $\nabla F_k(\tilde{\mathbf{w}}_t)$ become constant vectors. Use $\mathbb{1}_A$ to denote the indicator function of an event $A$. Uniform sampling implies $\mathbb{E}_{\mathcal{S}_t}\left[\mathbb{1}_{k \in \mathcal{S}_t}\right] = \frac{S}{N}$. Then we have

$$\frac{1}{SR}\mathbb{E}_{\mathcal{S}_t}\left[\sum_{k \in \mathcal{S}_t}\sum_{r=0}^{R-1}\left\|\mathbf{g}_{k,r}^t - \nabla F_k(\tilde{\mathbf{w}}_t)\right\|_2^2\right] = \frac{1}{SR}\sum_{k=1}^{N}\sum_{r=0}^{R-1}\left\|\mathbf{g}_{k,r}^t - \nabla F_k(\tilde{\mathbf{w}}_t)\right\|_2^2\mathbb{E}_{\mathcal{S}_t}\left[\mathbb{1}_{k \in \mathcal{S}_t}\right]$$

$$= \frac{1}{NR}\sum_{k=1}^{N}\sum_{r=0}^{R-1}\left\|\mathbf{g}_{k,r}^t - \nabla F_k(\tilde{\mathbf{w}}_t)\right\|_2^2,$$

This completes the proof. $\qquad\square$

# B Federated Linear Regression

In this section, we consider a federated linear regression model, which is different from that in Li et al. [44].

Suppose that the true parameter on client $k$ is $\mathbf{w}_k$, there are $n$ samples on each client and the covariate on client $k$ is $\{\xi_{k,i}\}_{i=1}^n$ and fixed. The observations are generated by $y_{k,i} = \xi_{k,i}^\top \mathbf{w}_k + z_{k,i}$ where the noises $z_{k,i}$ are i.i.d. and distributed as $\mathcal{N}(0, \sigma^2)$. Then the loss on client $k$ is $f_k(\mathbf{x}_k) = \frac{1}{2n} \sum_{i=1}^n (y_{k,i} - \xi_{k,i}^\top \mathbf{x}_k)^2$

Li et al. [44] focused on a Bayesian framework where the true parameters $\mathbf{w}_k$ are drawn from a Gaussian distribution and the mean of this Gaussian distribution is drawn from the non-informative prior, while we treat $\mathbf{w}_k$ as fixed vectors. We compare the performance of `local` (pure local training), `FedAvg` [51], `pFedMe` [13], `Ditto` [44] and our method `lp-proj-2` in terms of test losses, robustness and fairness.

## B.1 Solutions of Different Methods

In this subsection, we derive the solutions of different methods. Let $\boldsymbol{\Xi}_k = (\xi_{k,1}, \xi_{k,2}, \ldots, \xi_{k,n})^\top$ and $\mathbf{y}_k = (y_{k,1}, y_{k,2}, \ldots, y_{k,n})^\top$. Then the loss on client $k$ can be rewritten as $f_k(\mathbf{x}_k) = \frac{1}{2n} \|\boldsymbol{\Xi}_k \mathbf{x}_k - \mathbf{y}_k\|_2^2$. Suppose $\mathrm{rank}(\boldsymbol{\Xi}_k) = d$. The least square estimator of $\mathbf{w}_k$ is

$$\hat{\mathbf{w}}_k = (\boldsymbol{\Xi}_k^\top \boldsymbol{\Xi}_k)^{-1} \boldsymbol{\Xi}_k^\top \mathbf{y}_k. \tag{21}$$

`local`   For pure local training, the solution on client $k$ is defined as $\mathbf{w}_k^{\mathrm{loc}} = \mathrm{argmin}_{\mathbf{x}_k \in \mathbb{R}^d} f_k(\mathbf{x}_k) = \hat{\mathbf{w}}_k$.

`FedAvg`   For FedAvg, the solution is defined as $\mathbf{w}^{\mathrm{Avg}} = \mathrm{argmin}_{\mathbf{w} \in \mathbb{R}^d} \frac{1}{N} \sum_{k=1}^N f_k(\mathbf{w})$. One can check that $\mathbf{w}^{\mathrm{Avg}} = \left( \sum_{k=1}^N \boldsymbol{\Xi}_k^\top \boldsymbol{\Xi}_k \right)^{-1} \sum_{k=1}^N \boldsymbol{\Xi}_k^\top \mathbf{y}_k = \left( \sum_{k=1}^N \boldsymbol{\Xi}_k^\top \boldsymbol{\Xi}_k \right)^{-1} \sum_{k=1}^N \boldsymbol{\Xi}_k^\top \boldsymbol{\Xi}_k \hat{\mathbf{w}}_k$.

`pFedMe`   pFedMe corresponds to our method with $\boldsymbol{P} = \boldsymbol{I}_d$. Then the optimization problem is $\min_{\mathbf{w} \in \mathbb{R}^d} F(\mathbf{w}) = \frac{1}{N} \sum_{k=1}^N F_k(\mathbf{w})$ where $F_k(\mathbf{w}) = \min_{\mathbf{x}_k \in \mathbb{R}^d} \{ f_k(\mathbf{x}_k) + \frac{\lambda}{2} \|\mathbf{w} - \mathbf{x}_k\|_2^2 \}$. The solution of the global model is defined as $\mathbf{w}^{\mathrm{Me}} = \mathrm{argmin}_{\mathbf{w} \in \mathbb{R}^d} \frac{1}{N} \sum_{k=1}^N F_k(\mathbf{w})$ and the solution of the local model is defined as $\mathbf{x}_k^{\mathrm{Me}} = \mathrm{argmin}_{\mathbf{x}_k \in \mathbb{R}^d} \left\{ f_k(\mathbf{x}_k) + \frac{\lambda}{2} \|\mathbf{w}^{\mathrm{Me}} - \mathbf{x}_k\|_2^2 \right\}$.

Now we give the explicit forms of $\mathbf{w}^{\mathrm{Me}}$ and $\mathbf{x}_k^{\mathrm{Me}}$. Define $\hat{\mathbf{x}}_k(\mathbf{w}) := \mathrm{argmin}_{\mathbf{x}_k \in \mathbb{R}^d} \{ f_k(\mathbf{x}_k) + \frac{\lambda}{2} \|\mathbf{w} - \mathbf{x}_k\|_2^2 \}$. It is easy to check $\hat{\mathbf{x}}_k(\mathbf{w}) = (\boldsymbol{\Xi}_k^\top \boldsymbol{\Xi}_k / n + \lambda \boldsymbol{I}_d)^{-1} (\boldsymbol{\Xi}_k^\top \mathbf{y}_k / n + \lambda \mathbf{w})$. Then we have

$$
\begin{aligned}
F_k(\mathbf{w}) &= f_k(\hat{\mathbf{x}}_k(\mathbf{w})) + \frac{\lambda}{2} \|\mathbf{w} - \hat{\mathbf{x}}_k(\mathbf{w})\|_2^2 \\
&= -\frac{1}{2} \left( \frac{\boldsymbol{\Xi}_k^\top \mathbf{y}_k}{n} + \lambda \mathbf{w} \right)^\top \left( \frac{\boldsymbol{\Xi}_k^\top \boldsymbol{\Xi}_k}{n} + \lambda \boldsymbol{I}_d \right)^{-1} \left( \frac{\boldsymbol{\Xi}_k^\top \mathbf{y}_k}{n} + \lambda \mathbf{w} \right) + \frac{\lambda}{2} \|\mathbf{w}\|_2^2 + \frac{\|\mathbf{y}_k\|_2^2}{2n} \\
&= \frac{\lambda \mathbf{w}^\top}{2} \left( \frac{\boldsymbol{\Xi}_k^\top \boldsymbol{\Xi}_k}{n} + \lambda \boldsymbol{I}_d \right)^{-1} \left( \frac{\boldsymbol{\Xi}_k^\top \boldsymbol{\Xi}_k \mathbf{w}}{n} \right) - \lambda \mathbf{w}^\top \left( \frac{\boldsymbol{\Xi}_k^\top \boldsymbol{\Xi}_k}{n} + \lambda \boldsymbol{I}_d \right)^{-1} \frac{\boldsymbol{\Xi}_k^\top \mathbf{y}_k}{n} \\
&\quad + \frac{\|\mathbf{y}_k\|_2^2}{2n} - \frac{\mathbf{y}_k^\top \boldsymbol{\Xi}_k}{2n} \left( \frac{\boldsymbol{\Xi}_k^\top \boldsymbol{\Xi}_k}{n} + \lambda \boldsymbol{I}_d \right)^{-1} \frac{\boldsymbol{\Xi}_k^\top \mathbf{y}_k}{n}.
\end{aligned}
$$

It follows that

$$F(\mathbf{w}) = \frac{1}{N} \sum_{k=1}^N \frac{\lambda \mathbf{w}^\top}{2} \left( \frac{\boldsymbol{\Xi}_k^\top \boldsymbol{\Xi}_k}{n} + \lambda \boldsymbol{I}_d \right)^{-1} \left( \frac{\boldsymbol{\Xi}_k^\top \boldsymbol{\Xi}_k \mathbf{w}}{n} \right) - \frac{1}{N} \sum_{k=1}^N \lambda \mathbf{w}^\top \left( \frac{\boldsymbol{\Xi}_k^\top \boldsymbol{\Xi}_k}{n} + \lambda \boldsymbol{I}_d \right)^{-1} \frac{\boldsymbol{\Xi}_k^\top \mathbf{y}_k}{n} + C_0,$$

where $C_0$ is a constant number. Then $\mathbf{w}^{\text{Me}}$ is the solution to

$$\frac{1}{2}\sum_{k=1}^{N}\left[\left(\frac{\Xi_k^\top\Xi_k}{n}+\lambda\boldsymbol{I}_d\right)^{-1}\frac{\Xi_k^\top\Xi_k}{n}+\frac{\Xi_k^\top\Xi_k}{n}\left(\frac{\Xi_k^\top\Xi_k}{n}+\lambda\boldsymbol{I}_d\right)^{-1}\right]\mathbf{w}=\sum_{k=1}^{N}\left(\frac{\Xi_k^\top\Xi_k}{n}+\lambda\boldsymbol{I}_d\right)^{-1}\frac{\Xi_k^\top\mathbf{y}_k}{n}.$$

(22)

By the Sherman–Morrison–Woodbury formula, we have

$$\left(\frac{\Xi_k^\top\Xi_k}{n}+\lambda\boldsymbol{I}_d\right)^{-1}=\frac{\boldsymbol{I}_d}{\lambda}-\frac{\Xi_k^\top}{\lambda}\left(n\boldsymbol{I}_n+\frac{\Xi_k\Xi_k^\top}{\lambda}\right)^{-1}\frac{\Xi_k}{\lambda}.$$

It follows that

$$\left(\frac{\Xi_k^\top\Xi_k}{n}+\lambda\boldsymbol{I}_d\right)^{-1}\frac{\Xi_k^\top\Xi_k}{n}=\frac{\Xi_k^\top\Xi_k}{\lambda n}-\frac{\Xi_k^\top}{\lambda}\left(n\boldsymbol{I}_n+\frac{\Xi_k\Xi_k^\top}{\lambda}\right)^{-1}\frac{\Xi_k\Xi_k^\top}{\lambda}\frac{\Xi_k}{n}$$

$$=\frac{\Xi_k^\top}{\lambda}\left(n\boldsymbol{I}_n+\frac{\Xi_k\Xi_k^\top}{\lambda}\right)^{-1}\Xi_k.$$

Similarly, we can obtain $\frac{\Xi_k^\top\Xi_k}{n}\left(\frac{\Xi_k^\top\Xi_k}{n}+\lambda\boldsymbol{I}_d\right)^{-1}=\frac{\Xi_k^\top}{\lambda}\left(n\boldsymbol{I}_n+\frac{\Xi_k\Xi_k^\top}{\lambda}\right)^{-1}\Xi_k.$ This implies

that $\left(\frac{\Xi_k^\top\Xi_k}{n}+\lambda\boldsymbol{I}_d\right)^{-1}\frac{\Xi_k^\top\Xi_k}{n}=\frac{\Xi_k^\top\Xi_k}{n}\left(\frac{\Xi_k^\top\Xi_k}{n}+\lambda\boldsymbol{I}_d\right)^{-1}.$ Thus, the solution to (22) is

$$\mathbf{w}^{\text{Me}}=\left[\sum_{k=1}^{N}\left(\frac{\Xi_k^\top\Xi_k}{n}+\lambda\boldsymbol{I}_d\right)^{-1}\frac{\Xi_k^\top\Xi_k}{n}\right]^{-1}\left[\sum_{k=1}^{N}\left(\frac{\Xi_k^\top\Xi_k}{n}+\lambda\boldsymbol{I}_d\right)^{-1}\frac{\Xi_k^\top\mathbf{y}_k}{n}\right]$$

$$=\left[\sum_{k=1}^{N}\left(\frac{\Xi_k^\top\Xi_k}{n}+\lambda\boldsymbol{I}_d\right)^{-1}\frac{\Xi_k^\top\Xi_k}{n}\right]^{-1}\left[\sum_{k=1}^{N}\left(\frac{\Xi_k^\top\Xi_k}{n}+\lambda\boldsymbol{I}_d\right)^{-1}\frac{\Xi_k^\top\Xi_k}{n}\hat{\mathbf{w}}_k\right],$$

which can be seen as a weighted average of $\hat{\mathbf{w}}_k$ with weight $\left(\frac{\Xi_k^\top\Xi_k}{n}+\lambda\boldsymbol{I}_d\right)^{-1}\frac{\Xi_k^\top\Xi_k}{n}$. Then solution of the local model is $\mathbf{x}_k^{\text{Me}}=\hat{\mathbf{x}}_k(\mathbf{w}^{\text{Me}})=(\Xi_k^\top\Xi_k/n+\lambda\boldsymbol{I}_d)^{-1}(\Xi_k^\top\mathbf{y}_k/n+\lambda\mathbf{w}^{\text{Me}})=(\Xi_k^\top\Xi_k/n+\lambda\boldsymbol{I}_d)^{-1}(\Xi_k^\top\Xi_k\hat{\mathbf{w}}_k/n+\lambda\mathbf{w}^{\text{Me}}).$

Ditto    For `Ditto`, the solution of the global model is the same as that of `FedAvg`, i.e., $\mathbf{w}^{\text{Di}}=\text{argmin}_{\mathbf{w}\in\mathbb{R}^d}\frac{1}{N}\sum_{k=1}^{N}f_k(\mathbf{w})=\left(\sum_{k=1}^{N}\Xi_k^\top\Xi_k\right)^{-1}\sum_{k=1}^{N}\Xi_k^\top\Xi_k\hat{\mathbf{w}}_k.$ The solution of the local model is defined as $\mathbf{x}_k^{\text{Di}}=\text{argmin}_{\mathbf{x}_k\in\mathbb{R}^d}\left\{f_k(\mathbf{x}_k)+\frac{\lambda}{2}\left\|\mathbf{w}^{\text{Di}}-\mathbf{x}_k\right\|_2^2\right\}=(\Xi_k^\top\Xi_k/n+\lambda\boldsymbol{I}_d)^{-1}(\Xi_k^\top\mathbf{y}_k/n+\lambda\mathbf{w}^{\text{Di}})=(\Xi_k^\top\Xi_k/n+\lambda\boldsymbol{I}_d)^{-1}(\Xi_k^\top\Xi_k\hat{\mathbf{w}}_k/n+\lambda\mathbf{w}^{\text{Di}}).$

lp-proj-2    For our method `lp-proj-2`, the optimization problem is $\min_{\tilde{\mathbf{w}}\in\mathbb{R}^{d_{\text{sub}}}}F(\tilde{\mathbf{w}})=\frac{1}{N}\sum_{k=1}^{N}F_k(\tilde{\mathbf{w}})$ where $F_k(\tilde{\mathbf{w}})=\min_{\mathbf{x}_k\in\mathbb{R}^d}\{f_k(\mathbf{x}_k)+\frac{\lambda}{2}\left\|\tilde{\mathbf{w}}-\boldsymbol{P}\mathbf{x}_k\right\|_2^2\}.$ The solution of the global model is defined as $\tilde{\mathbf{w}}^{\text{l2}}=\text{argmin}_{\tilde{\mathbf{w}}\in\mathbb{R}^{d_{\text{sub}}}}\frac{1}{N}\sum_{k=1}^{N}F_k(\tilde{w})$ and the solution of the local solution is defined as $\mathbf{x}_k^{\text{l2}}=\text{argmin}_{\mathbf{x}\in\mathbb{R}^d}\left\{f_k(\mathbf{x}_k)+\frac{\lambda}{2}\left\|\tilde{\mathbf{w}}^{\text{l2}}-\boldsymbol{P}\mathbf{x}_k\right\|_2^2\right\}.$ Let $\check{\mathbf{x}}(\tilde{\mathbf{w}}):=\text{argmin}_{\mathbf{x}_k\in\mathbb{R}^d}\left\{f_k(\mathbf{x}_k)+\frac{\lambda}{2}\left\|\tilde{\mathbf{w}}-\boldsymbol{P}\mathbf{x}_k\right\|_2^2\right\}.$ It is easy to check $\check{\mathbf{x}}(\tilde{\mathbf{w}})=(\Xi_k^\top\Xi_k/n+\lambda\boldsymbol{P}^\top\boldsymbol{P})^{-1}(\Xi_k^\top\mathbf{y}_k/n+\lambda\boldsymbol{P}^\top\tilde{\mathbf{w}}).$ It follows that

$$F_k(\tilde{\mathbf{w}})=f_k(\check{\mathbf{x}}_k(\tilde{\mathbf{w}}))+\frac{\lambda}{2}\left\|\tilde{\mathbf{w}}-\check{\mathbf{x}}_k(\tilde{\mathbf{w}})\right\|_2^2$$

$$=-\frac{1}{2}\left(\frac{\Xi_k^\top\mathbf{y}_k}{n}+\lambda\boldsymbol{P}^\top\tilde{\mathbf{w}}\right)^\top\left(\frac{\Xi_k^\top\Xi_k}{n}+\lambda\boldsymbol{P}^\top\boldsymbol{P}\right)^{-1}\left(\frac{\Xi_k^\top\mathbf{y}_k}{n}+\lambda\boldsymbol{P}^\top\tilde{\mathbf{w}}\right)+\frac{\lambda}{2}\left\|\tilde{\mathbf{w}}\right\|_2^2+\frac{\left\|\mathbf{y}_k\right\|_2^2}{2n}$$

$$= \frac{\lambda}{2}\|\tilde{\mathbf{w}}\|_2^2 - \frac{\lambda^2}{2}\tilde{\mathbf{w}}^\top \boldsymbol{P}\left(\frac{\boldsymbol{\Xi}_k^\top \boldsymbol{\Xi}_k}{n} + \lambda \boldsymbol{P}^\top \boldsymbol{P}\right)^{-1}\boldsymbol{P}^\top \tilde{\mathbf{w}} - \lambda \tilde{\mathbf{w}}^\top \boldsymbol{P}\left(\frac{\boldsymbol{\Xi}_k^\top \boldsymbol{\Xi}_k}{n} + \lambda \boldsymbol{P}^\top \boldsymbol{P}\right)^{-1}\frac{\boldsymbol{\Xi}_k^\top \mathbf{y}_k}{n}$$

$$+ \frac{\|\mathbf{y}_k\|_2^2}{2n} - \frac{\mathbf{y}_k^\top \boldsymbol{\Xi}_k}{2n}\left(\frac{\boldsymbol{\Xi}_k^\top \boldsymbol{\Xi}_k}{n} + \lambda \boldsymbol{P}^\top \boldsymbol{P}\right)^{-1}\frac{\boldsymbol{\Xi}_k^\top \mathbf{y}_k}{n}.$$

Then we can obtain the expression of $F(\tilde{\mathbf{w}})$. However, for the general $\boldsymbol{\Xi}_k$, it is difficult to obtain a concise expression of the minimizer of $F(\tilde{\mathbf{w}})$. To make the calculations clean, we assume $\boldsymbol{\Xi}_k^\top \boldsymbol{\Xi}_k = nb_k \boldsymbol{I}_d$. Then the solutions of other methods can be simplified as

- `FedAvg`: $\mathbf{w}^{\text{Avg}} = \frac{\sum_{k=1}^N b_k \hat{\mathbf{w}}_k}{\sum_{k=1}^N b_k}$.

- `pFedMe`: $\mathbf{w}^{\text{Me}} = \frac{\sum_{k=1}^N b_k \hat{\mathbf{w}}_k/(b_k+\lambda)}{\sum_{k=1}^N b_k/(b_k+\lambda)}$ and $\mathbf{x}_k^{\text{Me}} = \frac{b_k \hat{\mathbf{w}}_k + \lambda \mathbf{w}^{\text{Me}}}{b_k+\lambda}$.

- `Ditto`: $\mathbf{w}^{\text{Di}} = \frac{\sum_{k=1}^N b_k \hat{\mathbf{w}}_k}{\sum_{k=1}^N b_k}$ and $\mathbf{x}_k^{\text{Di}} = \frac{b_k \hat{\mathbf{w}}_k + \lambda \mathbf{w}^{\text{Di}}}{b_k+\lambda}$.

Meanwhile, for `lp-proj-2`, without loss of generality, we can assume $\boldsymbol{P} = \boldsymbol{P}_0 := (\mathbf{e}_1, \mathbf{e}_2, \ldots, \mathbf{e}_{d_{\text{sub}}})^\top$, where $\mathbf{e}_i$ is the unit vector in $\mathbb{R}^d$ with the $i$-th element equal to $1$ and other elements equal to $0$. Otherwise, we can find a orthogonal matrix $\boldsymbol{Q}$ such that $\boldsymbol{P} = \boldsymbol{P}_0 \boldsymbol{Q}$. Then we have

$$f_k(\mathbf{x}_k) + \frac{\lambda}{2}\|\tilde{\mathbf{w}} - \boldsymbol{P}\mathbf{x}_k\|_2^2 = \frac{1}{2n}\|\boldsymbol{\Xi}_k \mathbf{x}_k - \mathbf{y}_k\|_2^2 + \frac{\lambda}{2}\|\tilde{\mathbf{w}} - \boldsymbol{P}\mathbf{x}_k\|_2^2$$

$$= \frac{1}{2n}\|\boldsymbol{\Xi}_k \boldsymbol{Q}^\top \boldsymbol{Q}\mathbf{x}_k - \mathbf{y}_k\|_2^2 + \frac{\lambda}{2}\|\tilde{\mathbf{w}} - \boldsymbol{P}_0 \boldsymbol{Q}\mathbf{x}_k\|_2^2$$

$$= \frac{1}{2n}\left\|\tilde{\boldsymbol{\Xi}}_k \tilde{\mathbf{x}}_k - \mathbf{y}_k\right\|_2^2 + \frac{\lambda}{2}\|\tilde{\mathbf{w}} - \boldsymbol{P}_0 \tilde{\mathbf{x}}_k\|_2^2$$

where $\tilde{\mathbf{x}}_k = \boldsymbol{Q}\mathbf{x}_k$ and $\tilde{\boldsymbol{\Xi}}_k = \boldsymbol{\Xi}_k \boldsymbol{Q}^\top$. Note that $\boldsymbol{\Xi}_k^\top \boldsymbol{\Xi}_k = nb_k \boldsymbol{I}_d$ implies $\tilde{\boldsymbol{\Xi}}_k^\top \tilde{\boldsymbol{\Xi}}_k = \boldsymbol{Q}\boldsymbol{\Xi}_k^\top \boldsymbol{\Xi}_k \boldsymbol{Q}^\top = nb_k \boldsymbol{I}_d$. After reparametrization, we return to the special case $\boldsymbol{P} = \boldsymbol{P}_0$.

Now we have

$$F_k(\tilde{\mathbf{w}}) = \frac{\lambda b_k}{2(b_k + \lambda)}\|\tilde{\mathbf{w}}\|_2^2 - \frac{\lambda}{(b_k+\lambda)n}\tilde{\mathbf{w}}^\top \boldsymbol{P}_0 \boldsymbol{\Xi}_k^\top \mathbf{y}_k + \frac{\|\mathbf{y}_k\|_2^2}{2n} - \frac{\mathbf{y}_k^\top \boldsymbol{\Xi}_k}{2n}\left(\frac{\boldsymbol{\Xi}_k^\top \boldsymbol{\Xi}_k}{n} + \lambda \boldsymbol{P}_0^\top \boldsymbol{P}_0\right)^{-1}\frac{\boldsymbol{\Xi}_k^\top \mathbf{y}_k}{n}$$

$$= \frac{\lambda b_k}{2(b_k + \lambda)}\|\tilde{\mathbf{w}}\|_2^2 - \frac{\lambda b_k}{b_k+\lambda}\tilde{\mathbf{w}}^\top \hat{\mathbf{w}}_{k,1} + \frac{\|\mathbf{y}_k\|_2^2}{2n} - \frac{\mathbf{y}_k^\top \boldsymbol{\Xi}_k}{2n}\left(\frac{\boldsymbol{\Xi}_k^\top \boldsymbol{\Xi}_k}{n} + \lambda \boldsymbol{P}_0^\top \boldsymbol{P}_0\right)^{-1}\frac{\boldsymbol{\Xi}_k^\top \mathbf{y}_k}{n},$$

where $\hat{\mathbf{w}}_{k,1} = \boldsymbol{P}_0 \hat{\mathbf{w}}_k$ is the first $d_{\text{sub}}$ elements of $\hat{\mathbf{w}}_k$. Then we obtain

$$F_k(\tilde{\mathbf{w}}) = \frac{1}{N}\sum_{k=1}^N \frac{\lambda b_k}{2(b_k + \lambda)}\|\tilde{\mathbf{w}}\|_2^2 - \frac{1}{N}\sum_{k=1}^N \frac{\lambda b_k}{b_k + \lambda}\tilde{\mathbf{w}}^\top \hat{\mathbf{w}}_{k,1} + C_1,$$

where $C_1$ is a constant. Thus the solution of the global model is $\tilde{\mathbf{w}}^{\text{l2}} = \frac{\sum_{k=1}^N b_k \hat{\mathbf{w}}_{k,1}/(b_k+\lambda)}{\sum_{k=1}^N b_k/(b_k+\lambda)}$, and the solution of the local model is $\mathbf{x}_k^{\text{l2}} = \check{\mathbf{x}}_k(\tilde{\mathbf{w}}^{\text{l2}}) = \begin{pmatrix}(b_k \hat{\mathbf{w}}_{k,1} + \lambda \tilde{\mathbf{w}}^{\text{l2}})/(b_k + \lambda) \\ \hat{\mathbf{w}}_{k,2}\end{pmatrix}$, where $\hat{\mathbf{w}}_{k,2}$ is the last $d - d_{\text{sub}}$ elements of $\hat{\mathbf{w}}_k$.

To summarize, the solutions of different models are listed as follows.

- `local`: $\mathbf{w}_k^{\text{loc}} = \hat{\mathbf{w}}_k$.

- `FedAvg`: $\mathbf{w}^{\text{Avg}} = \frac{\sum_{k=1}^N b_k \hat{\mathbf{w}}_k}{\sum_{k=1}^N b_k}$.

- `pFedMe`: $\mathbf{w}^{\text{Me}} = \frac{\sum_{k=1}^N b_k \hat{\mathbf{w}}_k/(b_k+\lambda)}{\sum_{k=1}^N b_k/(b_k+\lambda)}$ and $\mathbf{x}_k^{\text{Me}} = \frac{b_k \hat{\mathbf{w}}_k + \lambda \mathbf{w}^{\text{Me}}}{b_k+\lambda}$.

- `Ditto`: $\mathbf{w}^{\text{Di}} = \frac{\sum_{k=1}^N b_k \hat{\mathbf{w}}_k}{\sum_{k=1}^N b_k}$ and $\mathbf{x}_k^{\text{Di}} = \frac{b_k \hat{\mathbf{w}}_k + \lambda \mathbf{w}^{\text{Di}}}{b_k+\lambda}$.

- `lp-proj-2`: $\quad \tilde{\mathbf{w}}^{\mathrm{l2}} \quad = \quad \frac{\sum_{k=1}^{N} b_k \hat{\mathbf{w}}_{k,1}/(b_k+\lambda)}{\sum_{k=1}^{N} b_k/(b_k+\lambda)}$ and $\mathbf{x}_k^{\mathrm{l2}} \quad = \quad \check{\mathbf{x}}_k(\tilde{\mathbf{w}}^{\mathrm{l2}}) \quad =$
$\begin{pmatrix} (b_k \hat{\mathbf{w}}_{k,1} + \lambda \tilde{\mathbf{w}}^{\mathrm{l2}})/(b_k+\lambda) \\ \hat{\mathbf{w}}_{k,2} \end{pmatrix}.$

Note that $\mathbf{x}_k^{\mathrm{Me}}$ and $\mathbf{x}_k^{\mathrm{Di}}$ are both the weighted average of $\mathbf{w}^{\mathrm{Me}}/\mathbf{w}^{\mathrm{Di}}$ and $\hat{\mathbf{w}}_k$ with the same weight. $\mathbf{w}^{\mathrm{Me}}$ and $\mathbf{w}^{\mathrm{Di}}$ are weighted average of $\hat{\mathbf{w}}_k$ with different weights. If $\lambda = 0$, we have $\mathbf{w}^{\mathrm{Me}} = \frac{1}{N} \sum_{k=1}^{N} \hat{\mathbf{w}}_k$. If $\lambda \to \infty$, we have $\hat{\mathbf{w}}^{\mathrm{Me}} \to \mathbf{w}^{\mathrm{Avg}}$. Thus, the weight of pFedMe is more uniform than that of FedAvg. In Section 4.2, we assume $b_k = b$. This is reasonable since we often normalize the data. Then we have $\mathbf{w}^{\mathrm{Avg}} = \mathbf{w}^{\mathrm{Me}} = \mathbf{w}^{\mathrm{Di}} = \frac{1}{N} \sum_{k=1}^{N} \hat{\mathbf{w}}_k$ and $\mathbf{x}_k^{\mathrm{Me}} = \mathbf{x}_k^{\mathrm{Di}}$.

Moreover, `lp-proj-2` can be viewed as a interpolation of local and pFedMe. The first $d_{\mathrm{sub}}$ dimensions of $\mathbf{x}_k^{\mathrm{l2}}$ equal to those of $\mathbf{x}_k^{\mathrm{Me}}$ and the last $d - d_{\mathrm{sub}}$ dimensions equal to those of $\mathbf{w}_k^{\mathrm{loc}}$.

## B.2 Test Loss

In this subsection, we compute the test losses of different methods. From now on, we always assume $b_k = b$ to make calculations clean.

Recall that the dataset on client $k$ is $(\mathbf{\Xi}_k, \mathbf{y}_k)$, where $\mathbf{\Xi}_k$ is fixed and $\mathbf{y}_k$ follows Gaussian distribution $\mathcal{N}(\mathbf{\Xi}_k \mathbf{w}_k, \sigma^2 \mathbf{I}_n)$. Then the data heterogeneity across clients only lies in the heterogeneity of $\mathbf{w}_k$. We can obtain the distribution of the solutions of different methods.

Let $\bar{\mathbf{w}} = \frac{\sum_{k=1}^{N} \mathbf{w}_k}{N}$. We have

- `local`: $\mathbf{w}_k^{\mathrm{loc}} \sim \mathcal{N}\left(\mathbf{w}_k, \frac{\sigma^2}{bn} \mathbf{I}_d\right)$.

- `FedAvg`: $\mathbf{w}^{\mathrm{Avg}} \sim \mathcal{N}\left(\bar{\mathbf{w}}, \frac{\sigma^2}{bNn} \mathbf{I}_d\right)$.

- `pFedMe`: $\mathbf{w}^{\mathrm{Me}} \sim \mathcal{N}\left(\bar{\mathbf{w}}, \frac{\sigma^2}{bNn} \mathbf{I}_d\right)$ and $\mathbf{x}_k^{\mathrm{Me}} \sim \mathcal{N}\left(\frac{b\mathbf{w}_k + \lambda \bar{\mathbf{w}}}{b+\lambda}, \frac{\left(b^2 + \frac{2b\lambda}{N}\right) \frac{\sigma^2}{bn} + \frac{\lambda^2}{N} \cdot \frac{\sigma^2}{bn}}{(b_k+\lambda)^2} \mathbf{I}_d\right)$.

- `Ditto`: $\mathbf{w}^{\mathrm{Di}} = \mathbf{w}^{\mathrm{Me}}$ and $\mathbf{x}_k^{\mathrm{Di}} = \mathbf{x}_k^{\mathrm{Me}}$.

- `lp-proj-2`: $\tilde{\mathbf{w}}^{\mathrm{l2}} \sim \mathcal{N}\left(\bar{\mathbf{w}}_{\cdot,1}, \frac{\sigma^2}{bNn} \mathbf{I}_{d_{\mathrm{sub}}}\right)$ and

$$\mathbf{x}_k^{\mathrm{l2}} \sim \mathcal{N}\left(\begin{pmatrix} \frac{b\mathbf{w}_{k,1} + \lambda \bar{\mathbf{w}}_{\cdot,1}}{b_k+\lambda} \\ \mathbf{w}_{k,2} \end{pmatrix}, \begin{pmatrix} \frac{\left(b^2 + \frac{2b\lambda}{N}\right) \frac{\sigma^2}{bn} + \frac{\lambda^2}{N^2} \cdot \frac{\sigma^2}{bn}}{(b_k+\lambda)^2} \mathbf{I}_{d_{\mathrm{sub}}} & \\ & \frac{\sigma^2}{bn} \mathbf{I}_{d-d_{\mathrm{sub}}} \end{pmatrix}\right)$$

where $\mathbf{w}_{k,1}$ is the first $d$ elements of $\mathbf{w}_k$, $\mathbf{w}_{k,2}$ is the last $d - d_{\mathrm{sub}}$ elements of $\mathbf{w}_k$ and $\bar{\mathbf{w}}_{\cdot,1}$ is the first $k$ elements of $\bar{\mathbf{w}}$.

Since $\mathbf{\Xi}_k$ is fixed, we assume the test data is $(\mathbf{\Xi}_k, \mathbf{y}_k')$ where $\mathbf{y}_k' = \mathbf{\Xi}_k \mathbf{w}_k + \mathbf{z}_k'$ with $\mathbf{z}_k' \sim \mathcal{N}(\mathbf{0}_n, \sigma^2 \mathbf{I}_n)$ independent of $\mathbf{z}_k$. Then the test loss on client $k$ is defined as

$$\begin{aligned} f_k^{\mathrm{te}}(\mathbf{x}_k) &= \frac{1}{2n} \mathbb{E} \left\| \mathbf{\Xi}_k \mathbf{x}_k - \mathbf{y}_k' \right\|_2^2 \\ &= \frac{1}{2n} \mathbb{E} \left\| \mathbf{\Xi}_k \mathbf{x}_k - (\mathbf{\Xi}_k \mathbf{w}_k + \mathbf{z}_k') \right\|_2^2 \\ &= \frac{\sigma^2}{2} + \frac{1}{2n} \mathbb{E} \left\| \mathbf{\Xi}_k (\mathbf{x}_k - \mathbf{w}_k) \right\|_2^2 \\ &= \frac{\sigma^2}{2} + \frac{b}{2} \mathbb{E} \left\| \mathbf{x}_k - \mathbf{w}_k \right\|_2^2 \\ &= \frac{\sigma^2}{2} + \frac{b}{2} \mathrm{tr}(\mathrm{var}(\mathbf{x}_k)) + \frac{b}{2} \left\| \mathbb{E} \mathbf{x}_k - \mathbf{w}_k \right\|_2^2. \end{aligned} \quad (23)$$

and the averaged test loss is

$$\frac{1}{N} \sum_{k=1}^{N} f_k^{\mathrm{te}}(\mathbf{x}_k) = \frac{\sigma^2}{2} + \frac{b}{2N} \sum_{k=1}^{N} \mathrm{tr}(\mathrm{var}(\mathbf{x}_k)) + \frac{b}{2N} \sum_{k=1}^{N} \left\| \mathbb{E} \mathbf{x}_k - \mathbf{w}_k \right\|_2^2.$$

Then we can compute the test losses for different methods. Since the solutions of `Ditto` and `pFedMe` are the same, we omit the analysis for `Ditto`. We have

$$L^{\text{loc}} = \frac{1}{N}\sum_{k=1}^{N} f_k^{\text{te}}(\mathbf{w}_k^{\text{loc}}) = \frac{\sigma^2}{2} + \frac{\sigma^2 d}{2n},$$

$$L^{\text{Avg}} = \frac{1}{N}\sum_{k=1}^{N} f_k^{\text{te}}(\mathbf{w}^{\text{Avg}}) = \frac{\sigma^2}{2} + \frac{\sigma^2 d}{2Nn} + \frac{b}{2N}\sum_{k=1}^{N}\|\bar{\mathbf{w}} - \mathbf{w}_k\|_2^2,$$

$$L^{\text{Me}}(\lambda) = \frac{1}{N}\sum_{k=1}^{N} f_k^{\text{te}}(\mathbf{x}_k^{\text{Me}}) = \frac{\sigma^2}{2} + \frac{b^2 + \frac{2b\lambda}{N} + \frac{\lambda^2}{N}}{(b+\lambda)^2}\cdot\frac{\sigma^2 d}{2n} + \frac{b\lambda^2}{2N(b+\lambda)^2}\sum_{k=1}^{N}\|\bar{\mathbf{w}} - \mathbf{w}_k\|_2^2,$$

$$L^{\text{l2}}(\lambda) = \frac{1}{N}\sum_{k=1}^{N} f_k^{\text{te}}(\mathbf{x}_k^{\text{l2}}) = \frac{\sigma^2}{2} + \frac{b^2 + \frac{2b\lambda}{N} + \frac{\lambda^2}{N}}{(b+\lambda)^2}\cdot\frac{\sigma^2 d_{\text{sub}}}{2n} + \frac{\sigma^2(d - d_{\text{sub}})}{2n} + \frac{b\lambda^2}{2N(b+\lambda)^2}\sum_{k=1}^{N}\|\bar{\mathbf{w}}_{\cdot,1} - \mathbf{w}_{k,1}\|_2^2.$$

Note that the test losses for `pFedMe` and `lp-proj-2` are functions of $\lambda$.

One can check that the optimal $\lambda$ for `pFedMe` is $\lambda^{\text{Me}} = \frac{(1-1/N)\sigma^2 d/n}{\sum_{k=1}^{N}\|\bar{\mathbf{w}} - \mathbf{w}_k\|_2^2/N}$ and the optimal $\lambda$ for `lp-proj-2` is $\lambda^{\text{l2}} = \frac{(1-1/N)\sigma^2 d_{\text{sub}}/n}{\sum_{k=1}^{N}\|\bar{\mathbf{w}}_{\cdot,1} - \mathbf{w}_{k,1}\|_2^2/N}$. Then we can compute the minimal test losses for `pFedMe` and `lp-proj-2` as follows.

$$L_*^{\text{Me}} = L^{\text{Me}}(\lambda^{\text{Me}}) = \frac{\sigma^2}{2} + \frac{\sigma^2 d}{2n}\cdot\frac{(\lambda^{\text{Me}})^2/N + (1+1/N)b\lambda^{\text{Me}} + b^2}{(b+\lambda^{\text{Me}})^2},$$

$$L_*^{\text{l2}} = L^{\text{l2}}(\lambda^{\text{l2}}) = \frac{\sigma^2}{2} + \frac{\sigma^2 d_{\text{sub}}}{2n}\cdot\frac{(\lambda^{\text{l2}})^2/N + (1+1/N)b\lambda^{\text{l2}} + b^2}{(b+\lambda^{\text{l2}})^2} + \frac{\sigma^2(d - d_{\text{sub}})}{2n}.$$

However, it is not easy the compare these losses directly. We further assume the heterogeneity in terms of $\mathbf{w}_k$ is uniform in all dimensions, that is

$$\frac{1}{dN}\sum_{k=1}^{N}\|\bar{\mathbf{w}} - \mathbf{w}_k\|_2^2 = \frac{1}{d_{\text{sub}}N}\sum_{k=1}^{N}\|\bar{\mathbf{w}}_{\cdot,1} - \mathbf{w}_{k,1}\|_2^2 := \Sigma. \tag{24}$$

Then $\lambda^{\text{Me}}$ and $\lambda^{\text{l2}}$ coincide and are equal to $\lambda^* := (1 - 1/N)\sigma^2/(n\Sigma)$. Note that $\sigma^2$ is the variance of the observation noises on different clients, and $n$ is the number of samples on each client. Thus $\lambda^*$ can reflect the relative magnitude of the variance and the heterogeneity .

With the uniform heterogeneity, we have

$$L^{\text{loc}} = \frac{\sigma^2}{2} + \frac{\sigma^2 d}{2n(b+\lambda^*)^2}\left[(\lambda^*)^2 + 2b\lambda^* + b^2\right],$$

$$L^{\text{Avg}} = \frac{\sigma^2}{2} + \frac{\sigma^2 d}{2n(b+\lambda^*)^2}\left[\frac{(\lambda^*)^2}{N} + \frac{N+1}{N}b\lambda^* + \frac{2N-1}{N}b^2 + \frac{N-1}{N}\frac{b^3}{\lambda^*}\right],$$

$$L_*^{\text{Me}} = \frac{\sigma^2}{2} + \frac{\sigma^2 d}{2n(b+\lambda^*)^2}\left[\frac{(\lambda^*)^2}{N} + \frac{N+1}{N}b\lambda^* + b^2\right],$$

$$L_*^{\text{l2}} = \frac{\sigma^2}{2} + \frac{\sigma^2 d_{\text{sub}}}{2n(b+\lambda^*)^2}\left[\frac{(\lambda^*)^2}{N} + \frac{N+1}{N}b\lambda^* + b^2\right] + \frac{\sigma^2(d - d_{\text{sub}})}{2n(b+\lambda^*)^2}\left[(\lambda^*)^2 + 2b\lambda^* + b^2\right].$$

Comparing their (optimal) losses, we can obtain the following observations.

- $L^{\text{loc}} \geq L_*^{\text{l2}} \geq L_*^{\text{Me}}$ and $L^{\text{Avg}} \geq L_*^{\text{Me}}$. This means that `pFedMe` with the optimal $\lambda$ always has the minimal loss. Moreover, since `lp-proj-2` can be regarded as an interpolation of `local` and `pFedMe`, $L_*^{\text{l2}}$ is also a interpolation of $L^{\text{loc}}$ and $L_*^{\text{Me}}$.

- $L^{\text{loc}} \leq L^{\text{Avg}}$ if and only if $\lambda^* \leq b$. This means that if the heterogeneity or the number of local data is sufficiently large, then `local` is better than `FedAvg`.

- $L_*^{\text{l2}} \leq L^{\text{Avg}}$ if and only if $\lambda^* \leq \sqrt{\frac{d}{d - d_{\text{sub}}}}b$. The range of $\lambda^*$ over which `lp-proj-2` is better than `FedAvg` is slightly larger than the range of that over which `local` is better than `FedAvg`.

- Fix $\sigma^2$ and $n$ and let $\Sigma \to \infty$. Then we have $\lambda^* \to 0$, $\lim_{\lambda^* \to 0} L^{\mathrm{loc}} = \lim_{\lambda^* \to 0} L_*^{\mathrm{Me}} = \lim_{\lambda^* \to 0} L_*^{\mathrm{l2}}$ and $\lim_{\lambda^* \to 0} L^{\mathrm{Avg}} = \infty$. This implies that if the heterogeneity is sufficiently large, the optimal lambda is nearly $0$ and there is little difference between `local`, `pFedMe` and `lp-proj-2`. And the loss of `FedAvg` is large. So there is no need for federated learning.

Up to now, we have only focused on the optimal value of $\lambda$. However, in practice, we can hardly know this value. Thus we need to compare these losses under different values of $\lambda$. With (24) holding, we have the following results.

- $L^{\mathrm{loc}} \le L^{\mathrm{Avg}}$ if and only if $\Sigma \ge \frac{N-1}{N} \frac{\sigma^2}{bn}$ ($\lambda^* \le b$).

- $L^{\mathrm{loc}} \le L^{\mathrm{l2}}(\lambda) \le L^{\mathrm{Me}}(\lambda)$ if and only if $\Sigma \ge \frac{N-1}{N} \frac{2b+\lambda}{\lambda} \frac{\sigma^2}{bn}$. If $\Sigma > \frac{N-1}{N} \frac{\sigma^2}{bn}$ ($\lambda^* < b$), this is equivalent to $\lambda \ge \frac{2\lambda^*}{1-\lambda^*/b}$.

- $L^{\mathrm{Me}}(\lambda) \le L^{\mathrm{Avg}}$ if and only if $\Sigma \ge \frac{N-1}{N} \frac{\sigma^2}{(b+2\lambda)n}$. This is equivalent to $\lambda \ge \frac{\lambda^*-b}{2}$.

- $L^{\mathrm{l2}}(\lambda) \le L^{\mathrm{Avg}}$ if and only if $\Sigma \ge \frac{N-1}{N} \frac{\sigma^2}{bn} \frac{d(b+\lambda)^2 - d_{\mathrm{sub}}\lambda(2b+\lambda)}{d(b+\lambda)^2 - d_{\mathrm{sub}}\lambda^2}$. About $\frac{d(b+\lambda)^2 - d_{\mathrm{sub}}\lambda(2b+\lambda)}{d(b+\lambda)^2 - d_{\mathrm{sub}}\lambda^2}$, we have $\frac{1+\sqrt{\frac{d-d_{\mathrm{sub}}}{d}}}{1+\sqrt{\frac{d}{d-d_{\mathrm{sub}}}}} \le \frac{d(b+\lambda)^2 - d_{\mathrm{sub}}\lambda(2b+\lambda)}{d(b+\lambda)^2 - d_{\mathrm{sub}}\lambda^2} \le 1$. When $\lambda = 0$ or $\lambda \to \infty$, the fraction goes to 1. When $\lambda = \sqrt{\frac{d}{d-d_{\mathrm{sub}}}} b$, the fraction attains the minimal value.

Then we can sort these losses.

If the heterogeneity is small, i.e., $\Sigma < \frac{N-1}{N} \frac{\sigma^2}{bn} \frac{1+\sqrt{\frac{d-d_{\mathrm{sub}}}{d}}}{1+\sqrt{\frac{d}{d-d_{\mathrm{sub}}}}}$, then $\lambda^* > b$. When $\lambda < \frac{\lambda^*-b}{2}$, we have $L^{\mathrm{Avg}} \le L^{\mathrm{Me}}(\lambda) \le L^{\mathrm{l2}}(\lambda) \le L^{\mathrm{loc}}$; when $\lambda > \frac{\lambda^*-b}{2}$, we have $L^{\mathrm{Me}}(\lambda) \le L^{\mathrm{Avg}} \le L^{\mathrm{l2}}(\lambda) \le L^{\mathrm{loc}}$. In this case, `FedAvg` and `pFedMe` are always better than `lp-proj-2` and `local`. If $\lambda$ is larger than a threshold value, `pFedMe` is better than `FedAvg`.

If the heterogeneity is large, i.e., $\Sigma > \frac{N-1}{N} \frac{\sigma^2}{bn}$, then $\lambda^* < b$. When $\lambda \le \frac{2\lambda^*}{1-\lambda^*/b}$, we have $L^{\mathrm{Me}}(\lambda) \le L^{\mathrm{l2}}(\lambda) \le L^{\mathrm{loc}} \le L^{\mathrm{Avg}}$; when $\lambda > \frac{2\lambda^*}{1-\lambda^*/b}$, we have $L^{\mathrm{loc}} \le L^{\mathrm{l2}}(\lambda) \le L^{\mathrm{Me}}(\lambda) \le L^{\mathrm{Avg}}$. In this case, `FedAvg` is the worst method and `lp-proj-2` always lies between `local` and `pFedMe`.

## B.3  Robustness

In this subsection, we consider the robustness of different methods against Byzantine attacks. Recall that in the last subsection, we only consider the exact solution of these methods and ignore the process of the algorithms. In terms of robustness, we must take the procedures of different methods into account, especially the communication between the central server and local clients. Moreover, we focus on the simplified setting where the number of local update steps is infinite, there is only one round of communication and all clients participate in the communication.

As indicated in Section 4.2 , we examine three types of Byzantine attacks. Throughout this subsection, we suppose that there are $N_b$ benign clients and $N_a$ malicious clients with $N_a + N_b = N$, and let $I_b$ denote the indices of benign clients and $I_a$ denote the indices of malicious clients.

We will analyze how these attacks will affect the solution of different methods, and compare the averaged test losses on benign clients.

### B.3.1  The Simplified Setting

We first show that in our simplified setting, after one round of communication, all the methods will obtain their exact solutions defined in Appendix B.1.

`local`   The objective of the local client is $\min_{\mathbf{w} \in \mathbb{R}^d} f_k(\mathbf{w})$. If the number of local update steps is infinite, we will obtain the least square estimator $\hat{\mathbf{w}}_k = \mathbf{w}_k^{\mathrm{loc}}$. For the convergence of SGD, see Nemirovski et al. [53].

**FedAvg** Similar to `local`, the local client will obtain $\hat{\mathbf{w}}_k$ and sends it to the server. Then the server obtains $\frac{1}{N}\sum_{k=1}^N \hat{\mathbf{w}}_k = \mathbf{w}^{\text{Avg}}$ and broadcasts $\mathbf{w}^{\text{Avg}}$ to all the clients.

**pFedMe** pFedMe corresponds to `lp-proj-2` with $\boldsymbol{P} = \boldsymbol{I}_d$. The local update step is $\mathbf{w}_{k,r+1}^t = \mathbf{w}_{k,r} - \eta\lambda(\mathbf{w}_{k,r}^t - \mathbf{x}_{k,r}^t)$ where $\mathbf{x}_{k,r}^t = \hat{\mathbf{x}}_k(\mathbf{w}_{k,r}^t) = \operatorname{argmin}_{\mathbf{x}\in\mathbb{R}^d}\left\{ f_k(\mathbf{x}_k) + \frac{\lambda}{2}\left\|\mathbf{w}_{k,r}^t - \mathbf{x}_k\right\|_2^2\right\} = \frac{b\hat{\mathbf{w}}_k + \lambda\mathbf{w}_{k,r}}{b+\lambda}$. (When the number of local update steps is infinite, it is reasonable to assume that we can obtain the exact value of $\mathbf{x}_{k,r}^t$.) The local update rule can be rewritten as $\mathbf{w}_{k,r+1}^t = \mathbf{w}_{k,r} - \frac{\eta\lambda b}{b+\lambda}(\mathbf{w}_{k,r}^t - \hat{\mathbf{w}}_k)$, which can be regarded as a step of gradient descent with step size $\frac{\eta\lambda b}{b+\lambda}$ to minimize $\frac{1}{2}\|\mathbf{w} - \hat{\mathbf{w}}_k\|_2^2$. As long as the step size is not too large, we have $\lim_{R\to\infty}\mathbf{w}_{k,R}^t = \hat{\mathbf{w}}_k$. This means that if we do infinite steps of local update, the local version of global parameter is $\hat{\mathbf{w}}_k$. Then each client sends this local version to the server and the server obtains $\frac{1}{N}\sum_{k=1}^N \hat{\mathbf{w}}_k = \mathbf{w}^{\text{Me}}$. After that, the server broadcasts $\mathbf{w}^{\text{Me}}$ to all clients. Finally, the client $k$ solves $\min_{\mathbf{x}\in\mathbb{R}^d}\left\{ f_k(\mathbf{x}_k) + \frac{\lambda}{2}\left\|\mathbf{w}^{\text{Me}} - \mathbf{x}_k\right\|_2^2\right\}$ and obtains $\mathbf{x}_k^{\text{Me}} = \hat{\mathbf{x}}_k(\mathbf{w}^{\text{Me}})$.

**Ditto** The global model of `Ditto` is the same as the model of `FedAvg`. So the server will also obtain $\frac{1}{N}\sum_{k=1}^N \hat{\mathbf{w}}_k = \mathbf{w}^{\text{Di}}$. Then the server broadcasts $\mathbf{w}^{\text{Di}}$ to all the clients and the client $k$ solves $\min_{\mathbf{x}_k\in\mathbb{R}^d}\left\{ f_k(\mathbf{x}_k) + \frac{\lambda}{2}\left\|\mathbf{w}^{\text{Di}} - \mathbf{x}_k\right\|_2^2\right\}$ and gets $\mathbf{x}_k^{\text{Di}}$.

**lp-proj-2** For `lp-proj-2`, without loss of generality, we can still assume $\boldsymbol{P} = \boldsymbol{P}_0 := (\mathbf{e}_1, \mathbf{e}_2, \ldots, \mathbf{e}_{d_{\text{sub}}})$. The local update step is $\tilde{\mathbf{w}}_{k,r+1}^t = \check{\mathbf{x}}_k(\tilde{\mathbf{w}}_{k,r}^t) = \tilde{\mathbf{w}}_{k,r}^t - \eta\lambda(\tilde{\mathbf{w}}_{k,r}^t - \boldsymbol{P}\mathbf{x}_{k,r}^t) = \tilde{\mathbf{w}}_{k,r}^t - \frac{\eta\lambda b}{b+\lambda}(\tilde{\mathbf{w}}_{k,r}^t - \hat{\mathbf{w}}_{k,1})$, where $\mathbf{x}_{k,r}^t = \operatorname{argmin}_{\mathbf{x}_k\in\mathbb{R}^d}\left\{ f_k(\mathbf{x}_k) + \frac{\lambda}{2}\left\|\tilde{\mathbf{w}}_{k,r}^t - \boldsymbol{P}_0\mathbf{x}_k\right\|_2^2\right\} = \begin{pmatrix}(b\hat{\mathbf{w}}_{k,1} + \lambda\tilde{\mathbf{w}}_{k,r}^t)/(b+\lambda) \\ \hat{\mathbf{w}}_{k,2}\end{pmatrix}$. Similar to `pFedMe`, the local update step can be regarded as a step pf gradient descent with step size $\frac{\eta\lambda b}{b+\lambda}$ to minimize $\frac{1}{2}\|\tilde{\mathbf{w}} - \hat{\mathbf{w}}_{k,1}\|_2^2$. As long as the step size is not too large, we have $\lim_{R\to\infty}\tilde{\mathbf{w}}_{k,R}^t = \hat{\mathbf{w}}_{k,1}$. After the communication, the server gets $\frac{1}{N}\sum_{k=1}^N \hat{\mathbf{w}}_{k,1} = \tilde{\mathbf{w}}^{\text{l2}}$ and the client $k$ obtains $\mathbf{x}_k^{\text{l2}} = \check{\mathbf{x}}_k(\tilde{\mathbf{w}}^{\text{l2}})$.

### B.3.2 Same-value Attacks

Now we focus on the same-value attacks.

**local** For pure local training, there is no communication between the central server and local clients. So the averaged test loss on benign clients is $L^{\text{loc, att1}} = \frac{1}{N_b}\sum_{k\in I_b} f_k^{\text{te}}(\mathbf{w}_k^{\text{loc}}) = \frac{\sigma^2}{2} + \frac{\sigma^2 d}{2n}$.

**FedAvg** For `FedAvg`, the local problem $\min_{\mathbf{w}\in\mathbb{R}^d} f_k(\mathbf{w})$ remains unchanged, no matter what the server sends to the local client. As long as the number of local update steps goes to $\infty$, the local parameter will go to the least square estimator $\hat{\mathbf{w}}_k$.

If the $k$-th client is benign, it will send $\hat{\mathbf{w}}_k$ to the server. Recall that $\hat{\mathbf{w}}_k = (\boldsymbol{\Xi}_k\boldsymbol{\Xi}_k)^{-1}\boldsymbol{\Xi}_k\mathbf{y}_k \sim \mathcal{N}(\mathbf{w}_k, \sigma^2\boldsymbol{I}_d/(bn))$. This means that $\hat{\mathbf{w}}_k$ can be viewed as an unbiased observation of $\mathbf{w}_k$ with covariance matrix $\frac{\sigma^2}{bn}\boldsymbol{I}_d$.

If the $k$-th client is malicious, it will send $\mathbf{w}_k^{(ma)} = c\boldsymbol{I}_d$ to the server with $c\sim\mathcal{N}(0,\tau^2)$. Then $\mathbf{w}_k^{(ma)}$ is an unbiased observation of $\mathbf{0}_m$ with covariance matrix $\tau^2\boldsymbol{J}_d$ where $\boldsymbol{J}_d = \begin{pmatrix} 1 & 1 & \cdots & 1 \\ 1 & 1 & \cdots & 1 \\ \vdots & \vdots & \ddots & \vdots \\ 1 & 1 & \cdots & 1 \end{pmatrix} \in$ $\mathbb{R}^{d\times d}$. In this case, the number of local update steps will not affect the messages transferred by the malicious client. Then the server obtains $\mathbf{w}^{\text{Avg,att1}} = \frac{1}{N}\left(\sum_{k\in I_b}\hat{\mathbf{w}}_k + \sum_{k\in I_a}\mathbf{w}_k^{(ma)}\right)$. We have

$\mathbf{w}^{\text{Avg,att1}} \sim \mathcal{N}\left(\frac{1}{N}\sum_{k\in I_b}\mathbf{w}_k, \frac{1}{N^2}\left(\frac{N_b\sigma^2}{bn}\boldsymbol{I}_d + N_a\tau^2\boldsymbol{J}_d\right)\right)$. Then we can compute the averaged test loss on benign clients as

$$L^{\text{Avg,att1}} = \frac{1}{N_b}\sum_{k\in I_b}f_k^{\text{te}}(\mathbf{w}^{\text{Avg,att1}}) = \frac{\sigma^2}{2} + \frac{bd}{2N^2}\left(\frac{N_b\sigma^2}{bn} + N_a\tau^2\right) + \frac{b}{2N}\sum_{k=1}^{N}\left\|\frac{\sum_{i\in I_b}\mathbf{w}_i}{N} - \mathbf{w}_k\right\|_2^2.$$

pFedMe    Similar to FedAvg, the attack will not influence the minimization of the local model. If the $k$-th client is benign, it sends $\hat{\mathbf{w}}_k$ to the server. If the $k$-th client is malicious, it sends $\mathbf{w}_k^{(ma)} = c\mathbf{1}_d$ to the server with $c \sim \mathcal{N}(0,\tau^2)$. The server obtains $\mathbf{w}^{\text{Me,att1}} = \frac{1}{N}\left(\sum_{k\in I_b}\hat{\mathbf{w}}_k + \sum_{k\in I_a}\mathbf{w}_k^{(ma)}\right) = \mathbf{w}^{\text{Avg, att1}}$.

Then the server broadcasts $\mathbf{w}^{\text{Me,att1}}$ to all the clients. And the benign client $k$ compute the local parameter $\mathbf{x}_k^{\text{Me,att1}} = \hat{\mathbf{x}}_k(\mathbf{w}^{\text{Me,att1}}) = \frac{b\hat{\mathbf{w}}_k + \lambda\mathbf{w}^{\text{Me,att1}}}{b+\lambda}$. We have

$$\mathbf{x}_k^{\text{Me,att1}} \sim \mathcal{N}\left(\frac{b\mathbf{w}_k + \lambda\sum_{i\in I_b}\mathbf{w}_i/N}{b+\lambda}, \frac{\left[\left(b+\frac{\lambda}{N}\right)^2\frac{\sigma^2}{bn} + (N_b-1)\frac{\lambda^2}{N^2}\frac{\sigma^2}{bn}\right]\boldsymbol{I}_d + N_a\frac{\lambda^2}{N^2}\tau^2\boldsymbol{J}_d}{(b+\lambda)^2}\right).$$

Then we can compute the averaged loss on benign clients as

$$L^{\text{Me,att1}}(\lambda) = \frac{1}{N_b}\sum_{k\in I_b}f_k^{\text{te}}(\mathbf{x}_k^{\text{Avg,att1}})$$

$$= \frac{\sigma^2}{2} + \frac{bd}{2}\cdot\frac{\left(b^2 + \frac{2b\lambda}{N} + \frac{N_b\lambda^2}{N^2}\right)\frac{\sigma^2}{bn} + \frac{N_a\lambda^2}{N^2}\tau^2}{(b+\lambda)^2} + \frac{b\lambda^2}{2(b+\lambda)^2}\cdot\frac{1}{N_b}\sum_{i\in I_b}\left\|\frac{\sum_{i\in I_b}\mathbf{w}_i}{N} - \mathbf{w}_k\right\|_2^2.$$

And one can check the optimal $\lambda$ is $\lambda^{\text{Me, att1}} = \dfrac{(1-1/N)\sigma^2 d/n}{\frac{1}{N_b}\sum_{k\in I_b}\left\|\frac{\sum_{i\in I_b}\mathbf{w}_i}{N} - \mathbf{w}_k\right\|_2^2 + \frac{dN_a}{N^2}(\tau^2 - \sigma^2/(bn))}$.

Ditto    Since the global model of Ditto is the same as the model of FedAvg, we have $\mathbf{w}^{\text{Di, att1}} = \mathbf{w}^{\text{Avg, att1}}$. Then the server broadcasts $\mathbf{w}^{\text{Di, att1}}$ to all the clients and the benign client $k$ obtains $\mathbf{x}_k^{\text{Di, att1}} = \hat{\mathbf{x}}_k(\mathbf{w}^{\text{Di, att1}}) = \frac{b\hat{\mathbf{w}}_k + \lambda\mathbf{w}^{\text{Di, att1}}}{b+\lambda} = \mathbf{x}_k^{\text{Me,att1}}$. Then Ditto and pFedMe have the same loss. So we will omit the analysis for Ditto.

lp-proj-2    Similar to pFedMe, if the $k$-th client is benign, it sends $\hat{\mathbf{w}}_{k,1}$ to the server. If the $k$-th client is malicious, it sends $\tilde{\mathbf{w}}_k^{(ma)} = c\mathbf{1}_{d_{\text{sub}}}$ to the server where $c \sim \mathcal{N}(0,\tau^2)$. The server receives the messages and obtains $\tilde{\mathbf{w}}^{\text{l2,att1}} = \frac{1}{N}\left(\sum_{k\in I_b}\hat{\mathbf{w}}_{k,1} + \sum_{k\in I_a}\tilde{\mathbf{w}}_k^{(ma)}\right)$. And we have $\tilde{\mathbf{w}}^{\text{l2,att1}} \sim \mathcal{N}\left(\frac{1}{N}\sum_{k\in I_b}\mathbf{w}_{k,1}, \frac{1}{N^2}\left(\frac{N_b\sigma^2}{bn}\boldsymbol{I}_{d_{\text{sub}}} + N_a\tau^2\boldsymbol{J}_{d_{\text{sub}}}\right)\right)$. Then the server broadcasts $\tilde{\mathbf{w}}^{\text{l2,att1}}$ to all the clients and the benign client $k$ computes the optimal local parameter $\mathbf{x}_k^{\text{l2, att1}} = \check{\mathbf{x}}_k(\tilde{\mathbf{w}}^{\text{l2,att1}}) = \begin{pmatrix}(b\hat{\mathbf{w}}_{k,1} + \lambda\tilde{\mathbf{w}}^{\text{l2, att1}})/(b+\lambda) \\ \hat{\mathbf{w}}_{k,2}\end{pmatrix}$. It follows that

$$\mathbf{x}_k^{\text{l2,att1}} \sim \mathcal{N}\left(\begin{pmatrix}\frac{b\mathbf{w}_{k,1} + \lambda\sum_{i\in I_b}\mathbf{w}_{i,1}/N}{b+\lambda} \\ \mathbf{x}_{k,2}\end{pmatrix}, \begin{pmatrix}\frac{\left[\left(b+\frac{\lambda}{N}\right)^2\frac{\sigma^2}{bn} + (N_b-1)\frac{\sigma^2}{bn}\right]\boldsymbol{I}_{d_{\text{sub}}} + N_a\frac{\lambda^2}{N^2}\tau^2\boldsymbol{J}_{d_{\text{sub}}}}{(b+\lambda)^2} & \\ & \frac{\sigma^2}{bn}\boldsymbol{I}_{d_{\text{sub}}}\end{pmatrix}\right).$$

Then we can compute the averaged loss on benign clients as

$$L^{\text{l2, att1}}(\lambda) = \frac{1}{N_b}\sum_{k\in I_b}f_k^{\text{te}}(\mathbf{x}_k^{\text{l2, att1}})$$

$$= \frac{\sigma^2}{2} + \frac{bd_{\text{sub}}}{2}\cdot\frac{\left(b^2 + \frac{2b\lambda}{N} + \frac{N_b\lambda^2}{N^2}\right)\frac{\sigma^2}{bn} + \frac{N_a\lambda^2}{N^2}\tau^2}{(b+\lambda)^2} + \frac{(d - d_{\text{sub}})\sigma^2}{2n}$$

$$+ \frac{b\lambda^2}{2(b+\lambda)^2} \cdot \frac{1}{N_b} \sum_{i \in I_b} \left\| \frac{\sum_{i \in I_b} \mathbf{w}_{i,1}}{N} - \mathbf{w}_{k,1} \right\|_2^2.$$

And the optimal $\lambda$ is $\lambda^{\text{l2,att1}} = \dfrac{(1-1/N)\sigma^2 d_{\text{sub}}/n}{\frac{1}{N_b} \sum_{k \in I_b} \left\| \frac{\sum_{i \in I_a} \mathbf{w}_{i,1}}{N} - \mathbf{w}_{k,1} \right\|_2^2 + \frac{d_{\text{sub}} N_a}{N^2}(\tau^2 - \sigma^2/(bn))}.$

To make calculations clean, we still assume that the heterogeneity is uniform in all dimensions, i.e.,

$$\frac{1}{dN_b} \sum_{k \in \mathbf{I}_b} \left\| \frac{\sum_{i \in I_b} \mathbf{w}_i}{N} - \mathbf{w}_k \right\|_2^2 = \frac{1}{d_{\text{sub}} N_b} \sum_{k \in I_b} \left\| \frac{\sum_{i \in I_b} \mathbf{w}_{i,1}}{N} - \mathbf{w}_{k,1} \right\|_2^2 := \Sigma_1. \tag{25}$$

Then we have $\lambda^{\text{Me, att1}} = \lambda^{\text{l2,att1}} = \lambda_1^* := \dfrac{(1-1/N)\sigma^2/n}{\Sigma_1 + \frac{N_a}{N^2}(\tau^2 - \sigma^2/(bn))}$. The numerator of $\lambda_1^*$ is the variance of noises over the number of samples. The denominator is the sum of data heterogeneity and variance of attacks.

Now we can obtain the losses of different methods at $\lambda_1^*$.

$$L^{\text{loc, att1}} = \frac{\sigma^2}{2} + \frac{\sigma^2 d}{2n} = \frac{\sigma^2}{2} + \frac{\sigma^2 d}{2n(b+\lambda_1^*)^2}\left[(\lambda_1^*)^2 + 2b\lambda_1^* + b^2\right],$$

$$L^{\text{Avg,att1}} = \frac{\sigma^2}{2} + \frac{\sigma^2 d}{2n(b+\lambda_1^*)^2}\left[\frac{(\lambda_1^*)^2}{N} + \frac{N+1}{N}b\lambda_1^* + \frac{2N-1}{N}b^2 + \frac{N-1}{N}\frac{b^3}{\lambda_1^*}\right],$$

$$L_*^{\text{Me, att1}} = L^{\text{Me,att1}}(\lambda_1^*) = \frac{\sigma^2}{2} + \frac{\sigma^2 d}{2n(b+\lambda_1^*)}\left[\frac{(\lambda_1^*)^2}{N} + \frac{N+1}{N}b\lambda_1^* + b^2\right],$$

$$L_*^{\text{l2, att1}} = L^{\text{l2,att1}}(\lambda_1^*) = \frac{\sigma^2}{2} + \frac{\sigma^2 d_{\text{sub}}}{2n(b+\lambda_1^*)^2}\left[\frac{(\lambda_1^*)^2}{N} + \frac{N+1}{N}b\lambda_1^* + b^2\right] + \frac{\sigma^2(d-d_{\text{sub}})}{2n(b+\lambda_1^*)^2}\left[(\lambda_1^*)^2 + 2b\lambda_1^* + b^2\right].$$

$$\tag{26}$$

We have the following observations.

- $L^{\text{loc, att1}} \geq L_*^{\text{l2, att1}} \geq L_*^{\text{Me, att1}}$ and $L^{\text{Avg, att1}} \geq L_*^{\text{Me, att1}}$. This means that `pFedMe` with the optimal $\lambda$ always has the minimal loss.

- $L^{\text{loc, att1}} \leq L^{\text{Avg, att1}}$ if and only if $\lambda_1^* \leq b$. This means that if the heterogeneity or the noise of attacks is sufficiently large, then `local` is better than `FedAvg`.

- $L_*^{\text{l2}} \leq L^{\text{Avg, att1}}$ if and only if $\lambda_1^* \leq \sqrt{\frac{d}{d-d_{\text{sub}}}}b$. The range of $\lambda^*$ over which `lp-proj-2` is better than `FedAvg` is slightly larger than the range of that over which `local` is better than `FedAvg`.

Since $\tau^2$ can be very large, $\lambda_1^*$ is much smaller than $\lambda^*$. Recall that in the settings of Figure 1, we have $\lambda_1^* =$4.9e-04.

Now we compare the losses for different values of $\lambda$ and give the formal version of Proposition 1.

**Theorem 2** (Formal version of Proposition 1). *We have*

$$L^{loc, att1} = \frac{\sigma^2}{2} + \frac{\sigma^2 d}{2n},$$

$$L^{Avg,att1} = \frac{\sigma^2}{2} + \frac{bd}{2N^2}\left(\frac{N_b \sigma^2}{bn} + N_a \tau^2\right) + \frac{bd\Sigma_1}{2},$$

$$L^{Me,att1}(\lambda) = \frac{\sigma^2}{2} + \frac{bd}{2} \cdot \frac{\left(b^2 + \frac{2b\lambda}{N} + \frac{N_b \lambda^2}{N^2}\right)\frac{\sigma^2}{bn} + \frac{N_a \lambda^2}{N^2}\tau^2}{(b+\lambda)^2} + \frac{b\lambda^2 d\Sigma_1}{2(b+\lambda)^2},$$

$$L^{Di, att1}(\lambda) = L^{Me,att1}(\lambda),$$

$$L^{l2, att1}(\lambda) = \frac{\sigma^2}{2} + \frac{bd_{\text{sub}}}{2} \cdot \frac{\left(b^2 + \frac{2b\lambda}{N} + \frac{N_b \lambda^2}{N^2}\right)\frac{\sigma^2}{bn} + \frac{N_a \lambda^2}{N^2}\tau^2}{(b+\lambda)^2} + \frac{(d-d_{\text{sub}})\sigma^2}{2n} + \frac{b\lambda^2 d_{\text{sub}}\Sigma_1}{2(b+\lambda)^2}.$$

$$\tag{27}$$

*And the following propositions hold.*

- $L^{loc,att1} \leq L^{Avg,att1}$ if and only if $\Sigma_1 + \frac{N_a}{N^2}\left(\tau^2 - \frac{\sigma^2}{bn}\right) \geq \frac{N-1}{N}\frac{\sigma^2}{bn}$ ($\lambda_1^* \leq b$).

- $L^{loc,\,att1} \leq L^{l2,\,att1}(\lambda) \leq L^{Me,att1}(\lambda)$ if and only if $\Sigma_1 + \frac{N_a}{N^2}\left(\tau^2 - \frac{\sigma^2}{bn}\right) \geq \frac{N-1}{N}\frac{2b+\lambda}{\lambda}\frac{\sigma^2}{bn}$. If $\Sigma_1 + \frac{N_a}{N^2}\left(\tau^2 - \frac{\sigma^2}{bn}\right) > \frac{N-1}{N}\frac{\sigma^2}{bn}$ ($\lambda_1^* < b$), this is equivalent to $\lambda \geq \frac{2\lambda_1^*}{1 - \lambda_1^*/b}$.

- $L^{Me,\,att1}(\lambda) \leq L^{Avg,\,att1}$ if and only if $\Sigma_1 + \frac{N_a}{N^2}\left(\tau^2 - \frac{\sigma^2}{bn}\right) \geq \frac{N-1}{N}\frac{\sigma^2}{(b+2\lambda)n}$. This is equivalent to $\lambda \geq \frac{\lambda_1^* - b}{2}$.

- $L^{l2,\,att1}(\lambda) \leq L^{Avg,\,att1}$ if and only if $\Sigma_1 + \frac{N_a}{N^2}\left(\tau^2 - \frac{\sigma^2}{bn}\right) \geq \frac{N-1}{N}\frac{\sigma^2}{bn}\frac{d(b+\lambda)^2 - d_{\text{sub}}\lambda(2b+\lambda)}{d(b+\lambda)^2 - d_{\text{sub}}\lambda^2}$.

With (27), it is easy to check the above propositions hold.

If the attacks are very serious, we can have $\Sigma_1 + \frac{N_a}{N^2}\left(\tau^2 - \frac{\sigma^2}{bn}\right) > \frac{N-1}{N}\frac{\sigma^2}{bn}$ ($\lambda_1^* < b$). Similar to the analysis at the end of Appendix B.2, when $\lambda \leq \frac{2\lambda_1^*}{1 - \lambda_1^*/b}$, we have $L^{Me,\,att1}(\lambda) \leq L^{l2,\,att1}(\lambda) \leq L^{loc,\,att1} \leq L^{Avg,\,att1}$; when $\lambda > \frac{2\lambda_1^*}{1 - \lambda_1^*/b}$, we have $L^{loc,\,att1} \leq L^{l2,\,att1}(\lambda) \leq L^{Me,\,att1}(\lambda) \leq L^{Avg,\,att1}$.

### B.3.3 Sign-flipping Attacks

The second type of attack is sign-flipping attacks. For simplicity, we define $\bar{\mathbf{w}}_b = \frac{1}{N_b}\sum_{i \in I_b}\mathbf{w}_i$, $\bar{\mathbf{w}}_a = \frac{1}{N_a}\sum_{i \in I_a}\mathbf{w}_i$, $\bar{\mathbf{w}}_{b,1} = \frac{1}{N_b}\sum_{i \in I_b}\mathbf{w}_{i,1}$ and $\bar{\mathbf{w}}_{a,1} = \frac{1}{N_a}\sum_{i \in I_a}\mathbf{w}_{i,1}$.

`local` This attack will not affect `local`. So the averaged test loss on benign clients is $L^{loc,\,att2} = \frac{1}{N_b}\sum_{k \in I_b}f_k^{\text{te}}(\mathbf{w}_k^{\text{loc}}) = \frac{\sigma^2}{2} + \frac{\sigma^2 d}{2n}$.

`FedAvg` If the $k$-th client is benign, it sends $\hat{\mathbf{w}}_k$ to the server. If the $k$-th client is malicious, it send $\mathbf{w}_k^{(ma)} = -|c|\hat{\mathbf{w}}_k$ to the server, where $c \sim \mathcal{N}(0, \tau^2)$. Recall that $\hat{\mathbf{w}}_k$ is an unbiased observation of $\mathbf{w}_k$ with covariance matrix $\frac{\sigma^2}{bn}\mathbf{I}_d$. Now we examine $\mathbf{w}_k^{(ma)}$. Although $\mathbf{w}_k^{(ma)}$ does not obey Gaussian distribution, we can still calculate its mean and covariance by the independence of $c$ and $\hat{\mathbf{w}}_k$. We have $\mathbb{E}\left[-|c|\hat{\mathbf{w}}_k\right] = -\sqrt{2/\pi}\tau\mathbf{w}_k$ and

$$\text{var}\left(-|c|\hat{\mathbf{w}}_k\right) = \mathbb{E}\left[c^2\hat{\mathbf{w}}_k\hat{\mathbf{w}}_k^\top\right] - \mathbb{E}\left[|c|\hat{\mathbf{w}}_k\right]\mathbb{E}\left[|c|\hat{\mathbf{w}}_k^\top\right]$$
$$= \tau^2\left(\mathbf{w}_k\mathbf{w}_k^\top + \frac{\sigma^2}{bn}\mathbf{I}_d\right) - \frac{2}{\pi}\tau^2\mathbf{w}_k\mathbf{w}_k^\top$$
$$= \frac{\pi - 2}{\pi}\tau^2\mathbf{w}_k\mathbf{w}_k^\top + \tau^2\frac{\sigma^2}{bn}\mathbf{I}_d.$$

Then $\mathbf{w}_k^{(ma)}$ can be regarded as an unbiased observation of $-\sqrt{2/\pi}\tau\mathbf{w}_k$ with covariance matrix $\mathbf{M}_k = \frac{\pi-2}{\pi}\tau^2\mathbf{w}_k\mathbf{w}_k^\top + \frac{\tau^2\sigma^2}{bn}\mathbf{I}_d$. Thus the server gets $\mathbf{w}^{\text{Avg, att2}} = \frac{1}{N}\left(\sum_{k \in I_b}\hat{\mathbf{w}}_k + \sum_{k \in I_a}\mathbf{w}_k^{(ma)}\right)$. We have $\mathbb{E}[\mathbf{w}^{\text{Avg, att2}}] = \frac{N_b\bar{\mathbf{w}}_b - N_a\sqrt{2/\pi}\tau\bar{\mathbf{w}}_a}{N}$ and $\text{var}(\mathbf{w}^{\text{Avg, att2}}) = \frac{1}{N^2}\left(\frac{N_b\sigma^2}{bn}\mathbf{I}_d + \sum_{k \in I_a}\mathbf{M}_k\right)$.

Then we can compute the averaged test loss on benign clients as

$$L^{\text{Avg, att2}} = \frac{1}{N_b}\sum_{k \in I_b}f_k^{\text{te}}(\mathbf{w}^{\text{Avg, att2}})$$
$$= \frac{\sigma^2}{2} + \frac{bd}{2N^2}\left[\frac{N_b\sigma^2}{bn} + \sum_{k \in I_a}\left(\frac{\pi-2}{\pi}\tau^2\|\mathbf{w}_k\|_2^2 + \frac{\tau^2\sigma^2}{bn}\right)\right] + \frac{b}{2N}\sum_{k \in I_a}\left\|\frac{N_b\bar{\mathbf{w}}_b - N_a\sqrt{2/\pi}\tau\bar{\mathbf{w}}_a}{N} - \mathbf{w}_k\right\|_2^2.$$

`pFedMe` If the $k$-th client is benign, it will send $\hat{\mathbf{w}}_k$ to the server. If the $k$-th client is malicious, it will send $\mathbf{w}_k^{(ma)} = -|c|\hat{\mathbf{w}}_k$ to the server. The server obtains $\mathbf{w}^{\text{Me, att2}} = \frac{1}{N}\left(\sum_{k \in I_b}\hat{\mathbf{w}}_k + \sum_{k \in I_a}\mathbf{w}_k^{(ma)}\right) = \mathbf{w}^{\text{Avg, att2}}$ and broadcasts $\mathbf{w}^{\text{Me, att2}}$ back to all the clients. For

the benign client $k$, it gets $\mathbf{x}_k^{\text{Me, att2}} = \hat{\mathbf{x}}_k(\mathbf{w}^{\text{Me, att2}}) = \frac{b\hat{\mathbf{w}}_k + \lambda \mathbf{w}^{\text{Me, att2}}}{b+\lambda}$. We have $\mathbb{E}[\mathbf{x}_k^{\text{Me, att2}}] = \frac{1}{b+\lambda}\left[b\mathbf{w}_k + \lambda \frac{N_b\bar{\mathbf{w}}_b - N_a\sqrt{2/\pi}\tau\bar{\mathbf{w}}_a}{N}\right]$ and $\text{var}(\mathbf{x}_k^{\text{Me, att2}}) = \frac{\left[\left(b+\frac{\lambda}{N}\right)^2\frac{\sigma^2}{bn} + (N_b-1)\frac{\lambda^2}{N^2}\frac{\sigma^2}{bn}\right]\mathbf{I}_d + \frac{\lambda^2}{N^2}\sum_{i\in I_a}\mathbf{M}_i}{(b+\lambda)^2}$.

Then we can compute the averaged test loss on benign clients as

$$L^{\text{Me, att2}}(\lambda) = \frac{1}{N_b}\sum_{k\in I_b} f_k^{\text{te}}(\mathbf{x}_k^{\text{Me, att2}})$$

$$= \frac{\sigma^2}{2} + \frac{bd}{2}\frac{\left(b^2 + \frac{2b\lambda}{N} + \frac{N_b\lambda^2}{N^2}\right)\frac{\sigma^2}{bn} + \frac{\lambda^2}{N^2}\sum_{i\in I_a}\frac{\text{tr}(\mathbf{M}_i)}{d}}{(b+\lambda)^2}$$

$$+ \frac{b\lambda^2}{2(b+\lambda)^2}\frac{1}{N_b}\sum_{k\in I_b}\left\|\frac{N_b\bar{\mathbf{w}}_b - N_a\sqrt{2/\pi}\tau\bar{\mathbf{w}}_a}{N} - \mathbf{w}_k\right\|_2^2,$$

where $\text{tr}(\mathbf{M}_i) = \frac{\pi-2}{\pi}\tau^2\|\mathbf{w}_i\|_2^2 + \frac{\tau^2\sigma^2 d}{bn}$.

And the optimal $\lambda$ is $\lambda^{\text{Me, att2}} = \frac{(1-1/N)\sigma^2 d/n}{\frac{1}{N_b}\sum_{k\in I_b}\left\|\frac{N_b\bar{\mathbf{w}}_b - N_a\sqrt{2/\pi}\tau\bar{\mathbf{w}}_a}{N} - \mathbf{w}_k\right\|_2^2 + \frac{dN_a}{N^2}\left(\frac{\sum_{i\in I_a}\text{tr}(\mathbf{M}_i)/d}{N_a} - \sigma^2/(bn)\right)}$.

`Ditto`  The global model of `Ditto` is the same as the model of `FedAvg`. Similar to the analysis of same-value attacks, we have $\mathbf{w}^{\text{Di, att2}} = \mathbf{w}^{\text{Avg, att2}}$ and $\mathbf{x}_k^{\text{Di, att2}} = \hat{\mathbf{x}}_k(\mathbf{w}^{\text{Di, att2}}) = \frac{b\hat{\mathbf{w}}_k + \lambda \mathbf{w}^{\text{Di, att2}}}{b+\lambda} = \mathbf{x}_k^{\text{Me, att2}}$. Then `Ditto` and `pFedMe` have the same losses. We will also omit the analysis for `Ditto`.

`lp-proj-2`  If the $k$-th client is benign, it will send $\hat{\mathbf{w}}_{k,1}$ to the server. If the $k$-th client is malicious, it will send $\tilde{\mathbf{w}}_k^{(ma)} = -|c|\hat{\mathbf{w}}_{k,1}$ to the server, where $c \sim \mathcal{N}(0,\tau^2)$. Then we have $\mathbb{E}[\tilde{\mathbf{w}}_k^{(ma)}] = -\sqrt{2/\pi}\tau\mathbf{w}_{k,1}$ and $\text{var}(\tilde{\mathbf{w}}_k^{(ma)}) = \frac{\pi-2}{\pi}\tau^2\mathbf{w}_{k,1}\mathbf{w}_{k,1}^\top + \tau^2\frac{\sigma^2}{bn}\mathbf{I}_{d_{\text{sub}}} := \widetilde{\mathbf{M}}_k$. The server receives these messages and gets $\tilde{\mathbf{w}}^{\text{l2, att2}} = \frac{1}{N}\left(\sum_{k\in I_b}\hat{\mathbf{w}}_{k,1} + \sum_{k\in I_a}\tilde{\mathbf{w}}_k^{(ma)}\right)$. And the benign client $k$ obtains $\mathbf{x}_k^{\text{l2, att2}} = \check{\mathbf{x}}_k(\widetilde{\mathbf{w}}^{\text{l2, att2}}) = \begin{pmatrix}(b\hat{\mathbf{w}}_{k,1} + \lambda\tilde{\mathbf{w}}^{\text{l2, att2}})/(b+\lambda) \\ \hat{\mathbf{w}}_{k,2}\end{pmatrix}$.

Then we have $\mathbb{E}[\mathbf{x}_k^{\text{l2, att2}}] = \begin{pmatrix}\frac{1}{b+\lambda}\left[b\mathbf{w}_{k,1} + \lambda\frac{N_b\bar{\mathbf{w}}_{b,1} - N_a\sqrt{2/\pi}\tau\bar{\mathbf{w}}_{a,1}}{N}\right] \\ \mathbf{w}_{k,2}\end{pmatrix}$ and $\text{var}(\mathbf{x}_k^{\text{l2, att2}}) = $

$\begin{pmatrix}\frac{\left[\left(b+\frac{\lambda}{N}\right)^2\frac{\sigma^2}{bn} + (N_b-1)\frac{\lambda^2}{N^2}\frac{\sigma^2}{bn}\right]\mathbf{I}_d + \frac{\lambda^2}{N^2}\sum_{i\in I_a}\widetilde{\mathbf{M}}_i}{(b+\lambda)^2} & \\ & \frac{\sigma^2}{bn}\mathbf{I}_{d-d_{\text{sub}}}\end{pmatrix}$. The averaged test loss on benign

clients is

$$L^{\text{l2, att2}}(\lambda) = \frac{1}{N_b}\sum_{k\in I_b} f_k^{\text{te}}(\mathbf{x}_k^{\text{l2, att2}})$$

$$= \frac{\sigma^2}{2} + \frac{bd_{\text{sub}}}{2}\frac{\left(b^2 + \frac{2b\lambda}{N} + \frac{N_b\lambda^2}{N^2}\right)\frac{\sigma^2}{bn} + \frac{\lambda^2}{N^2}\sum_{i\in I_a}\frac{\text{tr}(\widetilde{\mathbf{M}}_i)}{d_{\text{sub}}}}{(b+\lambda)^2} + \frac{(d-d_{\text{sub}})\sigma^2}{2n}$$

$$+ \frac{b\lambda^2}{2(b+\lambda)^2}\frac{1}{N_b}\sum_{k\in I_b}\left\|\frac{N_b\bar{\mathbf{w}}_{b,1} - N_a\sqrt{2/\pi}\tau\bar{\mathbf{w}}_{a,1}}{N} - \mathbf{w}_{k,1}\right\|_2^2,$$

where $\text{tr}(\widetilde{\mathbf{M}}_i) = \frac{\pi-2}{\pi}\tau^2\|\mathbf{w}_{i,1}\|_2^2 + \frac{\tau^2\sigma^2 d_{\text{sub}}}{bn}$.  And the optimal lambda is $\lambda^{\text{l2, att2}} = \frac{(1-1/N)\sigma^2 d_{\text{sub}}/n}{\frac{1}{N_b}\sum_{k\in I_b}\left\|\frac{N_b\bar{\mathbf{w}}_{b,1} - N_a\sqrt{2/\pi}\tau\bar{\mathbf{w}}_{a,1}}{N} - \mathbf{w}_{k,1}\right\|_2^2 + \frac{d_{\text{sub}}N_a}{N^2}\left(\frac{\sum_{i\in I_a}\text{tr}(\widetilde{\mathbf{M}}_i)/d_{\text{sub}}}{N_a} - \sigma^2/(bn)\right)}$

We still focus on the case where the heterogeneity and norm of $\mathbf{w}_k$ are uniform in all dimensions, that is,

$$\frac{1}{dN_b}\sum_{k\in I_b}\left\|\frac{N_b\bar{\mathbf{w}}_b - N_a\sqrt{2/\pi}\tau\bar{\mathbf{w}}_a}{N} - \mathbf{w}_k\right\|_2^2 = \frac{1}{d_{\text{sub}}N_b}\sum_{k\in I_b}\left\|\frac{N_b\bar{\mathbf{w}}_{b,1} - N_a\sqrt{2/\pi}\tau\bar{\mathbf{w}}_{a,1}}{N} - \mathbf{w}_{k,1}\right\|_2^2 := \Sigma_2$$

and $\frac{1}{dN_a}\sum_{i\in I_a}\mathrm{tr}(\boldsymbol{M}_i) = \frac{1}{d_{\mathrm{sub}}N_a}\sum_{i\in I_a}\mathrm{tr}(\widetilde{\boldsymbol{M}}_i) := M_0$. Then we have $\lambda^{\mathrm{Me,\,att2}} = \lambda^{\mathrm{l2,att2}} = \lambda_2^* :=$
$\frac{(1-1/N)\sigma^2/n}{\Sigma_2 + \frac{N_a}{N^2}(M_0 - \sigma^2/(bn))}$. The losses of different methods at optimal $\lambda_2^*$ have similar forms as (26), with
$\lambda_1^*$ replaced by $\lambda_2^*$. The discussion below (27) also applies here.

For the comparison of losses at different values of $\lambda$, we have

$$L^{\mathrm{loc,\,att2}} = \frac{\sigma^2}{2} + \frac{\sigma^2 d}{2n},$$

$$L^{\mathrm{Avg,att2}} = \frac{\sigma^2}{2} + \frac{bd}{2N^2}\left(\frac{N_b\sigma^2}{bn} + N_a M_0\right) + \frac{bd\Sigma_2}{2},$$

$$L^{\mathrm{Me,att2}}(\lambda) = \frac{\sigma^2}{2} + \frac{bd}{2}\cdot\frac{\left(b^2 + \frac{2b\lambda}{N} + \frac{N_b\lambda^2}{N^2}\right)\frac{\sigma^2}{bn} + \frac{N_a\lambda^2}{N^2}M_0}{(b+\lambda)^2} + \frac{b\lambda^2 d\Sigma_2}{2(b+\lambda)^2},$$

$$L^{\mathrm{l2,\,att2}}(\lambda) = \frac{\sigma^2}{2} + \frac{bd_{\mathrm{sub}}}{2}\cdot\frac{\left(b^2 + \frac{2b\lambda}{N} + \frac{N_b\lambda^2}{N^2}\right)\frac{\sigma^2}{bn} + \frac{N_a\lambda^2}{N^2}M_0}{(b+\lambda)^2} + \frac{(d-d_{\mathrm{sub}})\sigma^2}{2n} + \frac{b\lambda^2 d_{\mathrm{sub}}\Sigma_2}{2(b+\lambda)^2}.$$

Thus the propositions and discussion below (27) also hold here, with $\Sigma_1$, $\tau^2$, $\lambda_1^*$ replaced by $\Sigma_2$, $M_0$ and $\lambda_2^*$ respectively.

### B.3.4 Gaussian Attacks

Gaussian attacks are similar to same-value attacks. For `FedAvg`, `pFedMe` and `Ditto`, the malicious client sends $\mathbf{w}_k^{(ma)}$ to the server, where $\mathbf{w}_k^{(ma)} \sim \mathcal{N}(\mathbf{0}_d, \tau^2\boldsymbol{I}_d)$. For `lp-proj-2`, the malicious client sends $\tilde{\mathbf{w}}_k^{(ma)}$ to the server, where $\tilde{\mathbf{w}}_k^{(ma)} \sim \mathcal{N}(\mathbf{0}_{d_{\mathrm{sub}}}, \tau^2\boldsymbol{I}_{d_{\mathrm{sub}}})$.

Note that $\mathrm{tr}(\boldsymbol{I}_d) = \mathrm{tr}(\boldsymbol{J}_d)$ for any $d$ and the test loss (23) is only relevant to the trace of the covariance matrix. Thus the averaged test losses on benign clients under Gaussian attacks are the same as those under same-value attacks.

### B.4 Fairness

In this subsection, we examine the performance fairness of these methods. Recall that in Definition 1, we measure performance fairness in terms of the variance of test accuracy/losses. In Appendix B.2, the test loss on client $k$ is

$$f_k^{\mathrm{te}}(\mathbf{x}_k) = \frac{\sigma^2}{2} + \frac{b}{2}\mathrm{tr}(\mathrm{var}(\mathbf{x}_k)) + \frac{b}{2}\|\mathbb{E}\mathbf{x}_k - \mathbf{w}_k\|_2^2.$$

For different methods, we can compute that

$$f_k^{\mathrm{te}}(\mathbf{w}_k^{\mathrm{loc}}) = \frac{\sigma^2}{2} + \frac{\sigma^2 d}{2n},$$

$$f_k^{\mathrm{te}}(\mathbf{w}^{\mathrm{Avg}}) = \frac{\sigma^2}{2} + \frac{\sigma^2 d}{2Nn} + \frac{b}{2}\|\bar{\mathbf{w}} - \mathbf{w}_k\|_2^2,$$

$$f_k^{\mathrm{te}}(\mathbf{x}_k^{\mathrm{Me}}) = \frac{\sigma^2}{2} + \frac{b^2 + \frac{2b\lambda}{N} + \frac{\lambda^2}{N}}{(b+\lambda)^2}\frac{\sigma^2 d}{2n} + \frac{b\lambda^2}{2(b+\lambda)^2}\|\bar{\mathbf{w}} - \mathbf{w}_k\|_2^2,$$

$$f_k^{\mathrm{te}}(\mathbf{x}_k^{\mathrm{Di}}) = f_k^{\mathrm{te}}(\mathbf{x}_k^{\mathrm{Me}}),$$

$$f_k^{\mathrm{te}}(\mathbf{x}_k^{\mathrm{l2}}) = \frac{\sigma^2}{2} + \frac{b^2 + \frac{2b\lambda}{N} + \frac{\lambda^2}{N}}{(b+\lambda)^2}\frac{\sigma^2 d_{\mathrm{sub}}}{2n} + \frac{\sigma^2(d - d_{\mathrm{sub}})}{2n} + \frac{b\lambda^2}{2(b+\lambda)^2}\|\bar{\mathbf{w}}_{\cdot,1} - \mathbf{w}_{k,1}\|_2^2.$$

Define $\mathrm{var}_k(a_k) = \frac{1}{N}\sum_{k=1}^N a_k^2 - \left(\frac{1}{N}\sum_{k=1}^N a_k\right)^2$. Then we give the formal version of Proposition 2.

**Theorem 3** (Formal version of Proposition 2). *The variances of test losses on different clients for these methods are as follows.*

$$V^{loc} = \mathsf{var}_k(f_k^{te}(\mathbf{w}_k^{loc})) = 0,$$

$$V^{Avg} = \mathsf{var}_k(f_k^{te}(\mathbf{w}^{Avg})) = \frac{b^2}{4}\mathsf{var}_k(\|\bar{\mathbf{w}} - \mathbf{w}_k\|_2^2),$$

$$V^{Me}(\lambda) = \mathsf{var}_k(f_k^{te}(\mathbf{x}_k^{Me})) = \frac{b^2\lambda^4}{4(b+\lambda)^4}\mathsf{var}_k(\|\bar{\mathbf{w}} - \mathbf{w}_k\|_2^2),$$

$$V^{Di}(\lambda) = V^{Me}(\lambda),$$

$$V^{l2}(\lambda) = \mathsf{var}_k(f_k^{te}(\mathbf{x}_k^{l2})) = \frac{b^2\lambda^4}{4(b+\lambda)^4}\mathsf{var}_k(\|\bar{\mathbf{w}}_{\cdot,1} - \mathbf{w}_{k,1}\|_2^2).$$

*If $\mathbf{w}_k$ are i.i.d. and distributed as $\mathcal{N}(\mu_w, \boldsymbol{\Sigma}_w)$ with $\boldsymbol{\Sigma}_w = (\sigma_{ij})_{d \times d}$, we have*

$$\mathbb{E}V^{Avg} = b^2\left(1 - \frac{2}{N}\right)\sum_{i=1}^{d}\sum_{j=1}^{d}\sigma_{ij}^2 = O\left(d^2\right),$$

$$\mathbb{E}V^{Me}(\lambda) = \frac{b^2\lambda^4}{(b+\lambda)^4}\left(1 - \frac{2}{N}\right)\sum_{i=1}^{d}\sum_{j=1}^{d}\sigma_{ij}^2 = O\left(d^2\right),$$

$$\mathbb{E}V^{l2}(\lambda) = \frac{b^2\lambda^4}{(b+\lambda)^4}\left(1 - \frac{2}{N}\right)\sum_{i=1}^{d_{\text{sub}}}\sum_{j=1}^{d_{\text{sub}}}\sigma_{ij}^2 = O\left(d_{\text{sub}}^2\right).$$

By Theorem 3, we have $V^{\text{loc}} \leq V^{\text{Me}}(\lambda) \leq V^{\text{Avg}}$ and $V^{\text{loc}} \leq V^{\text{l2}}(\lambda)$. And larger $\lambda$ leads to more fairness. This is because in our settings, only the true parameters $\mathbf{w}_k$ on the clients are different. For `local`, $\mathbf{w}_k^{\text{loc}}$ is an unbiased estimation of $\mathbf{w}_k$. So $f_k^{\text{te}}(\mathbf{w}_k^{\text{loc}}) = f_l^{\text{te}}(\mathbf{w}_l^{\text{loc}})$ for any $k \neq l$. For other methods, $\mathbf{x}_k^{\text{Avg}}$, $\mathbf{x}_k^{\text{Me}}$ and $\mathbf{x}_k^{\text{l2}}$ are all biased. Thus test losses on different clients can vary a lot.

However, it is not easy to compare $\mathsf{var}_k(\|\bar{\mathbf{w}} - \mathbf{w}_k\|_2^2)$ and $\mathsf{var}_k(\|\bar{\mathbf{w}}_{\cdot,1} - \mathbf{w}_{k,1}\|_2^2)$ directly. If the variance of $\mathbf{w}_k$ concentrates on the the first $d_{\text{sub}}$ dimensions, $\mathsf{var}_k(\|\bar{\mathbf{w}}_{\cdot,1} - \mathbf{w}_{k,1}\|_2^2)$ can be larger than $\mathsf{var}_k(\|\bar{\mathbf{w}} - \mathbf{w}_k\|_2^2)$.

To gain more intuition, we further assume $\mathbf{w}_k$ are i.i.d. and distributed as $\mathcal{N}(\mu_w, \boldsymbol{\Sigma}_w)$. Then Theorem 3 implies that $\mathbb{E}V^{\text{loc}} \leq \mathbb{E}V^{\text{l2}}(\lambda) \leq \mathbb{E}V^{\text{Me}}(\lambda) \leq \mathbb{E}V^{\text{Avg}}$.

Now we give the proof of Theorem 3.

*Proof of Theorem 3.* The first part is easy to check. For the second part, we first give an equivalent form of $\mathsf{var}_k(\|\bar{\mathbf{w}} - \mathbf{w}_k\|_2^2)$.

$$\mathsf{var}_k(\|\bar{\mathbf{w}} - \mathbf{w}_k\|_2^2) = \frac{1}{N}\sum_{k=1}^{N}\|\bar{\mathbf{w}} - \mathbf{w}_k\|_2^4 - \left(\frac{1}{N}\sum_{k=1}^{N}\|\bar{\mathbf{w}} - \mathbf{w}_k\|_2^2\right)^2$$

$$= \frac{1}{N^2}\left[(N-1)\sum_{k=1}^{N}\|\mathbf{w}_k - \bar{\mathbf{w}}\|_2^4 - \sum_{k \neq l}\|\mathbf{w}_k - \bar{\mathbf{w}}\|_2^2\|\mathbf{w}_l - \bar{\mathbf{w}}\|_2^2\right]$$

$$= \frac{1}{N^2}\sum_{k \neq l}\frac{\left(\|\mathbf{w}_k - \bar{\mathbf{w}}\|_2^2 - \|\mathbf{w}_l - \bar{\mathbf{w}}\|_2^2\right)^2}{2}$$

$$= \frac{1}{N^2}\sum_{k \neq l}\langle\mathbf{w}_k - \mathbf{w}_l, \mathbf{w}_k + \mathbf{w}_l - 2\bar{\mathbf{w}}\rangle^2.$$

Since $\mathbf{w}_k$ are i.i.d. and $\mathbf{w}_k \sim \mathcal{N}(\mu_w, \boldsymbol{\Sigma}_w)$, one can check that $\mathsf{cov}(\mathbf{w}_k - \mathbf{w}_l, \mathbf{w}_k + \mathbf{w}_l - 2\bar{\mathbf{w}}) = 0$, which implies that $\mathbf{w}_k - \mathbf{w}_l$ and $\mathbf{w}_k + \mathbf{w}_l - 2\bar{\mathbf{w}}$ are independent. Moreover, for $k \neq l$, we have $\mathbf{w}_k - \mathbf{w}_l \sim \mathcal{N}(\mathbf{0}_m, 2\boldsymbol{\Sigma}_w)$ and $\mathbf{w}_k + \mathbf{w}_l - 2\bar{\mathbf{w}} \sim \mathcal{N}(\mathbf{0}_m, (2-4/N)\boldsymbol{\Sigma}_w)$. It follows that $\mathbb{E}\,\mathsf{var}_k(\|\bar{\mathbf{w}} - \mathbf{w}_k\|_2^2) = 4(1-2/N)\sum_{i=1}^{d}\sum_{j=1}^{d}\sigma_{ij}^2$. Similarly, we have $\mathbb{E}\,\mathsf{var}_k(\|\bar{\mathbf{w}}_{\cdot,1} - \mathbf{w}_{k,1}\|_2^2) = 4(1-2/N)\sum_{i=1}^{d_{\text{sub}}}\sum_{j=1}^{d_{\text{sub}}}\sigma_{ij}^2$. This completes the proof. $\square$

# C    Experimental Details

The datasets, corresponding models and tasks are summarized in Table 2 below. The performance of `lp-proj-1` and `lp-proj-2` are evaluated on both convex and non-convex models across a set of FL benchmarks, including both synthetic and real datasets.

The synthetic datasets are generated following the setup in Li et al. [43], we denote it as `Synthetic`$(\alpha, \beta)$, where $\alpha$ controls how much local models differ from each other and $\beta$ controls how much the local data for each client differs from that of other clients. Specifically, the synthetic samples $(\boldsymbol{X}_k, y_k)$ are generated from the model $y = \arg\max(\text{softmax}(\boldsymbol{W}_k x + \boldsymbol{b}_k))$ with $x \in \mathbb{R}^{60}$, $\boldsymbol{W} \in \mathbb{R}^{10 \times 60}$ and $\boldsymbol{b}_k \in \mathbb{R}^{10}$, where $\boldsymbol{X}_k \in \mathbb{R}^{n_k \times 60}$ and $y_k \in \mathbb{R}^{n_k}$. Each entry of $\boldsymbol{W}_k$ and $\boldsymbol{b}_k$ are modeled as $N(\mu_k, 1)$ with $\mu_k \sim (0, \alpha)$, and $(x_k)_j \sim N(v_k, \frac{1}{j^{1.2}})$ with $v_k \sim N(B_k, 1)$ and $B_k \sim N(0, \beta)$.

**Table 2:** Summary of datasets and models.

| Datasets | # of Clients | Average Sample Size for each Client | Tasks | Partitions | Models |
|---|---|---|---|---|---|
| Synthetic(0, 0) | 100 | 202 | 10-class classification | $\star$ | logistic |
| Synthetic(1, 1) | 100 | 202 | 10-class classification | $\star$ | logistic |
| EMNIST | 248 | 2000 | 62-class classification | 10 classes to each client | 2-hidden-layers NN |
| CIFAR | 200 | 200 | 10-class classification | 2 classes to each client | CNN |
| MNIST | 100 | 434 | 10-class classification | 2 classes to each client | 1-hidden-layer NN |
| FASHIONMNIST | 100 | 480 | 10-class classification | 2 classes to each client | CNN |

**Neural Network Architecture for the Models used in Numerical Experiments.**

- **1-hidden-layer NN for MNIST**: One hidden fully-connected layer with 100 neurons. We use ReLU as the activation function.

- **2-hidden-layer NN for EMNIST**: Two hidden fully-connected layers, each with 100 neurons. For each FC layer, ReLU is used as the activation function.

- **CNN for CIFAR**: The neural network used in our experiment consists of two convolutioal layers and three fully-connected layers. The architecture for each layer is listed as follows:
    - Convolutional layer 1: input_channel: 3, output_channel: 6, kernel_size: 5.
    - Convolutional layer 2: input_channel: 6, output_channel: 16, kernel_size: 5.
    - Fully-connected layer 1: input_features: 400, output_features: 120.
    - Fully-connected layer 2: input_features: 120, output_features: 84.
    - Fully-connected layer 3: input_features: 84, output_features: 10.

  For each convolutional layer, we would firstly apply a ReLU activation function right after the convolution, and then apply a max pooling with $kernel\_size = 2, stride = 2$ to extract the feature map. Besides, for the fully-connected layers, we use ReLU as the activation function.

- **CNN for FASHIONMNIST**: The neural network used for `FASHIONMNIST` dataset in our experiment is modified from He et al. [23], which consists of a normal convolutional layer, two resnet block and finally a fully connected layer. The architecture for each layer is lister as follows:
    - Convolutional layer: input_channel: 1, output_channel: 64, kernel_size: 7, stride: 2, padding: 3. Right after the convolution, we apply a batch normalization layer to standardize the features, and then the ReLU function is applied as the activation function, and finally, a max pooling layer with $kernel\_size = 3, stride = 2, padding = 1$ is applied to extract the feature map.
    - Resnet block 1: input_channels: 64, output_channels: 64, number of residuals: 2.
    - Resnet block 2: input_channels: 64, output_channels: 128, number of residuals: 2.
    - Fully-connected layer: input_features: 128, output_features: 10.

  Furthermore, we apply average pooling right after resnet block 2 to extract the feature map before feeding to the fully connected layer.

For implementation details, please refer to the source code provided in `https://github.com/desternylin/perfed`.

**Total Number of Parameters for the Full Model.**

- **Synthetic**: 610.
- **MNIST**: 79510.
- **CIFAR**: 62006.
- **EMNIST**: 94862.
- **FASHIONMNIST**: 678794.

**Computing Resource for Numerical Experiments.** All of our experiments are performed on GPUs. Specifically, every single experiment (a competing method with its given parameter setting and model) is performed on a single GPU, where the type of GPU is one of the following two:

- NVIDIA TITAN RTX with 24220MB memory, driver version: 470.63.01, CUDA version: 11.4.
- NVIDIA GeForce RTX 2080 Ti with 11019MB memory, driver version: 470.63.01, CUDA version: 11.4.

## C.1 Competing Methods

Several state-of-the-art methods in the literature aiming for different purposes such as personalization, robustness and communication efficiency are considered in our experiment. We list and provide a brief description of these methods below.

- **FedAvg** [51], which learns a shared model by averaging the locally-computed model updates in each communication round. This is a baseline algorithm in the FL literature, but it would probably suffer from the statistical heterogeneity among clients.
- **Local**, which trains a local model for each client separately. This algorithm does not have the communication burden issue, but may perform poorly when there is little local data.
- **Ditto** [44], which considers two overarching tasks: the global objectives and the local objectives, and uses a regularization term that encourages the personalized model to be close to the optimal global model.
- **LG-FedAvg** [46], which proposes to learn useful and compact features from the raw data locally and the central server only aggregates the learned representations to improve communication efficiency and get better personalization performance.
- **RSA** [42], which incorporates the objective function with an $L^p$ regularizer to robustify the learning task and mitigate the negative effects of Byzantine attacks.
- **pFedMe** [13], which considers a bi-level problem that concerns global and local objectives respectively. The main difference of `pFedMe` and `Ditto` is that when considering the global objective, `pFedMe` considers the whole loss function including the regularizer, while `Ditto` excludes the regularizer in the global level.
- **Per-FedAvg** [16], which applies MAML [18] to personalize federated models with a Hessian-product approximation to approximate the second-order gradients.
- **Sketch** [29], which carries out distributed SGD by communicating count sketches instead of full gradients to reduce communication cost. However, in our experiments, we find that the size of the sketches should be relatively large to retain accuracy performance in heterogeneous networks.
- **LBGM** (Look-Back Gradient Multiplier) [3], which exploits the low-rank property of the gradient space to enable gradient recycling between model update rounds of federated learning.
- **QSGD** (Quantized SGD) [1], which quantize each component by randomized rounding to a discrete set of values before message transmission. Furthermore, it employs efficient lossless code for quantized gradients, which exploits their statistical properties to generate efficient encodings.
- **DGC** (Deep Gradient Compression) [47], which compresses the gradient with momentum correction and local gradient clipping on top of the gradient sparsification. What's more, to overcome the staleness problem caused by reduced communication, it also uses momentum factor masking and warmup training.

## C.2 Parameter Settings

For each competing algorithm, different hyper-parameters need to be tuned. We provide two or three candidates for each hyper-parameter and perform grid search on all the possible combinations based on the accuracy performance on the validation dataset. The tuning hyper-parameter and their corresponding candidates for each algorithm are listed as follows.

- **FedAvg**: local learning rate: $\{0.05, 0.1, 0.5\}$, rounds for local update: $\{1, 5\}$.

- **Local**: local learning rate: $\{0.05, 0.1, 0.5\}$.

- **Ditto**: local learning rate: $\{0.05, 0.1, 0.5\}$, personalization model learning rate: $\{0.01, 0.05, 0.1\}$, $\lambda : \{0.1, 1, 10\}$, local computation rounds $R : \{1, 5\}$.

- **LG-FedAvg**: local learning rate: $\{0.05, 0.1, 0.5\}$, rounds for local update: $\{1, 5\}$.

- **RSA**: local learning rate: $\{0.05, 0.1, 0.5\}$, personalization model learning rate: $\{0.01, 0.05, 0.1\}$, $\lambda : \{0.1, 1, 10\}$, local computation rounds $R : \{1, 5\}$.

- **pFedMe**: local learning rate: $\{0.05, 0.1, 0.5\}$, personalization model learning rate: $\{0.01, 0.05, 0.1\}$, $\lambda : \{0.1, 1, 10\}$, local computation rounds $R : \{1, 5\}$.

- **Per-FedAvg**: local learning rate: $\{0.05, 0.1, 0.5\}$, personalization model learning rate: $\{0.01, 0.05, 0.1\}$.

- **Sketch**: local learning rate: $\{0.05, 0.1, 0.5\}$, columns of the sketch: $\{0.02, 0.05\} \times$ dimension of the full model, rows of the sketch: $\{0.005, 0.01\} \times$ dimension of the full model, $k$ for the recovered $k$-sparse gradient: $\{0.01, 0.05\} \times$ dimension of the full model.

- **lp-proj-1**: local learning rate: $\{0.05, 0.1, 0.5\}$, personalization model learning rate: $\{0.01, 0.05, 0.1\}$, $\lambda : \{0.1, 1, 10\}$, local computation rounds $R : \{1, 5\}$.

- **lp-proj-2**: local learning rate: $\{0.05, 0.1, 0.5\}$, personalization model learning rate: $\{0.01, 0.05, 0.1\}$, $\lambda : \{0.1, 1, 10\}$, local computation rounds $R : \{1, 5\}$.

- **LBGM**: learning rate: $\{0.05, 0.1, 0.5\}$, local computation rounds $R : \{1, 5\}$, look-back phase (LBP) error threshold $\delta^{\text{thre}} : \{0.2, 0.5, 0.8\}$.

- **QSGD**: learning rate: $\{0.05, 0.1, 0.5\}$, quantization level: $\{5, 10, 15\}$, bucket size: $\{500, 1000, 2000\}$.

- **DGC**: learning rate: $\{0.05, 0.1, 0.5\}$, initial sparsity level: $\{0.25, 0.5, 0.75\}$, sparsity rising level during warm-up training: $\{0.75, 0.5, 0.25\}$.

Other parameters shared by all algorithms:

- # of clients particiate in each communication: $10\% \times$ total # of clients.

- Accuracy level $\nu$ for inner loop for personalization methods: $10^{-10}$.

- Batch size for local SGD: 64.

- Projection dimension $d_{\text{sub}}$ for `lp-proj-1` and `lp-proj-2`: `Synthetic`: 21, `EMNIST`: 80, `CIFAR`: 60, `MNIST`: 50, `FASHIONMNIST`: 600. The projection dimension for each dataset and each model is determined by the full model size and communication budget, and we show theoretically (Lemma 1) that the accuracy performance only has mild dependence on the projection dimension.

- # of repeated experiments: 10.

## C.3 Complete Results on Personalization and Fairness Performance

Table 3 shows complete results on personalization performance in terms of train loss and test accuracy and performance fairness in terms of variance of the above two metrics. Figure 5 displays the training loss and test accuracy evolution as the training proceeds. We can see that `lp-proj-1` and `lp-proj-2` own better performance with lower train loss, higher test accuracy and lower variance across clients.

**Table 3:** Complete Result on Personalization and Fairness Performance in terms of Tran Loss and Test Accuracy.

| Dataset | method | Train Loss | Train Loss Var | Test Acc | Test Acc Var |
|---|---|---|---|---|---|
| Synthetic(0, 0) | Ditto | $0.3500 \pm 0.0038$ | $0.0780 \pm 0.0020$ | $0.8569 \pm 0.0012$ | $0.0178 \pm 0.0005$ |
| | pFedMe | $0.3542 \pm 0.0013$ | $0.0785 \pm 0.0009$ | $0.8580 \pm 0.0015$ | $0.0178 \pm 0.0007$ |
| | Per-fedavg | $0.6986 \pm 0.0184$ | $0.2106 \pm 0.0085$ | $0.7977 \pm 0.0010$ | $0.0410 \pm 0.0006$ |
| | FedAvg | $0.7988 \pm 0.0114$ | $0.2815 \pm 0.0112$ | $0.7714 \pm 0.0010$ | $0.0455 \pm 0.0023$ |
| | local | $0.2522 \pm 0.0045$ | $0.0451 \pm 0.0016$ | $0.8665 \pm 0.0016$ | $0.0159 \pm 0.0007$ |
| | lp-proj-1 | $\mathbf{0.0769} \pm 0.0097$ | $\mathbf{0.0048} \pm 0.0012$ | $\mathbf{0.8868} \pm 0.0010$ | $0.0106 \pm 0.0003$ |
| | lp-proj-2 | $0.0818 \pm 0.0041$ | $0.0053 \pm 0.0005$ | $0.8867 \pm 0.0013$ | $\mathbf{0.0105} \pm 0.0003$ |
| | RSA | $0.5319 \pm 0.0075$ | $0.1466 \pm 0.0034$ | $0.8314 \pm 0.0019$ | $0.0265 \pm 0.0008$ |
| Synthetic(1, 1) | Ditto | $0.3431 \pm 0.0165$ | $0.1596 \pm 0.0488$ | $0.8615 \pm 0.0011$ | $0.0193 \pm 0.0006$ |
| | pFedMe | $0.3010 \pm 0.0029$ | $0.0660 \pm 0.0014$ | $0.8666 \pm 0.0008$ | $0.0170 \pm 0.0004$ |
| | Per-fedavg | $0.6015 \pm 0.0151$ | $0.2194 \pm 0.0193$ | $0.7925 \pm 0.0046$ | $0.0465 \pm 0.0022$ |
| | FedAvg | $0.6938 \pm 0.0147$ | $0.3392 \pm 0.0214$ | $0.7875 \pm 0.0025$ | $0.0480 \pm 0.0023$ |
| | local | $0.2969 \pm 0.0157$ | $0.1283 \pm 0.0439$ | $0.8675 \pm 0.0018$ | $0.0177 \pm 0.0007$ |
| | lp-proj-1 | $\mathbf{0.0614} \pm 0.0143$ | $0.0162 \pm 0.0191$ | $\mathbf{0.8954} \pm 0.0019$ | $\mathbf{0.0123} \pm 0.0008$ |
| | lp-proj-2 | $0.0679 \pm 0.0068$ | $\mathbf{0.0074} \pm 0.0042$ | $0.8932 \pm 0.0018$ | $0.0125 \pm 0.0009$ |
| | RSA | $0.4547 \pm 0.0075$ | $0.1271 \pm 0.0032$ | $0.8416 \pm 0.0015$ | $0.0242 \pm 0.0009$ |
| EMNIST | Ditto | $0.2499 \pm 0.0032$ | $0.0066 \pm 0.0001$ | $\mathbf{0.9089} \pm 0.0008$ | $\mathbf{0.0016} \pm 0.0001$ |
| | pFedMe | $0.4397 \pm 0.0062$ | $0.0301 \pm 0.0092$ | $0.8556 \pm 0.0012$ | $0.0035 \pm 0.0004$ |
| | Per-fedavg | $0.9061 \pm 0.0882$ | $2.1828 \pm 2.7274$ | $0.7944 \pm 0.0083$ | $0.0104 \pm 0.0011$ |
| | FedAvg | $0.7219 \pm 0.0119$ | $0.0300 \pm 0.0036$ | $0.7713 \pm 0.0029$ | $0.0070 \pm 0.0004$ |
| | local | $0.3903 \pm 0.0013$ | $0.0110 \pm 0.0017$ | $0.8566 \pm 0.0008$ | $0.0022 \pm 0.0001$ |
| | lp-proj-1 | $\mathbf{0.0389} \pm 0.0036$ | $\mathbf{0.0039} \pm 0.0003$ | $0.9067 \pm 0.0003$ | $0.0017 \pm 0.0001$ |
| | lp-proj-2 | $0.0448 \pm 0.0022$ | $\mathbf{0.0039} \pm 0.0002$ | $0.9070 \pm 0.0001$ | $0.0017 \pm 0.0000$ |
| | RSA | $0.2740 \pm 0.0054$ | $0.0066 \pm 0.0004$ | $0.8714 \pm 0.0011$ | $0.0019 \pm 0.0001$ |
| | LG-FedAvg | $0.4500 \pm 0.0194$ | $0.1624 \pm 0.0691$ | $0.8453 \pm 0.0042$ | $0.0089 \pm 0.0015$ |
| CIFAR | Ditto | $0.1463 \pm 0.0335$ | $0.0232 \pm 0.0128$ | $0.7909 \pm 0.0084$ | $0.0110 \pm 0.0008$ |
| | pFedMe | $0.1837 \pm 0.0262$ | $0.0311 \pm 0.0072$ | $0.7913 \pm 0.0034$ | $0.0100 \pm 0.0006$ |
| | Per-fedavg | $1.0378 \pm 0.1614$ | $0.8320 \pm 1.0412$ | $0.7257 \pm 0.0220$ | $0.0183 \pm 0.0022$ |
| | FedAvg | $1.4739 \pm 0.0198$ | $0.0438 \pm 0.0092$ | $0.4594 \pm 0.0091$ | $0.0173 \pm 0.0026$ |
| | local | $0.3101 \pm 0.0098$ | $0.0409 \pm 0.0021$ | $0.7688 \pm 0.0026$ | $0.0131 \pm 0.0006$ |
| | lp-proj-1 | $0.0381 \pm 0.0296$ | $0.0077 \pm 0.0066$ | $\mathbf{0.7922} \pm 0.0017$ | $\mathbf{0.0097} \pm 0.0003$ |
| | lp-proj-2 | $\mathbf{0.0043} \pm 0.0105$ | $0.0009 \pm 0.0024$ | $0.7910 \pm 0.0015$ | $0.0099 \pm 0.0005$ |
| | RSA | $0.0073 \pm 0.0015$ | $\mathbf{0.0000} \pm 0.0000$ | $0.7768 \pm 0.0048$ | $\mathbf{0.0097} \pm 0.0008$ |
| | LG-FedAvg | $0.4231 \pm 0.0182$ | $0.0352 \pm 0.0028$ | $0.7523 \pm 0.0055$ | $0.0134 \pm 0.0009$ |
| MNIST | Ditto | $0.0266 \pm 0.0010$ | $0.0001 \pm 0.0000$ | $\mathbf{0.9863} \pm 0.0004$ | $\mathbf{0.0003} \pm 0.0000$ |
| | pFedMe | $0.0511 \pm 0.0037$ | $0.0006 \pm 0.0001$ | $0.9824 \pm 0.0005$ | $0.0005 \pm 0.0000$ |
| | Per-fedavg | $0.0555 \pm 0.0011$ | $0.0010 \pm 0.0004$ | $0.9831 \pm 0.0005$ | $0.0004 \pm 0.0000$ |
| | FedAvg | $0.2099 \pm 0.0013$ | $0.0029 \pm 0.0002$ | $0.9416 \pm 0.0009$ | $0.0015 \pm 0.0001$ |
| | local | $0.0204 \pm 0.0065$ | $0.0002 \pm 0.0001$ | $0.9822 \pm 0.0001$ | $0.0004 \pm 0.0000$ |
| | lp-proj-1 | $0.0101 \pm 0.0046$ | $\mathbf{0.0000} \pm 0.0000$ | $0.9822 \pm 0.0002$ | $0.0004 \pm 0.0000$ |
| | lp-proj-2 | $\mathbf{0.0060} \pm 0.0052$ | $\mathbf{0.0000} \pm 0.0000$ | $0.9825 \pm 0.0002$ | $0.0004 \pm 0.0000$ |
| | RSA | $0.0829 \pm 0.0032$ | $0.0010 \pm 0.0001$ | $0.9809 \pm 0.0002$ | $0.0005 \pm 0.0000$ |
| | LG-FedAvg | $0.0156 \pm 0.0019$ | $0.0001 \pm 0.0000$ | $0.9821 \pm 0.0003$ | $0.0004 \pm 0.0000$ |
| FASHIONMNIST | Ditto | $0.0141 \pm 0.0016$ | $0.0004 \pm 0.0005$ | $\mathbf{0.9770} \pm 0.0004$ | $\mathbf{0.0019} \pm 0.0001$ |
| | pFedMe | $0.0076 \pm 0.0012$ | $0.0001 \pm 0.0001$ | $0.9729 \pm 0.0004$ | $0.0024 \pm 0.0001$ |
| | Per-fedavg | $0.1834 \pm 0.0383$ | $0.3004 \pm 0.1443$ | $0.9500 \pm 0.0041$ | $0.0092 \pm 0.0013$ |
| | FedAvg | $0.1129 \pm 0.0109$ | $0.0194 \pm 0.0042$ | $0.9694 \pm 0.0021$ | $0.0029 \pm 0.0006$ |
| | local | $0.0020 \pm 0.0016$ | $0.0001 \pm 0.0002$ | $0.9748 \pm 0.0008$ | $0.0021 \pm 0.0001$ |
| | lp-proj-1 | $\mathbf{0.0002} \pm 0.0004$ | $\mathbf{0.0000} \pm 0.0000$ | $0.9752 \pm 0.0008$ | $0.0022 \pm 0.0002$ |
| | lp-proj-2 | $0.0004 \pm 0.0005$ | $\mathbf{0.0000} \pm 0.0001$ | $0.9749 \pm 0.0007$ | $0.0021 \pm 0.0002$ |
| | RSA | $0.0908 \pm 0.0368$ | $0.0046 \pm 0.0035$ | $0.9605 \pm 0.0033$ | $0.0039 \pm 0.0012$ |
| | LG-FedAvg | $0.0038 \pm 0.0037$ | $0.0001 \pm 0.0001$ | $0.9738 \pm 0.0005$ | $0.0021 \pm 0.0001$ |

## C.4   Complete Results on Communication Efficiency

Table 4 shows complete results on communication performance in terms of test accuracy and communication bytes. From the comparison result, we can see that, given a communication budget of bytes, `lp-proj-1` and `lp-proj-2` achieve the highest test accuracy. On the other hand, given a target test accuracy, these two approaches need the least bytes for communication, and the compression rate could be up to `1000x`.

**Table 4:** Complete Result on Communication Performance in terms of Test Accuracy and Communication Bytes. There are two comparisons: one is test accuracy on a given byte budget, the other is used bytes to achieve a target test accuracy. Under the given bytes budget, a ⋆ on the column "Test Acc" refers to the situation that bytes used in the first iteration of the corresponding algorithm have exceeded the budget. Under target test accuracy, a ⋆ on the column "Used Bytes" means the algorithm could not provide a solution that reaches the target accuracy.

| Dataset | Method | Bytes Budget | Test Acc | Target Acc | Used Bytes |
|---|---|---|---|---|---|
| Synthetic(0, 0) | FedAvg | 328020 | $0.625 \pm 0.006$ | 0.6 | $597800 \pm 0$ |
| | Sketch | 328020 | $0.456 \pm 0.020$ | 0.6 | $\star \pm \star$ |
| | lp-proj-1 | 328020 | $0.885 \pm 0.002$ | 0.6 | $\mathbf{4620} \pm 0$ |
| | lp-proj-2 | 328020 | $\mathbf{0.888} \pm 0.001$ | 0.6 | $\mathbf{4620} \pm 0$ |
| | LBGM | 328020 | $0.538 \pm 0.007$ | 0.6 | $365002 \pm 23578$ |
| | QSGD | 328020 | $0.115 \pm 0.069$ | 0.6 | $923350 \pm 174383$ |
| | DGC | 328020 | $\star \pm \star$ | 0.6 | $391800 \pm 186123$ |
| Synthetic(1, 1) | FedAvg | 401940 | $0.516 \pm 0.028$ | 0.6 | $523380 \pm 34268$ |
| | Sketch | 401940 | $0.359 \pm 0.017$ | 0.6 | $\star \pm \star$ |
| | lp-proj-1 | 401940 | $\mathbf{0.892} \pm 0.002$ | 0.6 | $\mathbf{4620} \pm 0$ |
| | lp-proj-2 | 401940 | $0.888 \pm 0.001$ | 0.6 | $\mathbf{4620} \pm 0$ |
| | LBGM | 401940 | $0.331 \pm 0.042$ | 0.6 | $1153406 \pm 371717$ |
| | QSGD | 401940 | $0.582 \pm 0.018$ | 0.6 | $2607290 \pm 905787$ |
| | DGC | 401940 | $0.114 \pm 0.208$ | 0.6 | $61500 \pm 184500$ |
| EMNIST | FedAvg | 4236900 | $\star \pm \star$ | 0.7 | $445851400 \pm 16265444$ |
| | Sketch | 4236900 | $\star \pm \star$ | 0.7 | $\star \pm \star$ |
| | lp-proj-1 | 4236900 | $\mathbf{0.906} \pm 0.000$ | 0.7 | $\mathbf{174720} \pm 10699$ |
| | lp-proj-2 | 4236900 | $\mathbf{0.906} \pm 0.000$ | 0.7 | $196560 \pm 6552$ |
| | LG-FedAvg | 4236900 | $0.071 \pm 0.016$ | 0.7 | $230786010 \pm 6629787$ |
| | LBGM | 4236900 | $\star \pm \star$ | 0.7 | $769902776 \pm 37552057$ |
| | QSGD | 4236900 | $\star \pm \star$ | 0.7 | $673302175 \pm 101397671$ |
| | DGC | 4236900 | $\star \pm \star$ | 0.7 | $\star \pm \star$ |
| CIFAR | FedAvg | 1029600 | $\star \pm \star$ | 0.4 | $392870016 \pm 33519046$ |
| | Sketch | 1029600 | $\star \pm \star$ | 0.4 | $\star \pm \star$ |
| | lp-proj-1 | 1029600 | $\mathbf{0.792} \pm 0.002$ | 0.4 | $\mathbf{26400} \pm 0$ |
| | lp-proj-2 | 1029600 | $0.790 \pm 0.002$ | 0.4 | $\mathbf{26400} \pm 0$ |
| | LG-FedAvg | 1029600 | $\star \pm \star$ | 0.4 | $51369296 \pm 10550837$ |
| | LBGM | 1029600 | $\star \pm \star$ | 0.4 | $476274601 \pm 41680525$ |
| | QSGD | 1029600 | $\star \pm \star$ | 0.4 | $72535230 \pm 24632864$ |
| | DGC | 1029600 | $\star \pm \star$ | 0.4 | $37640880 \pm 323640$ |
| MNIST | FedAvg | 228000 | $\star \pm \star$ | 0.7 | $56293080 \pm 6828608$ |
| | Sketch | 228000 | $\star \pm \star$ | 0.7 | $531338200 \pm 51486427$ |
| | lp-proj-1 | 228000 | $\mathbf{0.982} \pm 0.000$ | 0.7 | $\mathbf{12000} \pm 0$ |
| | lp-proj-2 | 228000 | $\mathbf{0.982} \pm 0.000$ | 0.7 | $\mathbf{12000} \pm 0$ |
| | LG-FedAvg | 228000 | $0.111 \pm 0.026$ | 0.7 | $763560 \pm 55540$ |
| | LBGM | 228000 | $\star \pm \star$ | 0.7 | $28353303 \pm 5492558$ |
| | QSGD | 228000 | $\star \pm \star$ | 0.7 | $16158000 \pm 4693474$ |
| | DGC | 228000 | $\star \pm \star$ | 0.7 | $16170900 \pm 2836077$ |
| FASHIONMNIST | FedAvg | 3384000 | $\star \pm \star$ | 0.7 | $1186531912 \pm 52998121$ |
| | lp-proj-1 | 3384000 | $\mathbf{0.975} \pm 0.001$ | 0.7 | $\mathbf{144000} \pm 0$ |
| | lp-proj-2 | 3384000 | $\mathbf{0.975} \pm 0.001$ | 0.7 | $\mathbf{144000} \pm 0$ |
| | LG-FedAvg | 3384000 | $0.892 \pm 0.010$ | 0.7 | $1725336 \pm 118790$ |
| | LBGM | 3384000 | $\star \pm \star$ | 0.7 | $1409583679 \pm 120013474$ |
| | QSGD | 3384000 | $\star \pm \star$ | 0.7 | $565519560 \pm 252785764$ |
| | DGC | 3384000 | $\star \pm \star$ | 0.7 | $135446730 \pm 165916659$ |

## C.5 Complete Results on Robustness

Complete results for different methods under various kinds and various levels of Byzantine attacks are shown in Table 5, 6 and 7, and complete results under data poison attack is shown in Table 8. `lp-proj-1` and `lp-proj-2` show stable performance and is the most robust to different attacks.

## C.6 Complete Results on Accuracy and Performance Fairness Trade-off

Figure 6 shows complete results for accuracy and performance fairness trade-off on all the datasets used for numerical experiments. `lp-proj-1` and `lp-proj-2` are comparable to other state-of-the-art methods.

**Table 5:** Complete Result on Robustness Performance in terms of test accuracy under same-value attacks. (The number in the parentheses is the corresponding variance.)

| Dataset | Method | Clean | 10% | 20% | 50% | 80% |
|---|---|---|---|---|---|---|
| Synthetic(0, 0) | Ditto | 0.857 (0.018) | 0.856 (0.017) | 0.851 (0.020) | 0.855 (0.017) | 0.837 (0.014) |
| | Global+Mean | 0.772 (0.044) | 0.558 (0.150) | 0.485 (0.161) | 0.462 (0.169) | 0.446 (0.166) |
| | Global+Median | 0.519 (0.140) | 0.558 (0.129) | 0.604 (0.121) | 0.434 (0.156) | 0.471 (0.155) |
| | Global+Krum | 0.235 (0.109) | 0.285 (0.127) | 0.318 (0.133) | 0.298 (0.131) | 0.285 (0.122) |
| | RSA | 0.832 (0.026) | **0.881 (0.011)** | 0.879 (**0.011**) | **0.885 (0.012)** | 0.863 (0.008) |
| | lp-proj-1 | **0.888 (0.010)** | 0.868 (0.013) | **0.880 (0.011)** | 0.884 (**0.012**) | **0.869 (0.010)** |
| | lp-proj-2 | 0.887 (**0.010**) | 0.865 (0.014) | 0.873 (0.012) | 0.875 (0.014) | 0.858 (0.012) |
| Synthetic(1, 1) | Ditto | 0.863 (0.018) | 0.882 (0.015) | 0.884 (0.014) | 0.885 (0.012) | 0.873 (0.012) |
| | Global+Mean | 0.785 (0.051) | 0.481 (0.168) | 0.440 (0.175) | 0.387 (0.171) | 0.477 (0.147) |
| | Global+Median | 0.525 (0.142) | 0.606 (0.124) | 0.655 (0.122) | 0.428 (0.163) | 0.484 (0.166) |
| | Global+Krum | 0.224 (0.105) | 0.294 (0.135) | 0.310 (0.139) | 0.396 (0.143) | 0.241 (0.149) |
| | RSA | 0.844 (0.023) | **0.901 (0.013)** | 0.903 (0.012) | 0.906 (**0.010**) | **0.916 (0.005)** |
| | lp-proj-1 | **0.893** (0.014) | 0.890 (0.014) | **0.907 (0.010)** | **0.908 (0.010)** | 0.914 (**0.005**) |
| | lp-proj-2 | 0.891 (**0.013**) | 0.890 (0.015) | 0.905 (0.011) | 0.898 (0.013) | 0.907 (0.007) |
| EMNIST | Ditto | **0.907 (0.002)** | 0.293 (0.004) | 0.294 (0.004) | 0.289 (0.004) | 0.282 (0.003) |
| | Global+Mean | 0.770 (0.007) | 0.057 (0.013) | 0.058 (0.013) | 0.061 (0.013) | 0.072 (0.015) |
| | Global+Median | 0.556 (0.015) | 0.057 (0.013) | 0.058 (0.013) | 0.061 (0.013) | 0.072 (0.015) |
| | Global+Krum | 0.504 (0.032) | 0.057 (0.013) | 0.058 (0.013) | 0.061 (0.013) | 0.072 (0.015) |
| | RSA | 0.872 (**0.002**) | 0.293 (0.004) | 0.294 (0.004) | 0.337 (0.012) | 0.431 (0.021) |
| | lp-proj-1 | 0.906 (**0.002**) | **0.908 (0.002)** | **0.905 (0.002)** | **0.908 (0.002)** | **0.908 (0.002)** |
| | lp-proj-2 | **0.907 (0.002)** | 0.900 (**0.002**) | 0.902 (**0.002**) | 0.904 (**0.002**) | 0.907 (**0.002**) |
| CIFAR | Ditto | **0.796** (0.010) | 0.501 (**0.000**) | 0.502 (**0.001**) | 0.502 (**0.001**) | 0.511 (**0.002**) |
| | Global+Mean | 0.456 (0.022) | 0.106 (0.042) | 0.116 (0.044) | 0.115 (0.044) | 0.150 (0.052) |
| | Global+Median | 0.247 (0.035) | 0.106 (0.042) | 0.116 (0.044) | 0.115 (0.044) | 0.150 (0.052) |
| | Global+Krum | 0.246 (0.038) | 0.106 (0.042) | 0.116 (0.044) | 0.115 (0.044) | 0.150 (0.052) |
| | RSA | 0.775 (0.010) | 0.539 (0.008) | 0.574 (0.012) | 0.590 (0.016) | 0.595 (0.013) |
| | lp-proj-1 | 0.791 (**0.009**) | **0.786 (0.009)** | **0.790 (0.009)** | **0.797 (0.010)** | **0.795 (0.012)** |
| | lp-proj-2 | 0.792 (**0.009**) | 0.783 (0.009) | 0.789 (0.010) | 0.791 (0.012) | 0.788 (0.011) |
| MNIST | Ditto | **0.986 (0.000)** | 0.529 (0.011) | 0.516 (0.009) | 0.958 (0.005) | 0.939 (0.005) |
| | Global+Mean | 0.942 (0.001) | 0.107 (0.042) | 0.120 (0.046) | 0.113 (0.046) | 0.198 (0.055) |
| | Global+Median | 0.808 (0.014) | 0.861 (0.006) | 0.120 (0.046) | 0.113 (0.046) | 0.175 (0.057) |
| | Global+Krum | 0.647 (0.062) | 0.668 (0.080) | 0.120 (0.046) | 0.113 (0.046) | 0.168 (0.061) |
| | RSA | 0.981 (0.001) | 0.980 (**0.000**) | 0.981 (0.001) | **0.984 (0.000)** | 0.985 (**0.000**) |
| | lp-proj-1 | 0.982 (**0.000**) | 0.982 (**0.000**) | **0.982 (0.000)** | **0.984 (0.000)** | 0.987 (**0.000**) |
| | lp-proj-2 | 0.982 (**0.000**) | **0.983** (0.001) | **0.982 (0.000)** | **0.984 (0.000)** | **0.989 (0.000)** |
| FASHIONMNIST | Ditto | **0.977 (0.002)** | 0.605 (0.020) | 0.611 (0.018) | 0.630 (0.016) | 0.615 (0.013) |
| | Global+Mean | 0.967 (0.004) | 0.130 (0.047) | 0.151 (0.052) | 0.153 (0.048) | 0.176 (0.057) |
| | Global+Median | 0.729 (0.033) | 0.739 (0.024) | 0.119 (0.045) | 0.120 (0.046) | 0.162 (0.044) |
| | Global+Krum | 0.374 (0.072) | 0.413 (0.122) | 0.119 (0.045) | 0.140 (0.050) | 0.192 (0.062) |
| | RSA | 0.960 (0.004) | 0.685 (0.033) | 0.767 (0.036) | 0.775 (0.022) | 0.827 (0.018) |
| | lp-proj-1 | 0.975 (**0.002**) | **0.973 (0.002)** | **0.980 (0.001)** | **0.976 (0.002)** | **0.976 (0.002)** |
| | lp-proj-2 | 0.974 (**0.002**) | 0.971 (0.003) | 0.979 (**0.001**) | 0.975 (**0.002**) | 0.975 (**0.002**) |

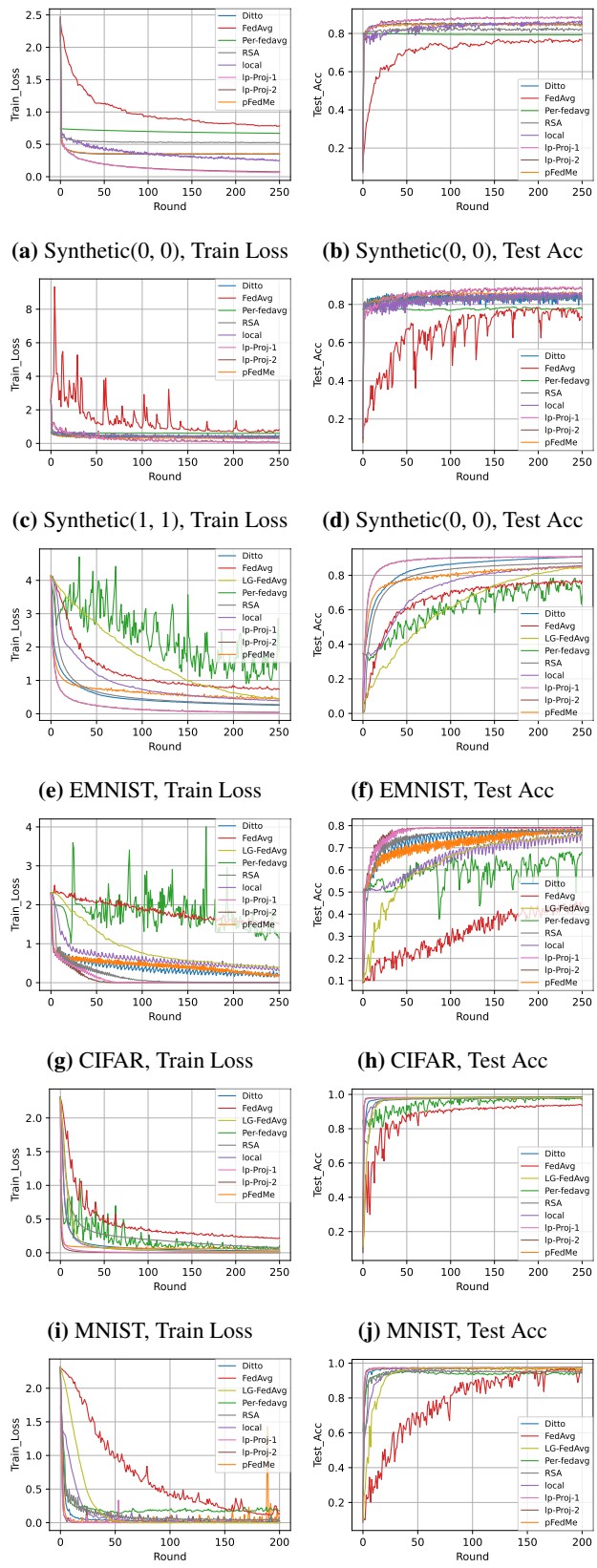

**(a)** Synthetic(0, 0), Train Loss    **(b)** Synthetic(0, 0), Test Acc

**(c)** Synthetic(1, 1), Train Loss    **(d)** Synthetic(0, 0), Test Acc

**(e)** EMNIST, Train Loss    **(f)** EMNIST, Test Acc

**(g)** CIFAR, Train Loss    **(h)** CIFAR, Test Acc

**(i)** MNIST, Train Loss    **(j)** MNIST, Test Acc

**(k)** FASHIONMNIST, Train Loss **(l)** FASHIONMNIST, Test Acc

**Figure 5:** Personalization performance of competing methods.

**Table 6:** Complete Result on Robustness Performance in terms of test accuracy under sign-flipping attacks. (The number in the parentheses is the corresponding variance.) A ⋆ on the cell means that the corresponding algorithm would collapse under the given intensity of adversarial attack and could not return a solution.

| Dataset | Method | Clean | 10% | 20% | 50% | 80% |
|---|---|---|---|---|---|---|
| Synthetic(0, 0) | Ditto | 0.857 (0.018) | 0.853 (0.017) | 0.850 (0.017) | ⋆ (⋆) | ⋆ (⋆) |
| | Global+Mean | 0.772 (0.044) | 0.412 (0.146) | 0.324 (0.130) | ⋆ (⋆) | ⋆ (⋆) |
| | Global+Median | 0.519 (0.140) | 0.421 (0.131) | 0.425 (0.139) | 0.192 (0.087) | ⋆ (⋆) |
| | Global+Krum | 0.280 (0.133) | 0.304 (0.139) | 0.308 (0.157) | 0.281 (0.134) | ⋆ (⋆) |
| | RSA | 0.832 (0.026) | 0.829 (0.025) | 0.834 (0.022) | 0.881 (0.012) | 0.848 (0.011) |
| | lp-proj-1 | **0.888 (0.010)** | 0.884 (0.011) | **0.885 (0.010)** | **0.885 (0.010)** | **0.863 (0.010)** |
| | lp-proj-2 | 0.887 (**0.010**) | **0.885 (0.010)** | 0.884 (**0.010**) | ⋆ (⋆) | ⋆ (⋆) |
| Synthetic(1, 1) | Ditto | 0.863 (0.018) | 0.879 (0.015) | 0.884 (0.013) | ⋆ (⋆) | ⋆ (⋆) |
| | Global+Mean | 0.785 (0.051) | 0.292 (0.142) | 0.243 (0.134) | ⋆ (⋆) | ⋆ (⋆) |
| | Global+Median | 0.525 (0.142) | 0.455 (0.140) | 0.436 (0.169) | 0.168 (0.092) | ⋆ (⋆) |
| | Global+Krum | 0.269 (0.134) | 0.287 (0.142) | 0.326 (0.141) | 0.372 (0.155) | ⋆ (⋆) |
| | RSA | 0.844 (0.023) | 0.856 (0.023) | 0.863 (0.020) | **0.905 (0.010)** | **0.920 (0.004)** |
| | lp-proj-1 | **0.893 (0.014)** | **0.905 (0.011)** | **0.909 (0.010)** | **0.905 (0.009)** | 0.918 (0.005) |
| | lp-proj-2 | 0.891 (**0.013**) | 0.902 (0.013) | 0.908 (0.011) | ⋆ (⋆) | ⋆ (⋆) |
| EMNIST | Ditto | **0.907 (0.002)** | 0.746 (0.004) | 0.748 (0.003) | ⋆ (⋆) | ⋆ (⋆) |
| | Global+Mean | 0.770 (0.007) | 0.057 (0.013) | 0.072 (0.012) | ⋆ (⋆) | ⋆ (⋆) |
| | Global+Median | 0.556 (0.015) | 0.382 (0.021) | 0.392 (0.024) | 0.107 (0.005) | ⋆ (⋆) |
| | Global+Krum | 0.501 (0.037) | 0.452 (0.029) | 0.409 (0.031) | 0.495 (0.035) | ⋆ (⋆) |
| | RSA | 0.872 (**0.002**) | 0.501 (0.007) | 0.598 (0.006) | **0.905 (0.002)** | **0.907 (0.002)** |
| | lp-proj-1 | 0.906 (**0.002**) | **0.908 (0.002)** | **0.910 (0.002)** | **0.905 (0.002)** | **0.907 (0.002)** |
| | lp-proj-2 | **0.907 (0.002)** | 0.907 (**0.002**) | 0.907 (**0.002**) | ⋆ (⋆) | ⋆ (⋆) |
| CIFAR | Ditto | **0.795** (0.010) | 0.746 (0.016) | 0.762 (0.015) | ⋆ (⋆) | ⋆ (⋆) |
| | Global+Mean | 0.456 (0.022) | 0.106 (0.042) | 0.128 (0.029) | ⋆ (⋆) | ⋆ (⋆) |
| | Global+Median | 0.247 (0.035) | 0.265 (0.039) | 0.224 (0.018) | ⋆ (⋆) | ⋆ (⋆) |
| | Global+Krum | 0.246 (0.038) | 0.240 (0.038) | 0.290 (0.019) | 0.200 (0.059) | ⋆ (⋆) |
| | RSA | 0.778 (**0.009**) | 0.646 (0.010) | 0.613 (0.013) | 0.788 (**0.010**) | **0.791 (0.010)** |
| | lp-proj-1 | 0.790 (**0.009**) | **0.795** (0.010) | **0.793 (0.009)** | **0.801 (0.010)** | 0.788 (0.011) |
| | lp-proj-2 | 0.792 (**0.009**) | 0.788 (**0.009**) | 0.786 (0.010) | ⋆ (⋆) | ⋆ (⋆) |
| MNIST | Ditto | **0.986 (0.000)** | 0.981 (**0.000**) | 0.981 (**0.000**) | ⋆ (⋆) | ⋆ (⋆) |
| | Global+Mean | 0.942 (0.001) | 0.188 (0.057) | 0.296 (0.061) | ⋆ (⋆) | ⋆ (⋆) |
| | Global+Median | 0.859 (0.007) | 0.103 (0.041) | 0.817 (0.008) | 0.206 (0.032) | ⋆ (⋆) |
| | Global+Krum | 0.679 (0.076) | 0.668 (0.080) | 0.723 (0.045) | 0.796 (0.029) | ⋆ (⋆) |
| | RSA | 0.981 (0.001) | 0.954 (0.006) | 0.976 (0.001) | **0.984 (0.000)** | **0.984 (0.000)** |
| | lp-proj-1 | 0.982 (**0.000**) | **0.982 (0.000)** | **0.982 (0.000)** | **0.984 (0.000)** | **0.984 (0.000)** |
| | lp-proj-2 | 0.982 (**0.000**) | **0.982 (0.000)** | **0.982 (0.000)** | ⋆ (⋆) | ⋆ (⋆) |
| FASHIONMNIST | Ditto | **0.977 (0.002)** | 0.973 (**0.002**) | 0.980 (**0.001**) | ⋆ (⋆) | ⋆ (⋆) |
| | Global+Mean | 0.967 (0.004) | 0.111 (0.043) | 0.119 (0.045) | ⋆ (⋆) | ⋆ (⋆) |
| | Global+Median | 0.729 (0.033) | 0.214 (0.031) | 0.537 (0.038) | ⋆ (⋆) | ⋆ (⋆) |
| | Global+Krum | 0.374 (0.072) | 0.430 (0.069) | 0.511 (0.107) | 0.728 (0.065) | ⋆ (⋆) |
| | RSA | 0.960 (0.004) | 0.891 (0.021) | 0.930 (0.011) | 0.974 (0.002) | **0.977 (0.002)** |
| | lp-proj-1 | 0.975 (**0.002**) | **0.975 (0.002)** | **0.981 (0.001)** | **0.976 (0.001)** | **0.977 (0.002)** |
| | lp-proj-2 | 0.974 (**0.002**) | 0.974 (**0.002**) | **0.981 (0.001)** | ⋆ (⋆) | ⋆ (⋆) |

**Table 7:** Complete Result on Robustness Performance in terms of test accuracy under Gaussian attacks. (The number in the parentheses is the corresponding variance.)

| Dataset | Method | Clean | 10% | 20% | 50% | 80% |
|---|---|---|---|---|---|---|
| Synthetic(0, 0) | Ditto | 0.857 (0.018) | 0.651 (0.052) | 0.674 (0.056) | 0.722 (0.045) | 0.710 (0.047) |
| | Global+Mean | 0.772 (0.044) | 0.174 (0.081) | 0.173 (0.080) | 0.189 (0.082) | 0.246 (0.102) |
| | Global+Median | 0.519 (0.140) | 0.124 (0.054) | 0.143 (0.071) | 0.189 (0.083) | 0.204 (0.091) |
| | Global+Krum | 0.235 (0.109) | 0.133 (0.072) | 0.148 (0.086) | 0.208 (0.105) | 0.290 (0.086) |
| | RSA | 0.832 (0.026) | 0.845 (0.019) | 0.851 (0.018) | 0.868 (0.017) | 0.837 (0.015) |
| | lp-proj-1 | **0.888** (**0.010**) | **0.876** (**0.013**) | **0.880** (**0.011**) | **0.885** (**0.011**) | **0.862** (**0.009**) |
| | lp-proj-2 | 0.887 (**0.010**) | 0.838 (0.023) | 0.846 (0.020) | 0.861 (0.018) | 0.844 (0.011) |
| Synthetic(1, 1) | Ditto | 0.863 (0.018) | 0.694 (0.060) | 0.741 (0.051) | 0.762 (0.032) | 0.795 (0.024) |
| | Global+Mean | 0.785 (0.051) | 0.194 (0.089) | 0.188 (0.101) | 0.196 (0.090) | 0.295 (0.143) |
| | Global+Median | 0.525 (0.142) | 0.132 (0.059) | 0.124 (0.078) | 0.231 (0.122) | 0.250 (0.144) |
| | Global+Krum | 0.224 (0.105) | 0.157 (0.096) | 0.159 (0.084) | 0.248 (0.116) | 0.268 (0.121) |
| | RSA | 0.844 (0.023) | 0.886 (0.014) | 0.887 (0.014) | 0.885 (0.015) | 0.902 (0.006) |
| | lp-proj-1 | **0.893** (0.014) | **0.898** (**0.013**) | **0.910** (**0.011**) | **0.905** (**0.011**) | **0.916** (**0.005**) |
| | lp-proj-2 | 0.891 (**0.013**) | 0.868 (0.020) | 0.887 (0.016) | 0.878 (0.013) | 0.893 (0.008) |
| EMNIST | Ditto | **0.907** (**0.002**) | 0.649 (0.004) | 0.673 (0.004) | 0.681 (0.006) | 0.703 (0.006) |
| | Global+Mean | 0.770 (0.007) | 0.068 (0.005) | 0.063 (0.005) | 0.060 (0.008) | 0.059 (0.009) |
| | Global+Median | 0.556 (0.015) | 0.061 (**0.001**) | 0.079 (**0.002**) | 0.081 (**0.002**) | 0.062 (0.011) |
| | Global+Krum | 0.504 (0.032) | 0.177 (0.005) | 0.164 (0.009) | 0.181 (0.006) | 0.180 (0.006) |
| | RSA | 0.872 (**0.002**) | 0.782 (0.003) | 0.786 (0.003) | 0.820 (**0.002**) | 0.844 (0.003) |
| | lp-proj-1 | 0.906 (**0.002**) | **0.899** (0.002) | **0.903** (0.002) | **0.905** (**0.002**) | **0.907** (**0.002**) |
| | lp-proj-2 | **0.907** (**0.002**) | 0.862 (0.002) | 0.862 (**0.002**) | 0.881 (**0.002**) | 0.883 (**0.002**) |
| CIFAR | Ditto | **0.796** (0.010) | 0.668 (0.011) | 0.674 (0.011) | 0.658 (0.014) | 0.604 (0.021) |
| | Global+Mean | 0.456 (0.022) | 0.139 (0.010) | 0.151 (0.038) | 0.146 (0.035) | 0.153 (0.033) |
| | Global+Median | 0.247 (0.035) | 0.112 (0.011) | 0.136 (0.024) | 0.159 (0.034) | 0.144 (**0.009**) |
| | Global+Krum | 0.246 (0.038) | 0.160 (**0.008**) | 0.166 (0.013) | 0.156 (**0.007**) | 0.169 (0.017) |
| | RSA | 0.775 (0.010) | 0.731 (0.011) | 0.736 (**0.010**) | 0.757 (0.009) | 0.772 (0.011) |
| | lp-proj-1 | 0.791 (**0.009**) | **0.790** (0.008) | **0.791** (0.010) | **0.797** (0.009) | **0.795** (0.010) |
| | lp-proj-2 | 0.792 (**0.009**) | 0.775 (0.011) | 0.779 (**0.010**) | 0.784 (0.010) | 0.776 (0.011) |
| MNIST | Ditto | **0.986** (**0.000**) | 0.931 (0.002) | 0.928 (0.002) | 0.932 (0.003) | 0.937 (0.002) |
| | Global+Mean | 0.942 (0.001) | 0.460 (0.027) | 0.272 (0.040) | 0.186 (0.027) | 0.210 (0.043) |
| | Global+Median | 0.808 (0.014) | 0.862 (0.006) | 0.141 (0.038) | 0.114 (0.039) | 0.207 (0.062) |
| | Global+Krum | 0.647 (0.062) | 0.669 (0.062) | 0.770 (0.012) | 0.778 (0.013) | 0.821 (0.013) |
| | RSA | 0.981 (0.001) | 0.957 (0.001) | 0.963 (0.001) | 0.979 (0.001) | 0.982 (**0.000**) |
| | lp-proj-1 | 0.982 (**0.000**) | **0.981** (**0.000**) | **0.982** (0.001) | 0.983 (**0.000**) | 0.984 (**0.000**) |
| | lp-proj-2 | 0.982 (**0.000**) | 0.978 (**0.000**) | 0.980 (**0.000**) | **0.984** (**0.000**) | **0.987** (**0.000**) |
| FASHIONMNIST | Ditto | **0.977** (**0.002**) | 0.886 (0.015) | 0.880 (0.014) | 0.873 (0.017) | 0.895 (0.007) |
| | Global+Mean | 0.967 (0.004) | 0.167 (0.041) | 0.165 (0.040) | 0.170 (0.050) | 0.174 (0.034) |
| | Global+Median | 0.729 (0.033) | 0.650 (0.046) | 0.346 (0.018) | 0.192 (0.035) | 0.192 (0.051) |
| | Global+Krum | 0.374 (0.072) | 0.437 (0.117) | 0.468 (0.067) | 0.494 (0.060) | 0.362 (0.072) |
| | RSA | 0.960 (0.004) | 0.947 (0.007) | 0.959 (0.002) | 0.964 (0.002) | 0.969 (**0.002**) |
| | lp-proj-1 | 0.975 (**0.002**) | **0.973** (0.002) | **0.978** (**0.001**) | **0.977** (0.001) | **0.974** (**0.002**) |
| | lp-proj-2 | 0.974 (**0.002**) | 0.964 (0.003) | 0.973 (**0.001**) | 0.968 (0.002) | 0.972 (**0.002**) |

**Table 8:** Complete Result on Robustness Performance in terms of test accuracy under data-poison attacks. (The number in the parentheses is the corresponding variance.)

| Dataset | Method | Clean | 2% | 5% | 10% | 20% |
|---|---|---|---|---|---|---|
| Synthetic(0, 0) | Ditto | 0.857 (0.018) | 0.853 (0.019) | 0.851 (0.019) | 0.749 (0.068) | 0.342 (0.123) |
| | Global+Mean | 0.772 (0.044) | 0.484 (0.163) | 0.304 (0.134) | 0.257 (0.129) | 0.141 (0.059) |
| | Global+Median | 0.519 (0.140) | 0.539 (0.141) | 0.552 (0.130) | 0.322 (0.118) | 0.435 (0.133) |
| | Global+Krum | 0.280 (0.133) | 0.288 (0.130) | 0.290 (0.134) | 0.291 (0.156) | 0.300 (0.146) |
| | RSA | 0.832 (0.026) | 0.850 (0.019) | 0.865 (0.016) | 0.876 (0.012) | 0.875 (**0.011**) |
| | lp-proj-1 | **0.888** (**0.010**) | **0.886** (**0.011**) | **0.884** (**0.011**) | **0.884** (**0.011**) | **0.881** (**0.011**) |
| | lp-proj-2 | 0.887 (**0.010**) | **0.886** (**0.011**) | 0.881 (0.012) | 0.868 (0.015) | 0.866 (0.014) |
| Synthetic(1, 1) | Ditto | 0.863 (0.018) | 0.863 (0.019) | 0.853 (0.023) | 0.810 (0.045) | 0.401 (0.184) |
| | Global+Mean | 0.785 (0.051) | 0.432 (0.156) | 0.253 (0.124) | 0.208 (0.111) | 0.146 (0.090) |
| | Global+Median | 0.525 (0.142) | 0.534 (0.151) | 0.554 (0.144) | 0.274 (0.132) | 0.373 (0.152) |
| | Global+Krum | 0.269 (0.134) | 0.277 (0.129) | 0.300 (0.145) | 0.249 (0.140) | 0.290 (0.163) |
| | RSA | 0.844 (0.023) | 0.864 (0.019) | 0.874 (0.016) | 0.894 (0.013) | 0.903 (0.010) |
| | lp-proj-1 | **0.893** (0.014) | 0.889 (0.013) | **0.901** (0.012) | **0.904** (0.011) | **0.914** (**0.009**) |
| | lp-proj-2 | 0.891 (**0.013**) | **0.896** (**0.012**) | 0.893 (**0.012**) | 0.895 (0.013) | 0.899 (0.013) |
| EMNIST | Ditto | **0.907** (**0.002**) | 0.761 (0.003) | 0.778 (0.005) | 0.859 (**0.002**) | ⋆ (⋆) |
| | Global+Mean | 0.770 (0.007) | 0.179 (0.024) | 0.110 (0.019) | 0.150 (0.019) | ⋆ (⋆) |
| | Global+Median | 0.556 (0.015) | 0.549 (0.013) | 0.564 (0.015) | 0.433 (0.012) | 0.419 (0.011) |
| | Global+Krum | 0.501 (0.037) | 0.454 (0.038) | 0.449 (0.022) | 0.329 (0.030) | 0.321 (0.031) |
| | RSA | 0.872 (**0.002**) | 0.832 (**0.002**) | 0.825 (**0.002**) | 0.871 (**0.002**) | 0.878 (**0.002**) |
| | lp-proj-1 | 0.906 (**0.002**) | **0.909** (**0.002**) | **0.910** (**0.002**) | 0.906 (**0.002**) | 0.905 (**0.002**) |
| | lp-proj-2 | **0.907** (**0.002**) | 0.907 (**0.002**) | 0.907 (**0.002**) | **0.908** (**0.002**) | **0.906** (**0.002**) |
| CIFAR | Ditto | **0.795** (0.010) | 0.750 (0.016) | 0.749 (0.015) | 0.739 (**0.009**) | 0.765 (0.011) |
| | Global+Mean | 0.456 (0.022) | 0.102 (0.041) | 0.139 (0.033) | 0.153 (0.025) | 0.155 (0.038) |
| | Global+Median | 0.247 (0.035) | 0.252 (0.023) | 0.247 (0.025) | 0.292 (0.015) | 0.288 (0.011) |
| | Global+Krum | 0.246 (0.038) | 0.250 (0.045) | 0.250 (0.027) | 0.301 (0.046) | 0.222 (0.019) |
| | RSA | 0.778 (**0.009**) | 0.719 (0.011) | 0.753 (0.010) | 0.739 (0.013) | 0.778 (0.011) |
| | lp-proj-1 | 0.790 (**0.009**) | **0.795** (0.010) | 0.793 (**0.008**) | **0.795** (0.010) | **0.801** (0.010) |
| | lp-proj-2 | 0.792 (**0.009**) | 0.794 (**0.009**) | 0.794 (**0.008**) | 0.786 (**0.009**) | 0.789 (**0.009**) |
| MNIST | Ditto | **0.986** (0.000) | **0.983** (0.000) | **0.982** (0.000) | **0.982** (0.001) | ⋆ (⋆) |
| | Global+Mean | 0.942 (0.001) | 0.832 (0.007) | 0.712 (0.025) | 0.627 (0.047) | 0.514 (0.028) |
| | Global+Median | 0.859 (0.007) | 0.860 (0.006) | 0.862 (0.006) | 0.857 (0.007) | 0.839 (0.006) |
| | Global+Krum | 0.679 (0.076) | 0.697 (0.078) | 0.659 (0.068) | 0.668 (0.080) | 0.697 (0.046) |
| | RSA | 0.981 (0.001) | 0.978 (0.001) | 0.975 (0.001) | 0.979 (0.001) | 0.979 (**0.001**) |
| | lp-proj-1 | 0.982 (**0.000**) | **0.983** (**0.000**) | **0.982** (**0.000**) | **0.982** (**0.000**) | **0.982** (**0.001**) |
| | lp-proj-2 | 0.982 (**0.000**) | 0.982 (**0.000**) | **0.982** (**0.000**) | **0.982** (**0.000**) | **0.982** (**0.001**) |
| FASHIONMNIST | Ditto | **0.977** (0.002) | 0.972 (**0.002**) | **0.973** (0.002) | 0.964 (0.004) | ⋆ (⋆) |
| | Global+Mean | 0.967 (0.004) | 0.208 (0.070) | 0.181 (0.070) | 0.161 (0.055) | ⋆ (⋆) |
| | Global+Median | 0.729 (0.033) | 0.739 (0.036) | 0.720 (0.025) | 0.721 (0.029) | 0.840 (0.037) |
| | Global+Krum | 0.374 (0.072) | 0.480 (0.082) | 0.488 (0.100) | 0.401 (0.072) | 0.581 (0.126) |
| | RSA | 0.960 (0.004) | 0.969 (0.003) | 0.959 (0.004) | 0.965 (0.003) | 0.976 (0.002) |
| | lp-proj-1 | 0.975 (**0.002**) | **0.974** (**0.002**) | **0.973** (**0.002**) | 0.974 (**0.002**) | 0.980 (**0.001**) |
| | lp-proj-2 | 0.974 (**0.002**) | 0.973 (**0.002**) | **0.973** (**0.002**) | **0.975** (**0.002**) | **0.981** (**0.001**) |

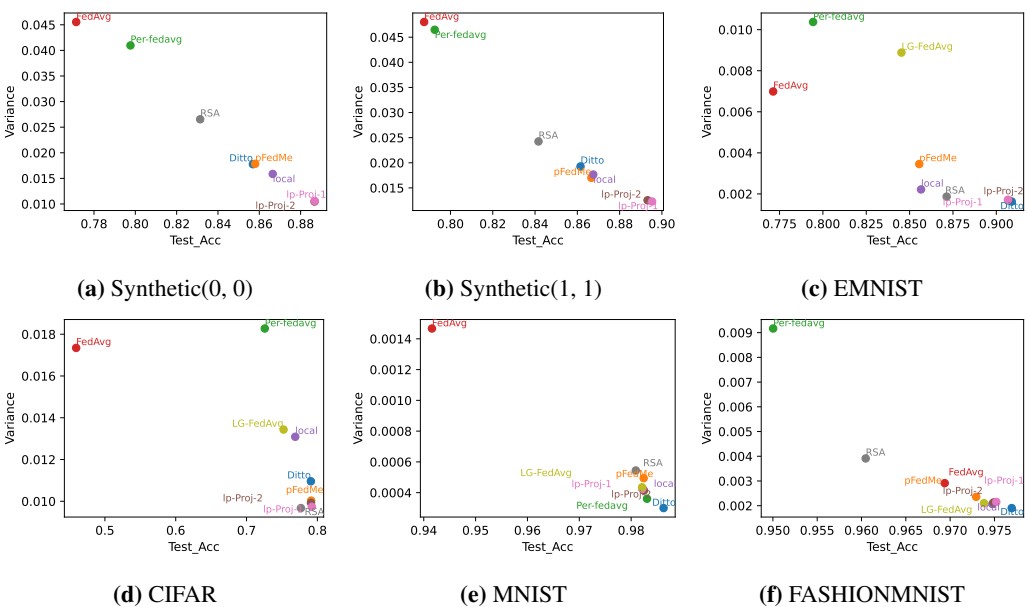

**Figure 6:** Complete results of performance fairness of competing methods. (The point closer to the bottom right corner is better.)