# OpenReview forum: "Personalized Federated Learning towards Communication Efficiency, Robustness and Fairness"
_NeurIPS.cc/2022/Conference — NeurIPS 2022 Accept_

### Official Review · Reviewer_LhiF · 2022-07-02

**Rating:** 5
**Confidence:** 3
**Soundness:** 3 good
**Presentation:** 3 good
**Contribution:** 2 fair

**Summary:**

The paper proposes a Personalized FL method which aims to achieve robustness, fairness and communication efficiency simultaneously.
They achieve fairness through personalization and regularization (i.e., $L^{p}$-regularization) similar to existing works , and propose low-dimensional random projection of the local client updates to improve communication cost. The experimental results show the improvement of these metrics compared to existing baselines.

**Questions:**

The Robustness performance of Ditto for EMNIST dataset seems a little bit surprising to me, because the results reported in Ditto for FEMNIST does not suggest this poor robustness performance on EMNIST, can you elaborate on this ? have you performed hyper parameter fine-tuning for Ditto Algo ?


**Strengths And Weaknesses:**

Strength
*  The paper appears to be theoretically sound as well as the claims seem to be supported by theoretical justifications as well as numerical experiments.
* The paper addresses three important challenges in Federated Learning Settings (Communication Efficiency, Fairness, Robustness) altogether, which seems to me that there is a gap in existing works.

Weakness
* The Novelty seems to be a little low compared to existing works ,  the embedding of local updates to lower dimension seems to be main novelty here.
* In terms of Personalization Accuracy Performance, the results are mixed between Ditto and new approach (Table 3 Appendix), however in the main body of the paper (i.e. Sec 5), it is claimed the new approach is better the competitors.
* In terms of communication performance, lack of comparison with some existing works such as the one proposed here [1].
* No experiment or discussion around the robustness of the proposed approach vs other attacks such as Backdoor attacks [2] is mentioned in the paper.

1. Federated Learning with Buffered Asynchronous Aggregation, 2022, Nguyen et al.
2. Attack of the Tails: Yes, You Really Can Backdoor Federated Learning, NeurIPS 2020, Wang et al..

---

> ### Author Response · Authors · 2022-08-02
> **Response to Reviewer LhiF**
>
> Thank you for your helpful feedback. We have answered all your concerns. In the following, we respond point by point.
>
> > The Novelty seems to be a little low compared to existing works , the embedding of local updates to lower dimension seems to be main novelty here.
>
> (A1) We not only propose the low-dimensional projection, but also provide convergence analysis and illustrate its benefits on robustness and fairness theoretically and empirically.
> To the best of our knowledge, there is no existing work considering all four issues (personalization, communication efficiency, robustness and fairness) simultaneously.
>
> > In terms of Personalization Accuracy Performance, the results are mixed between Ditto and new approach (Table 3 Appendix), however in the main body of the paper (i.e. Sec 5), it is claimed the new approach is better the competitors.
>
> (A2) Thanks for your comment and we will refine our presentation and make it more accurate in the revised paper.
> But we remark that our method is at least comparable to the existing SOTA methods, and in most cases it is superior to the competitors in terms of training loss.
>
> > In terms of communication performance, lack of comparison with some existing works such as the one proposed here [1].
>
> (A3) We feel no necessity to consider the referenced paper[1] since this paper mainly concerns scalability and privacy in FL,
> which has a different focus from our paper.
> Moreover, communication is not the goal of [1].
>
> > No experiment or discussion around the robustness of the proposed approach vs other attacks such as Backdoor attacks [2] is mentioned in the paper.
>
> (A4) In this paper, we mainly aim to consider Byzantine attacks, which are model update attacks and the adversary would directly operate on the transmitted messages.
> However, the backdoor attack proposed in [2] requires constructing an edge-case dataset with external data,
> which is different from the main concern in our paper.
> We can test it as an extension and add it to the appendix in our revised paper.
> And we speculate that the results under the backdoor attack would be similar to the data-poison attack that is already considered in our paper,
> since the black-box attack in [2] is similar to our data poison attack.
>
> In fact, we have reproduced the black-box attack and test it on `EMNIST` using `ARDIS` as the edge-case dataset as suggested in [2].
> But due to limited time of the rebuttal period, only preliminary results with 10\% clients as adversaries are available now,
> and we briefly report them as follows.
>
> | Method | Test Accuracy |
> | :-:    | :-:           |
> | Ditto | 0.826 |
> | Global+Mean | 0.144 |
> | RSA | 0.871 |
> | lp-proj-1 | 0.906 |
> | lp-proj-2 | 0.904 |
>
> Compare the above results with Table 8 in Appendix D.5,
> we see that the performance under edge-case attack is similar to that under the data-poison attack included in our paper,
> which preliminarily confirms our conjecture.
>
> > The Robustness performance of Ditto for EMNIST dataset seems a little bit surprising to me, because the results reported in Ditto for FEMNIST does not suggest this poor robustness performance on EMNIST, can you elaborate on this ? have you performed hyper parameter fine-tuning for Ditto Algo ?
>
> (A5) The adversarial attacks considered in our paper are different from those in [3].
> The most closely related attack in [3] is their *Random updates* defined in Section 2 of their paper.
> Our *Gaussian attack* (line 248) is defined in a similar way as *Random Updates*.
> Under this attack on the EMNIST dataset, we report the test accuracies of `Ditto` as 90.7\%, 67.3\%, 68.1\% and 70.3\% for the case of clean data, 20\%, 50\% and 80\% adversaries, respectively (Table 7 in Appendix D.5).
> While in [3], their reported test accuracies for the above cases are 83.6\%, 79.2\%, 74.3\%f and 67.4\%, respectively (Table 8 in Appendix D.4 in [3]).
> We think the difference in the performance is within acceptable limits
> and `Ditto` itself has already shown desired robustness under some adversarial attacks,
> while under some other attacks that have not been considered in [3], our method is better.
> Furthermore, we do have tuned all the required hyper-parameters for all competing benchmarks,
> the hyper-parameter setting and tuning details can be found in Appendix D.2.
>
> [1] Federated Learning with Buffered Asynchronous Aggregation, 2022, Nguyen et al.
>
> [2] Attack of the Tails: Yes, You Really Can Backdoor Federated Learning, NeurIPS 2020, Wang et al..
>
> [3] Tian Li, Shengyuan Hu, Ahmad Beirami, and Virginia Smith. Ditto: Fair and robust federated learning through personalization. In International Conference on Machine Learning, pages 6357–6368. PMLR, 2021.

---

### Official Review · Reviewer_vPmy · 2022-07-11

**Rating:** 6
**Confidence:** 3
**Soundness:** 4 excellent
**Presentation:** 4 excellent
**Contribution:** 3 good

**Summary:**

This paper tackles the problem of providing increasingly uniform performance across devices (fairness) through projection of local models into a shared low-dimensional subspace and infimal convolutions. The paper provides theoretical proofs for convergence and explores robustness and fairness using the proposed method. The effectiveness of this approach is demonstrated through extensive experiments on federated datasets, showing improved performance in personalization accuracy, communication efficiency, robustness and fairness.

**Questions:**

- Why does `lp-proj` help reduce the variance of performance among clients?
- Why does infimal convolution help personalized performance of the local model?

**Limitations:**

- Limited focus on how the model improves personalization performance using this method.

**Strengths And Weaknesses:**

**Strengths**:
- The paper is well-written and organized. Algorithms are easily understandable and the code is well-written.
- Thorough theoretical analysis with extensive proofs that demonstrate the convergence of `lp-proj`.
- Theoretical analysis and experiments support the method's claim to improve robustness and fairness.
- Extensive numerical experiments on strongly-convex objectives and heterogenous datasets demonstrate improved state-of-the-art personalization performance. The proposed method significantly reduces communication costs.

**Weaknesses**:
- Limited technical novelty with many of the ideas coming from `FedProx` and `pFedMe`. Note: extensive comparison analysis gives convincing evidence that the method is sufficiently different from previous work.

---

> ### Author Response · Authors · 2022-08-02
> **Response to Reviewer vPmy**
>
> Thank you for your helpful feedback.
> We have answered all your concerns.
> In the following, we respond point by point.
>
> > Limited technical novelty with many of the ideas coming from `FedProx` and `pFedMe`. Note: extensive comparison analysis gives convincing evidence that the method is sufficiently different from previous work.
>
> (A1) Firstly, our method is different from `FedProx` for the reason that `FedProx` is not designed for personalized FL.
> As a result, its optimization on the client side only involves a single level while our method requires solving a bi-level optimization problem.
> On the other hand, our proposed method is more general than `pFedMe` by introducing a projection matrix and infimal convolution.
> `pFedMe` can be viewed as a special case of `lp-proj-2` once we set the projection matrix as the identity matrix.
>
> > Why does `lp-proj` help reduce the variance of performance among clients?
>
> (A2) Intuitively, with the help of the randomly chosen linear subspace,
> `lp-proj` only requires (near) consensus for the model parameters on the subspace and leaves flexibility for the personalized model to fit the local data distribution,
> which helps improve the performance of each individual client, i.e., decrease the test loss.
> Once the performance of clients is improved simultaneously, the variance would naturally be reduced, since a lower bound for the test losses is 0.
> Theoretically, we have analyzed a case of linear models and the expectation of the variance of the clients' test losses is $\mathcal{O}(d^2)$ and $\mathcal{O}(d_{\mathrm{sub}}^2)$ for `pFedMe` and `lp-proj`, respectively  (Proposition 3 in lines 269-272).
> This implies that the dependence of the variance on the projection dimension is of squared order, which further verifies the intuition that projection helps reduce the variance of test losses among clients.
>
> > Why does infimal convolution help personalized performance of the local model?
>
> (A3) Infimal convolution is a smoothing technique which is originally used to smooth the extended real-valued convex function.
> In federated learning, infimal convolution smooths the personalized model towards the global model.
> With the help of the smoothing kernel $g$,
> we can decompose the model parameters into two parts, i.e., a personalized part and a global part.
> The global model itself aggregates the information from all the participated clients and it serves as a reliable reference point.
> Given the global model, the inner level optimization problem is equivalent to a constrained optimization problem, which aims to find a better solution that fits for the local data distribution around the reference point and hence improves personalized performance of the model.

---

> > ### Comment · Reviewer_vPmy · 2022-08-09
> > **Response to Authors**
> >
> > Thank you for your detailed answers to my questions. I have read the response and will be keeping my positive score.

---

### Official Review · Reviewer_xiRc · 2022-07-11

**Rating:** 6
**Confidence:** 4
**Soundness:** 3 good
**Presentation:** 3 good
**Contribution:** 3 good

**Summary:**

This paper formulates the goal to balance different constraints of interest (i.e., communication efficiency, robustness and fairness) simultaneously. The authors propose a personalized federated learning algorithm named lp-proj, which is supported by theoretical analysis. Extensive experimental results show the effectiveness of the proposed method.

**Questions:**

Please refer to Weakness.
I will increase my score if the authors can address my questions.


**Limitations:**

1) The theoretical analysis is based on the assumption that the objective function is strongly convex, which is very strong.
2) The fairness metric used in this paper is in terms of performance. I'm not sure whether it works well on other fairness metrics such as group fairness (DP, EO etc.).


**Strengths And Weaknesses:**

1) This paper proposes a practical problem which has not been studied widely.
2) The proposed method is technically sound and is well supported by theory.
3) Extensive experiments on three datasets show the effectiveness of the proposed algorithm.

I also have some questions:
1) (Main concern) I have noted that the authors show the results of trade-off between average performance and performance fairness. Is there a trade-off between performance and communication efficiency / robustness? How to balance them?
2) It seems that this paper simulates Non-IID settings only based on the target label. Can this work be applied to other Non-IID settings, e.g. domain shifts.
3) Is the proposed algorithm robust enough to deal with emergencies, e.g., some clients drop out in the training stage?

---

> ### Author Response · Authors · 2022-08-02
> **Response to Reviewer xiRc**
>
> Thank you for your helpful feedback. We have answered all your concerns. In the following, we respond point by point.
>
> > (Main concern) I have noted that the authors show the results of trade-off between average performance and performance fairness. Is there a trade-off between performance and communication efficiency/robustness? How to balance them?
>
> (A1) For communication efficiency:
> If we use a smaller projection dimension $d_{\mathrm{sub}}$, fewer bytes need to be transmitted between the server and the clients.
> However, with a smaller projection subspace, useful information that can be leveraged from other clients would be less.
> We claim that with an optimal regularization coefficient $\lambda$, the test loss of linear models is monotonically decreasing in the projection dimension, i.e., we have a similar conclusion with Theorem 3 in [1].
> In fact, this conclusion has already been implicitly implied by our analysis in Section 4.2, and can be visualized in Figure 1.
> `pFedMe` can be viewed as a special case of our method with the projection matrix $P$ chosen as the identity matrix and the projection dimension as $d$, we then see that when $\lambda$ is optimally tuned ($\lambda = \lambda_1^*$), the test loss is smaller with a larger projection dimension.
> Nevertheless, finding the best regularization is hard in practice, and we show the trade-off between performance and communication efficiency via numerical experiments in Table 16 in Appendix D.7.4. From the presented results, we see the performance only varies moderately as $d_{\mathrm{sub}}$ changes.
> Therefore, we suggest choosing the appropriate $d_{\mathrm{sub}}$ based on the computing resource and communication bandwidth in practice.
>
> (A2) For robustness, there is no doubt that we would obtain worse performance when the intensity of the adversarial attacks gets stronger.
> And we think it is hard to reach a balance/trade-off between performance and robustness in general, since the type and the intensity of the adversarial attacks could be arbitrary.
> In the three Byzantine attacks considered in our paper, the variance $\tau^2$ defined in lines 244-248 can be used to quantify the intensity of the attacks in our analysis, and we can see that the test loss on benign clients is linearly correlated to the attack intensity (please refer to line 1239 (rebuttal revision) in Appendix C as an example).
> This implies that analytically with stronger attacks, the performance would be worse off, and experimentally we can see that our proposed method is robust enough for various types of attacks.
>
> > It seems that this paper simulates Non-IID settings only based on the target label. Can this work be applied to other Non-IID settings, e.g. domain shifts.
>
> (A3) The Non-IID setting in FL refers to statistical heterogeneity among clients. Besides label skewness, we also considered another kind of statistical heterogeneity as an extension, i.e., data volume skewness, which means that different clients may not have equal sizes of the dataset. In particular, we simulate that the number of samples among clients follows a power law.  The experimental results can be found in Appendix D.7.1. Moreover, due to our limited knowledge, could you please explain more in detail what you mean by domain shifts in the context of FL?
>
> > Is the proposed algorithm robust enough to deal with emergencies, e.g., some clients drop out in the training stage?
>
> (A4) In our current experiments, we uniformly randomly sample a fraction of clients every round for communication, which means that we do not need all the clients to keep in touch all the time. If the clients drop out completely at random, the analysis and the experimental results would be exactly the same in our current settings. However, if the drop-out mechanism is not at random, then it would lead to biased gradient evaluation. If there is no advanced technique to solve the bias gradient issue, no meaningful convergence could be provided. And it is beyond the scope of our current paper.
>
> > The theoretical analysis is based on the assumption that the objective function is strongly convex, which is very strong.
>
> Please see "Response to all reviewers".
>
> > The fairness metric used in this paper is in terms of performance. I'm not sure whether it works well on other fairness metrics such as group fairness (DP, EO etc.).
>
> (A5) Besides performance fairness, we also consider collaboration fairness as an extension in Appendix D.7.2. In terms of collaboration fairness, our proposed method is also comparable to other competing benchmarks. Furthermore, we do not aim at achieving group fairness in this work, but we think that our proposed approach can be used as a building block and combined with other methods for group fairness to obtain more comprehensive performance.
>
> [1] Nakkiran, P., Venkat, P., Kakade, S. M., \& Ma, T. (2020, September). Optimal Regularization can Mitigate Double Descent. In International Conference on Learning Representations.

---

> > ### Comment · Reviewer_xiRc · 2022-08-09
> > **Response to authors**
> >
> > Thanks for your explanation! I have read the response and my concerns have been addressed.

---

> > > ### Author Response · Authors · 2022-08-09
> > > **Further Response to Reviewer xiRc**
> > >
> > > Thanks for your acknowledgement that our explanation has addressed all your concerns!
> > >
> > > We would appreciate it very much if you can increase your score,
> > > because you have ever promised in the initial review that "I will increase my score if the authors can address my questions".

---

### Official Review · Reviewer_RhCc · 2022-07-12

**Rating:** 5
**Confidence:** 2
**Soundness:** 3 good
**Presentation:** 2 fair
**Contribution:** 3 good

**Summary:**

This work aims to tackle three problems in FL: 1) communication efficiency, 2) robustness (Byzantine attacks) and 3) fairness. Specifically, this work projects local models into a shared-and-fixed low-dimensional random subspace and uses infimal convolution to control the deviation between the reference model and projected local models. Theoretically, it shows convergence on strongly convex objectives with square regularizers and mild dimension reduction. Numerical results are presented on synthetic data, MNIST, Cifar10 with MLP and CNN.

**Questions:**

1. How do we generalize the theoretical results to neural network training (non-convex)?
2. What is the key technical novelty of using the proposed projection vs sketching-based efficient FL methods?
3. As the proposed methods seem easy to implement, would it be possible to test them on more complex tasks to make the methods more convincing?

**Ethics Review Area:**

["Discrimination / Bias / Fairness Concerns"]

**Limitations:**

Yes

**Strengths And Weaknesses:**

Strength:
1. The problems addressed by this work are very important.
2. Theoretically and empirically show the advantage vs. the existing methods
3. Extensive experiments are conducted, including comparison and different variants of experiment settings.

Weaknesses:
1. The presentation of this paper is not easy to digest as multiple components are delivered without sufficient details and support. The authors also include "due to space limitation." I would suggest the authors consider the key messages to deliver in the conference paper.
2. The theoretical results seem interesting. However, the assumptions (strong convexity and smoothness) are relatively strong and have a clear gap between the real deep neural network training scenarios.
3. One related work could be included in the comperison:
Rothchild, D., Panda, A., Ullah, E., Ivkin, N., Stoica, I., Braverman, V., ... & Arora, R. (2020, November). Fetchsgd: Communication-efficient federated learning with sketching. In International Conference on Machine Learning (pp. 8253-8265). PMLR.

---

> ### Author Response · Authors · 2022-08-02
> **Response to Reviewer RhCc**
>
> Thank you for your helpful feedback. We have answered all your concerns. In the following, we respond point by point.
>
> > The presentation of this paper is not easy to digest as multiple components are delivered without sufficient details and support. The authors also include "due to space limitation." I would suggest the authors consider the key messages to deliver in the conference paper.
>
> (A1) Our key messages have been specified in lines 52-54.
> Since we evaluate our methods in three aspects, we hope you could understand there are many aspects we need to consider and any space is greatly valuable.
> We defer a lot of details to the appendix.
> However, thanks for your suggestions.
> We will add more details and refine our presentation in our revised paper.
>
> > The theoretical results seem interesting. However, the assumptions (strong convexity and smoothness) are relatively strong and have a clear gap between the real deep neural network training scenarios.
>
> (A2) The smoothness assumption can lead to a smaller condition number of $F_k$, but is indeed not necessary. We have clarified this in our revised paper (please see Remark 1 in Appendix A.2 on page 17 of the rebuttal revision).
> As for the strong convexity assumption, we would like to compare our assumptions and results with [1].
> In [1], the authors consider two cases: (i) strongly convex and (ii) non-convex and smooth.
> We claim that the analyses for these two cases do not vary too much and at least for strongly convex cases, there is no gap between theoretical and experimental results.
> For the details, please see "Response to all reviewers".
>
> > One related work could be included in the comperison: Rothchild, D., Panda, A., Ullah, E., Ivkin, N., Stoica, I., Braverman, V., ... \& Arora, R. (2020, November). Fetchsgd: Communication-efficient federated learning with sketching. In International Conference on Machine Learning (pp. 8253-8265). PMLR.
>
> (A3) Thank you for this comment and we can add this related work as a competing benchmark in our revised paper.
> We have reproduced the algorithm of `Fetchsgd`,
> but due to limited time of the rebuttal period,
> only preliminary results are available now, and we briefly report them as follows.
>
> | Dataset | Bytes Budget | Test Acc | Target Acc | Used Bytes | Size of the Sketch |
> | :----:  | :-:          | :-:      | :-:        | :-:        | :-: |
> | `Synthetic(0,0)` | 328020 | 0.196 | 0.6 | $\star$ | 31 $\times$ 6 |
> | `MNIST` | 228000 | $\star$ | 0.7 | 698651200 | 1590 $\times$ 795 |
>
> Compare the above results with Table 4 in Appendix D.4,
> we can see that the performance of `Fetchsgd` is similar to `sketched-sgd` (a benchmark has been considered in our paper) in our settings,
> where the latter is slightly better.
> We will perform more comprehensive comparison in the following period.
>
> > How do we generalize the theoretical results to neural network training (non-convex)?
>
> Please see "Response to all reviewers".
>
> > What is the key technical novelty of using the proposed projection vs sketching-based efficient FL methods?
>
> (A4) Sketching-based methods project the model parameters every time with a random basis,
> which only aims to reduce the size of the transmitted messages.
> On the other hand, our proposed projection determines the low-dimensional random subspace before training the model and it is fixed throughout the training session.
> The fixed subspace brings out not only communication-efficiency, but also robustness and fairness.
> We analyze these two important properties from both theoretical and experimental perspectives,
> which have not been carried out in sketching-based FL methods.
>
> > As the proposed methods seem easy to implement, would it be possible to test them on more complex tasks to make the methods more convincing?
>
> (A5) Although the idea of our proposed method is intuitive and easy to understand,
> the implementation is not so easy for the reason that the projection involves a large-scale matrix multiplication.
> It needs huge computation power and storage space, if the size of the model and the dataset is enormous.
> Our numerical results show that at least on medium-sized datasets and widely-used machine learning models, the proposed algorithm shows comparable or superior to existing works.
> Our comparison is fair with literature, for example,
> we mainly follow [2] (using similar settings but considering more baseline algorithms).
> Application to more complex tasks requires more sophisticated programming and engineering skills and we leave it for future work.
>
> [1] Canh T Dinh, Nguyen H Tran, and Tuan Dung Nguyen. Personalized federated learning with moreau envelopes. In Advances in Neural Information Processing Systems, volume 33, pages 21394–21405, 2020
>
> [2] Tian Li, Shengyuan Hu, Ahmad Beirami, and Virginia Smith. Ditto: Fair and robust federated learning through personalization. In International Conference on Machine Learning, pages 6357–6368. PMLR, 2021.

---

> > ### Comment · Reviewer_RhCc · 2022-08-04
> > **Thanks for your response**
> >
> > Thanks for your response. Some of my concerns have been addressed. Please consider my remaining concerns as part of our further discussion.
> >
> > > Practical value of the proposed methods
> >
> > Your algorithm is neat, and I appreciate the three benefits it offers. As you noted in your rebuttal, "the implementation is not so easy for the reason that the projection involves a large-scale matrix multiplication," especially when "the size of the model and the data set are enormous." Although promising numerical results were presented, as a methodology paper in FL (a practical field with many large-scale applications), the validity of the method in practice is extremely important. The paper conceals a disadvantage of high computational costs. A comparison of the proposed method's computation overhead with the selected baseline would be fairer. Readers will hopefully be better able to understand the method and know the limitations if they choose to use it.
> >
> > > Clarification and missing details
> >
> > Thanks for being willing to revise your paper. It is understood that there is a problem with space in this paper since three merits are emphasized. I still have the same minor concern. The fairness session, for example, is very short. In order to help readers get quick takeaways, a discussion of the results (as you did for communication efficiency and robustness experiments) would be helpful. It is also difficult to locate a pointer to the evaluation setup. Could the authors provide some insight on how these issues might be addressed?
> >
> > +: The font size of Fig 2 could be enlarged. The labels are a little hard to read.
> >
> > > Extension to non-convex setting
> >
> > Thanks for sharing your thoughts and adding the proof in the appendix. The overall idea sounds reasonable but I will take more time to look into it and get back to this later.

---

> > > ### Author Response · Authors · 2022-08-07
> > > **Further Response to Reviewer RhCc**
> > >
> > > Thanks for your response. Here we would answer your further concerns point by point.
> > >
> > > > Computational complexity of the proposed algorithm
> > >
> > > The computation time complexity of our proposed algorithm is $\mathcal{O}(|\tilde{\mathcal{D}}| d + d_{\mathrm{sub}} d)$ per iteration, while that of `pFedMe` ([1]) is $\mathcal{O}(|\tilde{\mathcal{D}}| d)$, where $|\tilde{\mathcal{D}}|$ is the batch-size for each iteration (for simplicity, we drop the subscript $k$ used in the main text to specify the $k$-th client).
> > > To reach an $\varepsilon$-accurate solution, the required communication rounds and inner loops are of the same order for both `lp-proj` and `pFedMe`.
> > > Therefore, given $d_{\mathrm{sub}} < |\tilde{\mathcal{D}}|$, our proposed algorithm shares the same computation time complexity as that of `pFedMe`.
> > >
> > > On the other hand, the space complexity of `pFedMe` is $\mathcal{O}(d)$, while in our current implementation of `lp-proj`, the space complexity is $\mathcal{O}(d_{\mathrm{sub}} d)$ since we store the whole projection matrix for convenience.
> > > However, if we record the random seed used to generate the projection matrix and regenerate it when we start to perform matrix multiplication, the best space complexity of `lp-proj` is reduced to $\mathcal{O}(d)$, the same order as `pFedMe`.
> > >
> > > Therefore, the large space cost is a drawback of our current implementation, not a drawback of our proposed methodology.
> > > There might exists other tricks to improve performance of our methods in large dataset and big models, which is however out of the scope of our current work but we believe is promising.
> > >
> > > Finally, we want to stress that in our current experiments, we already test our methods with many competitive algorithms in different neural networks and datasets.
> > > Our key message is low-dimensional projection has multiple benefits of communication efficiency, robustness and fairness.
> > > The results are promising and valid at least on medium-sized datasets and models.
> > >
> > > > Presentation of the paper
> > >
> > > Thanks for your suggestion on the revision of the fairness session.
> > > Due to potential violation of 9-page space limit, we didn't revise our main text directly.
> > > But we provide revised paragraphs in Appendix E.1 (in our updated revision of the appendix).
> > > We detail the evaluation setup and discuss our method with other SOTA methods there.
> > > The experiment results matches our theory (i.e., Proposition 3).
> > >
> > > Furthermore, we have enlarged the font size of Figure 2 and update the rebuttal revision, hopefully it is clearer to read.
> > >
> > > > Extension to non-convex setting
> > >
> > > Thanks for your comment and we are looking forward to your feedback!
> > >
> > > [1] Canh T Dinh, Nguyen H Tran, and Tuan Dung Nguyen. Personalized federated learning with moreau envelopes. In Advances in Neural Information Processing Systems, volume 33, pages 21394–21405, 2020

---

> > > > ### Comment · Reviewer_RhCc · 2022-08-09
> > > > **Thanks!**
> > > >
> > > > Thank you for the complexity analysis, and I've read your new proof.  I will keep my score as the complexity/cost issues seem to be a point worth improvement.

---

### Official Review · Reviewer_yNQy · 2022-07-15

**Rating:** 4
**Confidence:** 3
**Soundness:** 2 fair
**Presentation:** 3 good
**Contribution:** 2 fair

**Summary:**

This paper proposes a personalized federated learning (PFL) method to achieve communication efficiency, robustness and fairness, by projecting local models into a shared-and-fixed low-dimensional random subspace and using infimal convolution to control the deviation between the reference model and projected local models. Theoretical analysis about strongly convex objectives is provided. Experiments on both synthetic data and real data are used for evaluation.

**Questions:**

1. Does the method still converge for non-convex objectives?
2. How is the performance on widely used deep models like ResNet and larger datasets like ImageNet?


**Limitations:**

The authors have adequately addressed the limitations and potential negative societal impact of their work.

**Strengths And Weaknesses:**

Strengths:
1. The studied problem about designing method to achieve communication efficiency, robustness and fairness in PFL is interesting and important.
2. The paper is well written with good organization.
3. Experiments show that the proposed method does achieve good performance on the adopted models and datasets.

Weaknesses:
1. The theoretical analysis is weak due to some ideal assumptions. First, only results for strongly convex objectives are provided. But most models in real practice are non-convex. Second, to analyze robustness and fairness (Section 4.2), the authors assume that the number of local update steps is infinite, which is unreasonable because in non-convex cases the algorithm will not converge if the number of local update steps is set to be infinite.
2. The datasets are relatively small.
3. The models, especially deep models, seem to be specially designed. Why not just adopt some widely used models like ResNet?

---

> ### Author Response · Authors · 2022-08-02
> **Response to Reviewer yNQy**
>
> Thank you for your helpful feedback.
> We have answered all your concerns.
> In the following, we respond point by point.
>
> > The theoretical analysis is weak due to some ideal assumptions.
> First, only results for strongly convex objectives are provided. But most models in real practice are non-convex. Second, to analyze robustness and fairness (Section 4.2), the authors assume that the number of local update steps is infinite, which is unreasonable because in non-convex cases the algorithm will not converge if the number of local update steps is set to be infinite.
>
> (A1) In terms of the theoretical analysis for the non-convex case, please see "Response to all reviewers".
>
> (A2) As for the results in Section 4.2, our analysis is inspired by Section 3.3 and Appendix A.2 in [2]. The focus of this subsection is federated linear regression problems, whose objectives are naturally convex. Then it is reasonable to assume the number of local update steps is infinite since we can calculate the minimizer of the inner loop objective $\tilde{h}_k$ directly. We aim to provide some intuition through analysis on such a simplified setting and then illustrate the benefits of our method through extensive experiments.
>
> > The datasets are relatively small.
>
> (A3) The used datasets are at least medium-sized and they are widely adopted in the FL literature including [1][2].
> Furthermore, our proposed method needs to perform a projection (which is essentially a large matrix multiplication) on the client-side every time before communication.
> For large models or large datasets,
> the involved matrix is so large that challenges arise given limited computation and storage sources.
> However, within our computation sources, we have conducted extensive experiments which consider a large number of variables, non-exclusively including other competitive algorithms, important hyperparameters, datasets, models, and adversary attacks.
> We hope you can take these aspects into consideration.
>
> > The models, especially deep models, seem to be specially designed. Why not just adopt some widely used models like ResNet?
>
> (A4) We DO have tested ResNet in our experiments.
> For the FASHIONMNIST dataset, we apply a ResNet9 model,
> whose detailed network structure is shown in lines 1372-1383 in Appendix D of the rebuttal revision (or lines 1177-1188 of the original version).
> The numerical results are shown in Table 3-8 in Appendix D.
> With ResNet, our algorithm is also comparable or superior to the competing benchmarks as stated in the main body.
>
> > Does the method still converge for non-convex objectives?
>
> Please see "Response to all reviewers".
>
> > How is the performance on widely used deep models like ResNet and larger datasets like ImageNet?
>
> Please see (A3) and (A4).
>
> [1] Canh T Dinh, Nguyen H Tran, and Tuan Dung Nguyen. Personalized federated learning with moreau envelopes. In Advances in Neural Information Processing Systems, volume 33, pages 21394–21405, 2020
>
> [2] Tian Li, Shengyuan Hu, Ahmad Beirami, and Virginia Smith. Ditto: Fair and robust federated learning through personalization. In International Conference on Machine Learning, pages 6357–6368. PMLR, 2021.

---

> > ### Comment · Reviewer_yNQy · 2022-08-09
> > **Some concerns are still not addressed**
> >
> > Thanks for the response.
> >
> > I need more time to check the correctness of the proof. Here, I give some quick response to other concerns.
> >
> > The authors do have tested ResNet in experiments. However, the ResNet model in the paper is also specially designed, which is not one the widely used versions of ResNet. The original statement in the paper is "The neural network used for FASHIONMNIST dataset ...  is modified from ResNet, which ...".
> >
> > About the scale of datasets,  the authors stated in the response that "Furthermore, our proposed method needs to perform a projection (which is essentially a large matrix multiplication) on the client-side every time before communication. For large models or large datasets, the involved matrix is so large that challenges arise given limited computation and storage sources". I think this cannot be a suitable reason for not evaluating on larger datasets with larger models. Actually, the extra cost caused by the projection can be seen as a shortage of the proposed method.

---

> > > ### Author Response · Authors · 2022-08-09
> > > **Further Response to Reviewer yNQy**
> > >
> > > 1) Since the birth of ResNet, countless variants have been proposed in the literature.
> > > It is impossible to test them all.
> > > Moreover, the key of ResNet is the Residual Block, which has been considered in our experiments.
> > > The ResNet used in our experiment is a lightweight variant of the seminal work [1]. The reason why we consider such neural network lies in the nature of federated learning, that each client device (e.g., your smart phone) only have limited compute power and storage space, and it is more practical to consider lightweight models.
> > >
> > > 2) In our implementation of the algorithm, we store the whole projection matrix $P$ for convenience, which consumes large storage space.
> > > However, we claim that the large space cost is a drawback of our current implementation, not a drawback of our proposed methodology.
> > > The space complexity of our proposed method can be reduced to the same as `pFedMe`([2]) with suitable programming tricks (which is however out of the scope of our current work but we believe is promising for further application).
> > > For more details, please refer to "Computational complexity of the proposed algorithm" of "Furthrer Response to Reviewer RhCc".
> > >
> > > 3) Finally, we want to emphasize that the key message of our paper is low-dimensional projection has multiple benefits of communication efficiency, robustness, and fairness.
> > > To support it, we not only provide theoretical analysis, but also perform many numerical experiments on datasets and models commonly used by the ML community.
> > > Extensive comparison with various baselines shows that our method is valid on medium-sized models and common datasets.
> > >
> > > [1] He K, Zhang X, Ren S, et al. Deep residual learning for image recognition[C]//Proceedings of the IEEE conference on computer vision and pattern recognition. 2016: 770-778.
> > >
> > > [2] Canh T Dinh, Nguyen H Tran, and Tuan Dung Nguyen. Personalized federated learning with moreau envelopes. In Advances in Neural Information Processing Systems, volume 33, pages 21394–21405, 2020

---

### Author Response · Authors · 2022-08-02
**Response to all reviewers**

We would like to thank all the reviewers for their constructive comments. Since three reviewers ask about the convergence analysis for the non-convex case, we reply here in a unified manner.

In terms of the convergence analysis, we would like to compare our results with those in [1].
Although [1] provides convergence results for both strongly convex and non-convex objectives, the analyses for these cases do not vary too much.
Note that the algorithm `pFedMe` in [1] corresponds to `lp-proj-2` with the projection matrix $P$ equal to the identity matrix.
As long as the regularization parameter $\lambda$ is sufficiently large, the objective function of the inner loop $\tilde{h}_k$ defined in Eqn.(4) below line 147 becomes a strongly convex function of $x_k$.
As a consequence, the analysis for the inner loop (line 7 of Algorithm 1) remains the same no matter whether $\tilde{f}_k$ is strongly convex or non-convex.
The subsequent analysis for the non-convex case is just a combination of the classical techniques and the key lemmas shared by both strongly convex and non-convex objectives.

Nevertheless, for `lp-proj-2`, the projection matrix $P$ is a $d_{\mathrm{sub}} \times d $ matrix with $d_{\mathrm{sub}} \ll d$.
Then $\tilde{h}_k$ is only strongly convex on a low-dimensional subspace but still non-convex on the whole space. This poses a challenge to the analysis for the inner loop.
We can address it in the strongly convex case as our paper currently claims.
However, for the non-convex case, we need additional assumptions to give meaningful convergence.
For the details, please see Appendix A.6 on page 24 of the rebuttal revision.
Our Theorem 2 in lines 832-839 is
similar to Theorem 2 in [1] for the non-convex case.
Moreover, we can carefully tune the parameters
to achieve any given accuracy. For the details, please refer to Appendix B.3 on page 37 of the rebuttal revision.

It is worth highlighting that we provide experiments for the strongly convex objectives.
We test our methods on logistic models and two datasets `Synthetic(0, 0)` and `Synthetic(1, 1)` (listed in Table 2 in Appendix D).
Though the problem is convex, the objective function becomes strongly convex with weight decay.
For strongly convex cases, our theoretical results and experimental performance already match.
For non-convex cases, we conduct extensive experiments to test our methods.
Similar convergence patterns have been observed in non-convex cases.

[1] Canh T Dinh, Nguyen H Tran, and Tuan Dung Nguyen. Personalized federated learning with moreau envelopes. In Advances in Neural Information Processing Systems, volume 33, pages 21394–21405, 2020

---

### Meta-Review · Area_Chair_HnSy · 2022-09-13

**Recommendation:** Accept
**Confidence:** Certain

**Metareview:**

The paper gives an approach to personalized federated learning, which incorporates client heterogeneity, robustness to attacks and fairness in performance. The proposed method is interesting, though the simultaneous incorporation of multiple concerns makes it somewhat difficult to disentangle the role of different choices. The authors present theoretical results and a fairly comprehensive empirical evaluation showing effectiveness of their method.

The accept decision is primarily based on the substantial empirical analysis. As mentioned above, various algorithmic considerations should ideally be studied somewhat in isolation, such as through ablations. For instance, it is completely reasonable to incorporate personalization through the infimal convolution without a subspace projection. Presumably the authors do not do this due to the computational cost, but the joint effect of P and the personalized formulation is a bit tricky for the reader.

For the theory, assumption 1(a) is effectively vacuous in most ML settings, where the loss \tilde{f}_k is typically based on one or a few examples only, and hence cannot be strongly convex without including additional regularization in the loss. I appreciate the addition of the non-convex result in the revision, and would advise changing this to the main result. The per example strong convexity assumed here is not common in the literature at all (typically it is average loss strong convexity), so this result adds little value to the paper.

The reviewers also highlighted multiple weaknesses in the evaluation and method, such as scalability in practice due to the projection step and the use of non-standard models in evaluation. These should be properly acknowledged in the revision.

**Award:**

No

---

### Decision · Program_Chairs · 2022-09-14

Accept